# Mitigating Non-IID Drift in Zeroth-Order Federated LLM Fine-Tuning with Transferable Sparsity

**Yide Ran[1], Wentao Guo[2], Jingwei Sun[3], Yanzhou Pan[4], Xiaodong Yu[1], Hao Wang[1], Jianwen Xie[5], Yiran Chen[6], Denghui Zhang[1], Zhaozhuo Xu[1]***

[1]Stevens Institute of Technology, [2]Princeton University, [3]University of Florida,
[4]Google LLC, [5]Lambda, [6]Duke University

{yran1,xyu38,hwang9,dzhang42,zxu79}@stevens.edu
wg0420@princeton.edu, sun.jingwei@ufl.edu, yanzp@google.com
jianwen.xie@lambdal.com, yiran.chen@duke.edu

## Abstract

Federated Learning enables collaborative fine-tuning of Large Language Models (LLMs) across decentralized Non-Independent and Identically Distributed (Non-IID) clients, but such models' massive parameter sizes lead to significant memory and communication challenges. This work introduces MEERKAT, a sparse zeroth-order optimization (ZO) method designed for federated LLM fine-tuning. By limiting fine-tuning to a transferable, static, extremely sparse subset of parameters, MEERKAT achieves remarkable communication efficiency, enabling cost-effective high-frequency synchronization. With theoretical analysis and experiments, we show that this high-frequency communication effectively mitigates Non-IID data challenges and leads to superior performance compared to full-parameter ZO. Furthermore, experimental results show that MEERKAT outperforms existing sparsity baselines with better performance at the same communication frequency. To further handle Non-IID drift, MEERKAT leverages traceable local updates and forms a *virtual path* for each client. This virtual path mechanism reveals the GradIP phenomenon: the inner products between LLM pre-training gradients maintained by server and client gradients estimated via ZO converge for extreme Non-IID clients but oscillate for IID ones. This distinct behavior provides a signal for identifying clients with extreme data heterogeneity. Using this signal, MEERKAT-VP is proposed to analyze GradIP trajectories to identify extreme Non-IID clients and applies early stopping to enhance aggregated model quality. Experiments confirm that MEERKAT and MEERKAT-VP significantly improve the efficiency and effectiveness of ZO federated LLM fine-tuning.

## 1 Introduction

Federated Learning (FL) (McMahan et al., 2017) has emerged as a powerful paradigm for enabling decentralized collaboration, particularly relevant for fine-tuning Large Language Models (LLMs) across numerous client devices (Dubey et al., 2024; Brown et al., 2020). Unlike centralized training, FL allows clients to train models locally and share only model updates with a central server. However, fine-tuning LLMs in a FL setting faces two major challenges: the massive model parameter size and the Non-Independent and Identically Distributed (Non-IID) data distribution across clients. The former leads to high computation demands on clients and significant communication overhead, while the latter causes client drift and hinders global convergence. These challenges make LLM fine-tuning impractical on resource-constrained clients and hinder the effective use of decentralized data.

Zeroth-order Optimization (ZO) provides a promising avenue for addressing some of these challenges in federated LLM fine-tuning. By estimating gradients through model perturbations and forward passes, ZO bypasses the need for backpropagation and the storage of intermediate activations, leading

---

*Corresponding author: zxu79@stevens.edu

to more memory-efficient learning on client devices (Zhang et al., 2021; Fang et al., 2022; Ling et al., 2024; Liu et al., 2024; Malladi et al., 2023). However, applying standard ZO directly to the massive parameter space of LLMs can still be computationally inefficient and the optimization process unstable (Malladi et al., 2023). Moreover, adapting ZO for federated LLM fine-tuning remains challenging, particularly in balancing computational efficiency, communication overhead, and model performance under Non-IID data heterogeneity.

In order to address the above challenges, we propose MEERKAT, a sparse ZO method designed for efficient federated LLM fine-tuning. MEERKAT addresses the computational and communication burdens by focusing ZO updates on a static, extremely sparse (less than $0.1\%$), and transferable subset of LLM parameters. This subset is strategically identified using gradients derived from pre-training data, ensuring that updates target parameters most sensitive to the loss function. This selective approach dramatically reduces communication overhead and supports cost-effective high-frequency synchronization. As we will demonstrate through theoretical analysis and extensive experiments, the combination of high communication frequency and sparsity in MEERKAT enables frequent yet lightweight synchronization. This effectively reduces the convergence error floor in theory and practice, leading to consistently superior performance compared to full-parameter ZO fine-tuning and other sparsity methods under the same communication frequency.

Leveraging MEERKAT's efficient high-frequency synchronization to effectively mitigate Non-IID data challenges, we further enhance its adaptability to weak network conditions. By employing a virtual path mechanism to track client updates, we enable the server to analyze client training dynamics without accessing raw data, thus facilitating robust operation even when frequent direct communication is constrained. Within this virtual path, we observe the **GradIP phenomenon**, a pattern revealed by the GradIP score, which computes the inner product between local client gradients estimated via ZO and server pre-training gradients. GradIP scores converge for Non-IID clients while oscillating for IID clients, serving as a clear indicator of data heterogeneity. Leveraging this insight, we propose MEERKAT-VP that introduces a virtual path client selection method to identify clients with significant Non-IID characteristics and apply early stopping, thereby reducing their adverse impact on the aggregated model and enhancing its quality.

In summary, this paper makes the following contributions:

- **Performance Improvement with Sparsity.** Meerkat consistently outperforms full-parameter ZO optimization in both IID and Non-IID settings, demonstrating the effectiveness of our sparse update strategy. Extensive experiments show that Meerkat surpasses not only full-parameter ZO but also other sparse methods, such as LoRA and weight-magnitude, achieving superior performance.
- **High Frequency Communication with Sparsity Can Lower the Error Floor.** MEERKAT leverages extreme model sparsity to reduce local computational memory. Exchanging scalar gradients drastically decreases communication costs, enabling high-frequency communication.
- **Traceable Local Updates and GradIP Phenomenon**: MEERKAT leverages traceable sparse local updates and forms a *virtual path*. The virtual paths reveal the GradIP phenomenon: the inner product between LLM pre-training gradients maintained by server and client gradients estimated via ZO converges for extreme Non-IID clients but oscillates for IID ones. This distinct behavior serves as a signal for detecting clients with extreme data heterogeneity.
- **MEERKAT-VP: Early Stopping for Extreme Non-IID Clients.** Leveraging the GradIP phenomenon via virtual path client selection, MEERKAT-VP effectively manages extreme Non-IID clients, by early stopping these clients to improve global model quality.
- **Theoretical and Experimental Validation.** We present theoretical analysis and extensive experiments across diverse FL settings, validating the scalability and performance benefits of both MEERKAT and MEERKAT-VP.

## 2 SPARSE ZEROTH-ORDER OPTIMIZATION FOR FEDERATED LLM FINE-TUNING

This section introduces MEERKAT, a sparse ZO method for federated LLM fine-tuning, and its upgraded version, MEERKAT-VP, which incorporates Virtual Path Client Selection (VPCS) strategy. This strategy leverages the traceable virtual path of client local updates to identify clients with extremely Non-IID data and applies early stopping to mitigate their adverse impact on global model convergence. We first introduce the technical details of MEERKAT, as illustrated in Figure 1, and

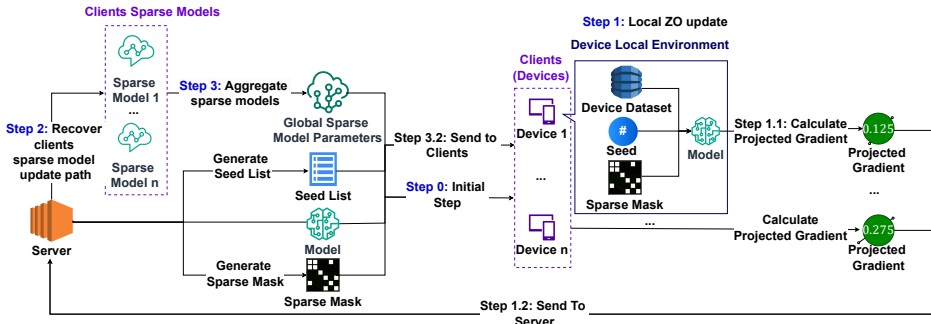

Figure 1: MEERKAT: Sparse zeroth-order optimization for federated LLM fine-tuning workflow.

subsequently describe MEERKAT-VP, shown in Figure 5. We then present theoretical convergence analysis for both methods and discuss their strengths in terms of cost-effectiveness, traceability, and the use of early stopping to mitigate client drift caused by Non-IID data.

## 2.1 MEERKAT: EXTREME SPARSE ZEROTH-ORDER FEDERATED LLM FINE-TUNING

**Sparse ZO On-Device LLM Fine-Tuning.** MEERKAT performs sparse ZO for LLM fine-tuning on the client device. Let $\mathcal{D}$ denote the client dataset we would like an LLM to fine-tune with loss function $f$. Given the LLM weight $\mathbf{w} \in \mathbb{R}^d$, we perform an iterative optimization by randomly sampling a batch $\mathcal{B} \subset \mathcal{D}$ for each step and performing the local update step as

$$g = \frac{f(\mathbf{w} + \epsilon(\mathbf{z} \odot \mathbf{m}); \mathcal{B}) - f(\mathbf{w} - \epsilon(\mathbf{z} \odot \mathbf{m}); \mathcal{B})}{2\epsilon}, \quad \hat{\nabla} f = g\,(\mathbf{z} \odot \mathbf{m})\,. \tag{1}$$

where $\mathbf{z} \in \mathbb{R}^d$ is a random vector sampled from a Gaussian distribution $\mathcal{N}(0, I_d)$, $\epsilon \in \mathbb{R}$ is the perturbation magnitude, and $\mathbf{m} \in \{0, 1\}^d$ is a binary sparse mask with density ratio $u$ that selects a subset of parameters for updates.

**Extremely Sparse Parameters Obtained from Pre-Training.** According to the formulation in equation 1, we focus the perturbation of the LLM on a subset of parameters determined by a binary mask $\mathbf{m}$. The mask $\mathbf{m}$ is derived from the pre-training process of the LLM. We compute the average squared gradients of each parameter over a subset of the C4 dataset (Raffel et al., 2020). Then, we select the top $u$ parameters with the highest average squared gradient values and mark them as $1$ in $\mathbf{m}$. In practice, we set $u$ to $0.1\%$, resulting in extremely sparse updates.

**FL with MEERKAT.** The workflow of MEERKAT is illustrated in Figure 1 and Algorithm 2. MEERKAT first loads each client with the pre-trained weight $\mathbf{w}_0$ and the sparse mask $\mathbf{m}$. Next, MEERKAT initializes a random seed list $\{s_1^1, \ldots, s_1^T\}$ at the server to generate the random Gaussian vector $\mathbf{z}$ for each local step in the first round. Next, MEERKAT performs an iterative federated optimization with $R$ rounds of client-server synchronization with each round as follows.

(1) *Local ZO update at each client.* Upon receiving global model weights $\mathbf{w}_{r-1}$ and seed list $\{s_r^1, \ldots, s_r^T\}$ from the server, each client performs $T$ local iteration steps. In each local step $t$, the client perturbs the model parameters selected by $\mathbf{m}$ with the random vector $\mathbf{z}_k^t$ generated by the random seed $s_r^t$. Each client then computes projected gradient $g_k^t$ (a scalar) according to equation 1. Using $g_k^t$, each client calculates the local gradient $\hat{\nabla} f_k^t$ and updates the local model $w_k$ with learning rate $\eta$. After $T$ local steps, each client uploads a list of projected gradients $\{g_k^1, g_k^2, \ldots, g_k^T\}$ to the server. (2) *Server reconstructs client update with virtual path.* Since the server shares the same random seed list with clients for the round, it can reconstruct each client's local model update path upon receiving their projected gradients. We term this server-side reconstruction process the *virtual path*, as it allows the server to follow the client's local steps without accessing raw data. As shown in Step 2 of Algorithm 2, the server uses the preserved random seed and receives project gradients of each local step from each client to recover the local model update path for each client. (3) *Server aggregates and initiates the next round:* After virtual path reconstruction, the server aggregates the reconstructed client model weights $\mathbf{w}_k^T$ to sparsely update the global model to $\mathbf{w}_r$. Subsequently, the server sends $\mathbf{w}_r$ and a new seed list $\{s_{r+1}^1, \ldots, s_{r+1}^T\}$ to clients and initializes next round.

**MEERKAT-VP: Virtual Path Client Selection and Early Stopping**. MEERKAT-VP extends MEERKAT by incorporating a VPCS strategy designed for heterogeneous environments. Leveraging the virtual path reconstruction capability, the server analyzes client update trajectories to identify those with extremely Non-IID data distributions. MEERKAT-VP then applies an early stopping mechanism to these identified clients, restricting them to a single local step to mitigate the negative impact of their skewed updates on global model convergence and performance.

## 2.2 THEORETICAL CONVERGENCE ANALYSIS

We theoretically analyze the convergence of MEERKAT and MEERKAT-VP under the Polyak–Łojasiewicz (PL)-type non-convex condition. All technical assumptions and the corresponding proof are presented in Appendix C.

**Theorem 2.1** (Convergence rate of MEERKAT). *Under Assumptions C.1–C.6, if the learning rate satisfies $\eta = \min\left\{\frac{1}{L(u+2)}, \frac{\mu\sqrt{c}\left(1+\sqrt{c_h}\right)}{2\,L^2(2+u)^2}\right\}$, then the global model $\{\mathbf{w}^r\}$ generated by the MEERKAT algorithm satisfies the following convergence bound:*

$$\frac{1}{R}\sum_{r=0}^{R-1}\left(f(w^r)-f^*\right) \leq \mathcal{O}\left(\frac{(2+u)^2}{TR}\cdot\mathbb{E}[f(w^0)-f(w^R)]\right) + \mathcal{O}\left(\frac{T}{2+u}\right) + O(1) \qquad (2)$$

**Theorem 2.2** (Convergence rate of MEERKAT-VP). *Under Assumptions C.1–C.6, if the learning rate satisfies $\eta = \min\left\{\frac{1}{L(u+2)}, \frac{\mu\sqrt{c}\left(K_g\,T+K_b\right)}{2\,K\,(2+u)^2\,L^2\,T\,\gamma}\right\}$ and each client $k \in K_b$ performs $T = 1$ local step while the remaining $K_g$ clients perform $T$ local steps, then the global model $\{\mathbf{w}^r\}$ generated by the MEERKAT-VP algorithm satisfies the following convergence bound:*

$$\frac{1}{R}\sum_{r=0}^{R-1}\mathbb{E}_{\bar{z}}\left[f(w^r)-f^*\right] \leq O\left(\frac{(K_g+K_b)^2\,(2+u)^2\,\gamma\,T}{c\,(K_gT+K_b)^2\,R}\right) + O\left(\frac{1+u}{K_g+K_b}\sum_{k=1}^{K_g+K_b}\Delta_k\right)$$
$$+ O\left(\frac{c\,T\,K_g}{(K_g+K_b)(1+u)\,\gamma}\right) + O\left(\frac{c\,K_b\,\sigma_h^2}{(K_g+K_b)(1+u)\,T\,\gamma}\right) + O(1).$$
$$(3)$$

The detailed theoretical analysis and proofs for Theorem 2.1 (MEERKAT) can be found in Appendix C.4, and for Theorem 2.2 (MEERKAT-VP) in Appendix C.5.

**Insights of MEERKAT.** MEERKAT's convergence reveals the intricate interplay of local steps $T$ and density $u$ on performance. (1) *MEERKAT's sparsity can theoretically improve performance.* Lower $u$ (higher sparsity) quadratically benefits the rate-dependent term $(\propto (2+u)^2)$, favoring faster initial convergence. However, it inflates the steady-state error $\left(\propto \frac{1}{2+u}\right)$. Comparing to the full-parameter case $(u = 1)$, sparsity $(u < 1)$ can reduce the overall bound by decreasing the rate-dependent term, offering communication and computational benefits. Yet, excessive sparsity can increase the steady-state error, suggesting an optimal density level $u \in (0, 1]$. (2) *High frequency communication with sparsity can lower the error floor.* Increasing $T$ improves the transient term scaling with $\mathcal{O}\left(\frac{(2+u)^2}{TR}\right)$, potentially accelerating convergence towards the steady state; however, it expands the steady-state term $\mathcal{O}\left(\frac{T}{2+u}\right)$, thereby increasing the error floor. Conversely, decreasing $T$ reduces the steady-state term, leading to a tighter final accuracy. Although smaller $T$ can lead to larger rate-dependent term. It's impact diminishes as the number of rounds $R$ increases. This analysis suggests that operating with frequent communication can theoretically reduce the steady-state error.

**Advantages of MEERKAT-VP.** We compare each component of the error bound under the same $T$ and $R$. First, the *transient term ratio* between MEERKAT-VP and MEERKAT is approximately $\gamma(1+\sqrt{c_h})^2 < 1$, and as $c_h \to 1$ so $\gamma \to 0$, the product $\gamma(1+\sqrt{c_h})^2 \to 0$, causing the transient error to vanish. Second, the *noise term ratio* is given by $\frac{\sigma_h^2/2}{\sigma_h^2/(\mu(1+\sqrt{c_h})^2)} = \frac{\mu(1+\sqrt{c_h})^2}{2}$, which remains below 1 whenever $\mu(1 + \sqrt{c_h})^2 < 2$. Since $\mu < 1$ empirically, this condition typically holds. Moreover, MEERKAT-VP introduces an additional variance term $\frac{cK_b\sigma_h^2}{2K(2+u)LT\gamma}$ that decays as $\mathcal{O}(1/T)$, making it negligible for large local steps. Lastly, in terms of heterogeneity, the coefficient of the

heterogeneity term $\sum_k \Delta_k$ in MEERKAT-VP is smaller: $\frac{(2+u)L}{4K} < \frac{L}{K}$, and the extra variance term scales inversely with $K$, thus diminishing in larger systems. Therefore, $E_{\text{MEERKAT-VP}} < E_{\text{MEERKAT}}$ and this gap widens as data heterogeneity $c_h$ increases. The detailed mathematical derivations and analysis, please refer to the Appendix C.5.

## 2.3 CLAIM 1: MEERKAT CAN OUTPERFORM FULL-PARAMETER FEDERATED ZO UNDER SAME SYNCHRONIZATION FREQUENCY

We claim that with fixed and extreme sparsity, MEERKAT outperforms full-parameter ZO in federated LLM fine-tuning under the same synchronization frequency and effectively mitigates the Non-IID client data problem through frequent synchronization and sparsity.

**Advantages of Sparsity in Federated ZO.** ZO has an intrinsic need for sparsity due to its reliance on nearly uniform perturbations across dimensions. Research on ZO shows that selecting sensitive parameters using gradient-based methods consistently outperforms alternative strategies such as weight magnitude or random parameter selection (Guo et al., 2024). Following this idea, MEERKAT produces LLM-sensitive parameters with gradient-based sparsification on pre-training data such as C4 (Raffel et al., 2020). Moreover, MEERKAT fine-tunes LLMs by estimating gradients through forward passes, completely bypassing backpropagation. This approach minimizes the need to cache gradients and activations, leading to significant memory savings. Focusing on sensitive parameters, MEERKAT ensures efficient and effective fine-tuning even under extreme sparsity levels (e.g., updating only $0.1\%$ of the parameters). Furthermore, these sensitive parameters exhibit transferability across downstream tasks. Theoretical analysis (Appendix C.4) also confirms that lower density $u$ leads to faster convergence via improved rate-dependent terms $\mathcal{O}((2+u)^2/(TR))$, while excessive sparsity increases the steady-state error $\mathcal{O}(T/(2+u))$, suggesting an optimal sparsity trade-off.

**Performance Under High Synchronization Frequency.** The lightweight communication of MEERKAT enables frequent client-server synchronization at a low cost, which is crucial for addressing data heterogeneity (Yang et al., 2024; Mendieta et al., 2022) in FL. In high-frequency communication scenarios, both the clients and the server only exchange a list of scalars (projected gradients) whereas in lower-frequency synchronization, clients have to upload projected gradients but still download sparse model parameters. By eliminating the need to download sparse model parameters in high-frequency synchronization, this approach is significantly more bandwidth-efficient, further minimizing communication overhead. We present the high-frequency synchronization algorithm of MEERKAT in Appendix C Algorithm 3. By facilitating frequent synchronization, training can better prevent clients from drifting. Our previous theoretical analysis also demonstrates that a smaller $T$ might influence the rate-dependent term, its beneficial impact on reducing the steady-state error is significant for achieving a tighter final accuracy over many rounds $R$.

## 2.4 CLAIM 2: EMPIRICAL GRADIP PHENOMENON REVEALS DATA HETEROGENEITY

MEERKAT's traceable virtual path allows us to analyze client local training dynamics, revealing an empirical phenomenon related to data heterogeneity via a metric we call GradIP.

**Definition 2.3.** Gradient Inner Product (GradIP) score: Let $\hat{\nabla} f_k^t$ (see Algorithm 2) denote the ZO gradient of LLM with equation 1 on client $k$ at local step $t$. Let $\nabla f_p$ denote the gradient of LLM computed by backpropagation on pre-training data. We define the GradIP score as $\langle \nabla f_p, \hat{\nabla} f_k^t \rangle$.

**GradIP As Indicator for Data Heterogeneity.** Leveraging the virtual path reconstruction capability of MEERKAT, the server can trace each client's local training trajectory. This process uses the uploaded projected gradients $g_k^t$ along with the shared random seeds (which regenerate $\mathbf{z}_k^t$) and the sparse mask $\mathbf{m}$ to reconstruct the local gradient $\hat{\nabla} f_k^t$. To understand the impact of a client's local data distribution on its training process, we introduce the *GradIP* metric. Inspired by the use of pre-training data gradients to identify sensitive parameters, GradIP quantifies the inner product between the local gradient computed during client training and the LLM pre-training gradient.

**Empirical GradIP Phenomenon.** Through the traceable virtual path provided by MEERKAT, we empirically investigated the behavior of the GradIP score among clients with different data distributions (IID and Non-IID) over their local training steps. Our analysis, presented in Appendix C.6, demonstrates distinct patterns in the dynamics of gradient norms based on data heterogeneity. While

IID client gradient norms exhibit fluctuations, those of extremely Non-IID clients decay and converge towards zero. The GradIP definition depends on the fixed pre-training gradient norm, local client gradient norm, and the angle $\theta$ between them. We hypothesize that $\theta$ between these two gradient vectors is nearly orthogonal. This leads us to expect a different manifestation of the GradIP Phenomenon when comparing IID and extremely Non-IID clients, primarily influenced by their differing local gradient norm trajectories.

## 2.5 Claim 3: Virtual Path Client Selection via GradIP Analysis

Building upon the traceable virtual path capability introduced in MEERKAT, we claim that VPCS, by leveraging GradIP analysis, effectively identifies and manages clients with extremely Non-IID data distribution, thereby improving global model performance and convergence. As established in Section 2.4, the GradIP score, computable by the server through virtual path reconstruction, provides an effective signal to identify such clients. VPCS utilizes this GradIP signal to detect extremely Non-IID clients. By analyzing the GradIP score trajectory and its behavior over local steps during a calibration phase, using metrics defined in Appendix table 3, the server empirically identifies clients exhibiting the characteristic diminishing GradIP behavior associated with extremely Non-IID data distribution. Upon identification via GradIP analysis, VPCS applies early stopping: these clients perform only one local training step per communication round. To ensure full data utilization over training, a data pointer tracks the batch processed, allowing clients to resume from that point in subsequent rounds. This strategy mitigates client drift from skewed data while ensuring their entire dataset is eventually processed. Algorithm 1 outlines the detailed procedure, and Figure 5 illustrates the workflow. Our previous theoretical analysis of MEERKAT-VP suggests that early stopping on extremely Non-IID clients can lead to improved global model performance.

---

**Algorithm 1** MEERKAT-VP

1: **Input:** calibration step $T_{\mathsf{cali}}$, pre-training gradients $\nabla f_{\mathsf{C4}}$, projected gradients $\{g_k^1, \ldots, g_k^{T_{\mathsf{cali}}}\}$, seed $s_r^t$, sparse mask $\mathbf{m}$, initial phase steps $T_{\mathsf{init}}$, later phase steps $T_{\mathsf{later}}$, convergence threshold $\sigma$, Initial to later ratio $\rho_{\mathsf{later}}$, quiescent step ratio $\rho_{\mathsf{quie}}$

2: **Step 1: Virtual Path Reconstruction & GradIP Calculation**

3: Generate $\mathbf{z}_k^t$ using $s_r^t$.

4: Compute $\hat{\nabla} f_k^t = g_k^t \cdot (\mathbf{z}_k^t \odot \mathbf{m})$

5: Compute $\mathsf{Gradip} = \hat{\nabla} f_k^t \cdot \nabla f_{\mathsf{C4}}$ (Definition 2.3).

6: **Step 2: Identify Extremely Non-IID Clients**

7: Compute the average value of $\mathsf{Gradip}$ over the initial-phase steps.

$$\mathsf{Gradip}_{\mathsf{init\_avg}} = \frac{1}{T_{\mathsf{init}}} \sum_{t=1}^{T_{\mathsf{init}}} \mathsf{Gradip}_t$$

8: Compute the average value of $\mathsf{Gradip}$ over the later-phase steps.

$$\mathsf{Gradip}_{\mathsf{later\_avg}} = \frac{1}{T_{\mathsf{later}}} \sum_{t=1}^{T_{\mathsf{later}}} \mathsf{Gradip}_t$$

9: Compute the client's Initial to later ratio $\rho_{\mathsf{later\_client}}$ and quiescent step ratio $\rho_{\mathsf{quie\_client}}$

$$\rho_{\mathsf{quie\_client}} = \frac{\{s \in \{1, 2, \ldots, T_{\mathsf{later}}\} \mid \mathsf{Gradip}_s < \sigma\}}{T_{\mathsf{later}}}$$

$$\rho_{\mathsf{later\_client}} = \frac{\mathsf{Gradip}_{\mathsf{init\_avg}}}{\mathsf{Gradip}_{\mathsf{later\_avg}}}$$

10: Record client IDs whose $\rho_{\mathsf{later\_client}}$ or $\rho_{\mathsf{quie\_client}}$ exceed $\rho_{\mathsf{later}}$ or $\rho_{\mathsf{quie}}$.

11: **Step 3: Early Stopping**

12: Require these identified clients to only perform one local training step.

---

## 3 Experiment

In this section, we aim to validate the effectiveness of MEERKAT and MEERKAT-VP. We aim to address the following research questions in response to claims in Section 2: (1) **RQ 1 for Claim 1 (2.3):** Is MEERKAT more effective than full parameter federated ZO under the same synchronization frequency, especially in heterogeneous environments? (2) **RQ 2 for Claim 2 (2.4):** Can the empirical GradIP phenomenon, observed via the virtual path, effectively reveal data heterogeneity by showing distinct behaviors for IID and Non-IID data distribution clients? (3) **RQ 3 for Claim 3 (2.5):** Can MEERKAT-VP, leveraging GradIP analysis, mitigate the impact of extreme Non-IID data compared to MEERKAT?

We focus on models Gemma-2-2b (Team, 2024), Qwen2-1.5B (Qwen Team, 2024), Llama-3.2-1B (Dubey et al., 2024). We conduct experiments on SST2 (Socher et al., 2013), AG's News (Zhang et al., 2015), Yelp polarity (yelp) (Zhang et al., 2015), RTE (Wang, 2018), BoolQ (Clark et al., 2019), WSC (Levesque et al., 2012), WiC (Pilehvar & Camacho-Collados, 2018) datasets. The datasets are

partitioned across clients following a Dirichlet distribution to simulate clients with Non-IID data. For more experimental settings, we refer the readers to Appendix D.1.

Table 1: Performance comparison of MEERKAT and Full-FedZO on multiple non-IID data distribution settings. "Acc" is the average test accuracy across tasks. Bold numbers indicate the highest value in each row.

| | Methods | Local Step | SST-2 | AgNews | Yelp | BoolQ | RTE | WSC | WIC | Acc |
|---|---|---|---|---|---|---|---|---|---|---|
| **LLaMA-3.2-1B** | Full-FedZO | 10 | 0.909 | 0.705 | 0.940 | 0.641 | 0.542 | 0.634 | 0.523 | 0.699 |
| | Weight Magnitude | 10 | 0.902 | 0.857 | 0.951 | 0.696 | 0.551 | 0.519 | 0.546 | 0.717 |
| | Lora-FedZO | 10 | 0.901 | 0.749 | 0.96 | 0.649 | 0.524 | 0.634 | 0.59 | 0.715 |
| | MEERKAT | 10 | **0.916** | **0.872** | **0.964** | 0.695 | **0.600** | **0.653** | **0.614** | **0.759** |
| | Full-FedZO | 30 | 0.904 | 0.706 | 0.935 | 0.636 | 0.533 | 0.634 | 0.539 | 0.698 |
| | Weight Magnitude | 30 | 0.902 | 0.84 | 0.946 | 0.674 | 0.542 | 0.556 | 0.550 | 0.716 |
| | Lora-FedZO | 30 | 0.904 | 0.556 | 0.964 | 0.652 | 0.533 | 0.634 | 0.545 | 0.684 |
| | MEERKAT | 30 | 0.897 | **0.862** | **0.965** | 0.646 | **0.577** | **0.644** | **0.583** | **0.739** |
| | Full-FedZO | 50 | 0.889 | 0.696 | 0.935 | 0.633 | 0.542 | 0.634 | 0.529 | 0.694 |
| | Weight Magnitude | 50 | 0.897 | 0.838 | 0.948 | 0.662 | 0.551 | 0.562 | 0.554 | 0.716 |
| | Lora-FedZO | 50 | 0.876 | 0.447 | 0.967 | 0.639 | 0.541 | 0.634 | 0.562 | 0.667 |
| | MEERKAT | 50 | **0.909** | 0.827 | **0.965** | 0.647 | **0.595** | 0.634 | **0.567** | **0.734** |
| | Full-FedZO | 100 | 0.901 | 0.705 | 0.939 | 0.632 | 0.533 | 0.634 | 0.525 | 0.695 |
| | Weight Magnitude | 100 | 0.885 | 0.83 | 0.946 | 0.66 | 0.56 | 0.534 | 0.548 | 0.709 |
| | Lora-FedZO | 100 | 0.868 | 0.247 | 0.953 | 0.642 | 0.521 | 0.634 | 0.529 | 0.628 |
| | MEERKAT | 100 | 0.896 | 0.777 | **0.961** | 0.658 | **0.577** | **0.644** | **0.573** | **0.726** |
| **Qwen2-1.5b** | Full-FedZO | 10 | 0.888 | 0.700 | 0.928 | 0.694 | 0.808 | 0.673 | 0.639 | 0.761 |
| | Weight Magnitude | 10 | 0.881 | 0.84 | 0.939 | 0.681 | 0.795 | 0.672 | 0.623 | 0.776 |
| | Lora-FedZO | 10 | 0.939 | 0.847 | 0.944 | 0.667 | 0.795 | 0.663 | 0.521 | 0.768 |
| | MEERKAT | 10 | **0.949** | **0.881** | 0.934 | **0.752** | **0.813** | **0.682** | 0.628 | **0.805** |
| | Full-FedZO | 30 | 0.892 | 0.699 | 0.926 | 0.708 | 0.791 | 0.663 | 0.594 | 0.753 |
| | Weight Magnitude | 30 | 0.88 | 0.843 | 0.939 | 0.681 | 0.786 | 0.673 | 0.594 | 0.771 |
| | Lora-FedZO | 30 | 0.923 | 0.843 | 0.948 | 0.666 | 0.777 | 0.673 | 0.519 | 0.764 |
| | MEERKAT | 30 | **0.944** | **0.878** | 0.928 | **0.734** | **0.800** | 0.663 | **0.624** | **0.795** |
| | Full-FedZO | 50 | 0.868 | 0.696 | 0.922 | 0.707 | 0.773 | 0.663 | 0.594 | 0.746 |
| | Weight Magnitude | 50 | 0.883 | 0.855 | 0.938 | 0.703 | 0.768 | 0.673 | 0.595 | 0.774 |
| | Lora-FedZO | 50 | 0.934 | 0.834 | 0.941 | 0.679 | 0.76 | 0.653 | 0.510 | 0.759 |
| | MEERKAT | 50 | **0.948** | **0.872** | 0.926 | **0.746** | **0.795** | 0.663 | 0.594 | **0.792** |
| | Full-FedZO | 100 | 0.864 | 0.691 | 0.917 | 0.675 | 0.777 | 0.653 | **0.620** | 0.742 |
| | Weight Magnitude | 100 | 0.888 | 0.842 | 0.934 | 0.695 | 0.768 | 0.656 | 0.579 | 0.766 |
| | Lora-FedZO | 100 | 0.934 | 0.785 | 0.937 | 0.664 | 0.786 | 0.653 | 0.512 | 0.753 |
| | MEERKAT | 100 | **0.936** | **0.878** | 0.925 | **0.741** | **0.795** | 0.663 | 0.610 | **0.792** |
| **Gemma2-2b** | Full-FedZO | 10 | 0.928 | 0.721 | 0.943 | 0.731 | 0.564 | 0.644 | 0.595 | 0.732 |
| | Weight Magnitude | 10 | 0.931 | 0.849 | 0.955 | 0.778 | 0.711 | 0.634 | 0.595 | 0.779 |
| | Lora-FedZO | 10 | 0.936 | 0.853 | 0.966 | 0.763 | 0.568 | 0.663 | 0.605 | 0.765 |
| | MEERKAT | 10 | **0.939** | **0.869** | 0.96 | **0.804** | 0.591 | 0.634 | **0.609** | 0.772 |
| | Full-FedZO | 30 | 0.927 | 0.802 | 0.932 | 0.725 | 0.568 | 0.634 | 0.581 | 0.738 |
| | Weight Magnitude | 30 | 0.935 | 0.851 | 0.951 | 0.771 | 0.653 | 0.634 | 0.598 | 0.770 |
| | Lora-FedZO | 30 | 0.932 | 0.804 | 0.966 | 0.671 | 0.551 | 0.634 | 0.589 | 0.735 |
| | MEERKAT | 30 | **0.94** | **0.855** | 0.947 | 0.734 | 0.568 | **0.644** | **0.601** | 0.756 |
| | Full-FedZO | 50 | 0.932 | 0.791 | 0.943 | 0.712 | 0.582 | **0.634** | 0.567 | 0.737 |
| | Weight Magnitude | 50 | 0.936 | 0.851 | 0.941 | 0.745 | 0.591 | 0.628 | 0.597 | 0.756 |
| | Lora-FedZO | 50 | 0.91 | 0.779 | 0.942 | 0.664 | 0.557 | **0.634** | 0.597 | 0.726 |
| | MEERKAT | 50 | **0.945** | **0.857** | **0.966** | **0.767** | **0.613** | **0.634** | **0.623** | **0.772** |
| | Full-FedZO | 100 | 0.925 | 0.818 | 0.933 | 0.672 | 0.533 | 0.615 | 0.567 | 0.723 |
| | Weight Magnitude | 100 | 0.922 | 0.839 | 0.942 | 0.723 | 0.568 | 0.644 | 0.592 | 0.747 |
| | Lora-FedZO | 100 | 0.922 | 0.247 | 0.942 | 0.62 | 0.541 | 0.634 | 0.573 | 0.640 |
| | MEERKAT | 100 | **0.94** | **0.851** | **0.951** | **0.745** | 0.551 | 0.634 | 0.574 | **0.749** |

## 3.1 ANSWER TO RQ1: SUPERIORITY OF MEERKAT COMPARED TO FULL-FEDZO IN FL

This section experimentally validates Claim 1 (Section 2.3), demonstrating MEERKAT's superiority over full-parameter Federated ZO under the same synchronization frequency and its effectiveness in mitigating Non-IID challenges via high-frequency synchronization.

First, to assess sparsity's benefits, we compare MEERKAT to Full-FedZO and other sparse methods (Weight Magnitude, LoRA-FedZO, Random-Select) with equivalent synchronization frequencies (local steps $T \in \{10, 30, 50, 100\}$). With a fixed $0.1\%$ mask, MEERKAT reduces communication budget by over $1000\times$ compared to Full-FedZO and achieves a strong computational and communication efficiency (Table 24). Using C4 as a calibration dataset, our analysis shows that the sensitivity of the gradient is highly concentrated: the top $0.1\%$ of the parameters have $52\times$ larger average square gradients than the next $0.1$–$1\%$ bucket (Table 9), which motivates extreme sparsity. The mask is transferred across domain-shifted calibration datasets, and a client-aggregated UnionMask

performs comparably (Table 11). Across IID and Non-IID data distributions, MEERKAT outperforms Full-FedZO and other sparsity methods on many tasks (Tables 1, 10, 13). Under the same settings, MEERKAT also outperforms DeComFL (Li et al., 2024) (Table 19).

Next, we evaluate performance under an extreme communication regime with a single local step ($T=1$). We compare MEERKAT with Full-FedZO and LoRA-FedZO in the IID and Non-IID data distributions (Dirichlet $\alpha \in \{0.5, 0.3, 0.1\}$). Figure 2 presents the results for $\alpha = 0.5$, the results for $\alpha = 0.3$ and $0.1$ are available in Appendix D.2 figure 6. Specifically, Figure 2 reveals a remarkable finding: on the Qwen2-1.5b model, MEERKAT's average test accuracy over seven tasks under Non-IID data distribution matches that under IID data distribution. Beyond this exact match, results show that at a local step of $T = 1$, MEERKAT effectively bridges the performance gap between IID and Non-IID data distribution settings, achieving nearly comparable test accuracy across both data distributions, and consistently outperforms baselines. Varying sparsity under $T = 1$ (Table 15) confirms strong accuracy even at $10^{-3}$–$10^{-4}$, substantially reducing client memory demands and making it ideal for resource-constrained FL. These results support **Claim 1**: high-frequency communication combined with extreme sparsity mitigates Non-IID drift. We also explored sensitive parameter selection using downstream task data. Since performance remained comparable under identical communication frequencies and sparsity levels, we prioritized pre-training data to better preserve client privacy (Appendix D.2, Tables 21, 20, 22).

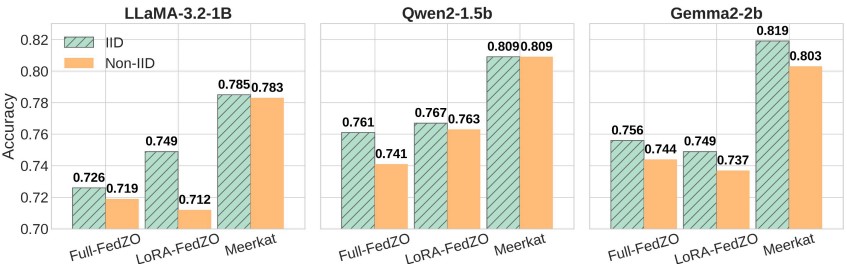

Figure 2: This figure compares three methods—Full-FedZO, LoRA-FedZO, and MEERKAT—on three LLMs: LLaMA-3.2-1B, Qwen2-1.5b, and Gemma2-2b. The x-axis shows the different methods, and each method has two bars indicating performance under IID and Non-IID settings. The Non-IID results are obtained under a Dirichlet distribution with $\alpha = 0.5$ .The y-axis represents the average test accuracy across multiple downstream tasks—SST2, AgNews, Yelp, BoolQ, RTE, WSC, and WiC. All detailed results for these tasks are provided in Appendix D.2, Table 15.

### 3.2 ANSWER TO RQ2: GRADIP TRAJECTORIES AS EFFECTIVE INDICATORS OF DATA HETEROGENEITY

This section experimentally validates Claim 2 (Section 2.4), investigating GradIP trajectories as indicators of data heterogeneity. Based on our theoretical analysis assuming single-label Non-IID data (Section C.6), we study the dynamics of gradient-related metrics during local training. We first compare two extremes: IID clients vs. clients with single-label (extreme Non-IID) data. We track three metrics: GradIP score, local gradient norm, and cosine value between the local and pre-training gradients. As shown in Figures 3 and 7, GradIP for extreme Non-IID clients steadily decays to zero over 100 steps, while for IID clients it fluctuates persistently. To understand this, we analyze its components: Figure 8(a) shows cosine value stays near zero (i.e., gradients are nearly orthogonal) for both settings, suggesting the gradient norm is the key factor. Indeed, Figure 8(b) shows that the gradient norm mirrors GradIP's behavior across the two settings. More-

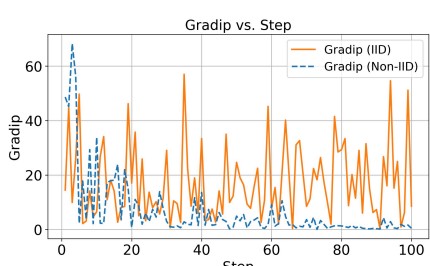

Figure 3: Under a density ratio of $5 \times 10^{-3}$, we track the GradIP (see Definition 2.3) over 100 local training steps on the SST-2 dataset using LLaMA-3.2-1B model, comparing a client with IID data to a client with Non-IID data.

over, in later stages, GradIP declines more sharply for Non-IID clients than for IID ones, making this stage-wise mean difference an additional criterion for identifying Non-IID clients. We further extend

our analysis to more general Non-IID scenarios (Figure 9, Figure 10, Figure 11), where GradIP exhibits similar dynamics that correlate with the degree of heterogeneity.

### 3.3 ANSWER TO RQ3: VPCS EARLY STOPPING FOR CLIENTS WITH EXTREMELY NON-IID DATA DISTRIBUTION

This section experimentally validates Claim 3 (Section 2.5). As established in Section 3.2, GradIP trajectories provide an effective signal for identifying clients with extremely Non-IID data, exhibiting distinct behaviors. Leveraging this signal, VPCS detects extremely Non-IID clients during a calibration phase and applies early stopping, limiting them to one local training step per communication round (Algorithm 1). To validate the effectiveness of this VPCS strategy in improving performance, we compared MEERKAT-VP with MEERKAT and Random Client Selection, which randomly selects the same number of clients for early stopping as VPCS, under Non-IID data distributions Dirichlet $\alpha = 0.5$ and the same communication frequencies. Crucially, for the same model, dataset, and communication frequency, the three methods employed the same sparsity level. Figure 4 illustrates the average test accuracy across multiple downstream tasks for MEERKAT-VP compared to MEERKAT and RANDOM CLIENT SELECTION. Detailed results for individual tasks are presented in Appendix D.2 Table 14. As shown in Figure 4, MEERKAT-VP consistently outperforms both MEERKAT and RANDOM CLIENT SELECTION in different communication frequencies. Furthermore, Table 25 shows that MEERKAT-VP achieves performance competitive with a back-propagation upper bound and significantly outperforms an adapted FedDYN (Acar et al., 2021) baseline. These experimental results strongly validate Claim 3, confirming that VPCS effectively leverages GradIP analysis to manage extremely Non-IID data distribution clients, leading to improved performance for ZO federated LLM fine-tuning.

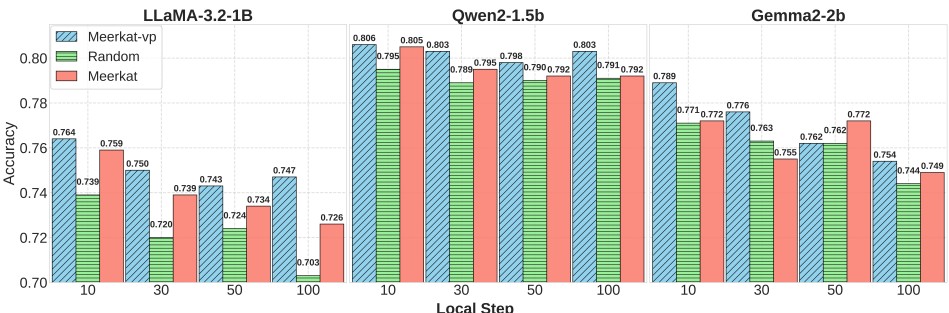

Figure 4: This figure compares three methods—MEERKAT-VP, MEERKAT and Random Client Selection—across three LLMs: LLaMA-3.2-1B, Qwen2-1.5b, and Gemma2-2b. The x-axis shows the local step values (10, 30, 50, 100), while the y-axis indicates the average test accuracy over multiple downstream tasks—SST-2, AgNews, Yelp, BoolQ, RTE, WSC, and WiC—in a Non-IID data distribution setting. All detailed results for these tasks are presented in Appendix D.2 Table 14.

## 4 RELATED WORK

Our research leverages advances in ZO federated optimization, sparsity techniques for LLMs, and communication frequency adjustments strategies for addressing data heterogeneity. ZO methods significantly reduce computational and communication overhead. Integrating sparsity into LLM fine-tuning amplifies these benefits, substantially decreasing resource demands during training and inference. Concurrently, communication frequency adjustments mitigate performance degradation induced by Non-IID data, emphasizing a crucial trade-off between communication budget and global model performance. A detailed discussion is provided in Appendix B.

## 5 CONCLUSION

In this paper, we introduce MEERKAT, a sparse zeroth-order federated fine-tuning methodology. Experiments show MEERKAT outperforms Full-FedZO and other sparsity methods on most tasks at

equivalent communication frequencies. MEERKAT's efficiency enables high-frequency communication, effectively mitigating Non-IID drift. Moreover, we propose MEERKAT-VP. This methodology utilizes VPCS, which analyzes GradIP via virtual paths to enable the selective early stopping of extreme Non-IID clients. This approach is shown to improve model performance. Our work thus offers effective methods for efficient ZO federated LLM fine-tuning under varying network conditions and data heterogeneity. Given the technical focus of this work on algorithmic contributions, there are no direct negative societal consequences inherent to it that need to be emphasized; potential negative impacts would arise from the specific applications where these methods are deployed.

## ACKNOWLEDGMENTS

We gratefully acknowledge the support of Lambda, Inc. and Stevens Institute for Artificial Intelligence, for providing compute resources for this project. This research also used resources of the Argonne Leadership Computing Facility, which is a U.S. Department of Energy Office of Science User Facility operated under contract DE-AC02-06CH11357. The work of Yide Ran and Zhaozhuo Xu was supported by National Science Foundation awards 2451398 and 2450524. The work of Hao Wang was supported by National Science Foundation awards 2523997, 2534286, and 2315612. The work of Xiaodong Yu was supported by National Science Foundation award 2348465. This work was also supported in part by National Science Foundation awards 2112562, 2328805, and ARO W911NF-23-2-0224.

## ETHICS STATEMENT

This work follows the ICLR Code of Ethics. This paper aims to advance zeroth-order optimization for federated LLM fine-tuning by addressing key challenges related to efficiency and data heterogeneity. All datasets used are publicly available for academic evaluation. Our method is designed to protect user privacy by operating within the Federated Learning framework, where raw data remains on local devices. Respecting the broader research community, we acknowledge prior work appropriately and ensure our contributions are situated within ongoing academic efforts. We declare no conflicts of interest or external sponsorships associated with this work.

## REPRODUCIBILITY STATEMENT

We are committed to ensuring the reproducibility of our work. The datasets and models used in this study are detailed in the experiments section (Section 3). The workflow for MEERKAT is presented in Figure 1, with further details in Section 2 and Algorithm 2. The workflow for MEERKAT-VP is demonstrated in Figure 5 and Algorithm 1. All experimental parameters are listed in Appendix D.1 Tables 4-8. The complete theoretical analysis for our methods can be found in Appendix C.

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

## APPENDIX

In Section A, we discuss the usage of large language models in this work. In Section B, we present the related work relevant to this study. In Section C, we present the theoretical convergence analysis of MEERKAT, including its high-frequency communication method. Additionally, we analyze the convergence of MEERKAT-VP and demonstrate its superior performance compared to MEERKAT. We further prove that under extreme Non-IID settings, the gradient norm gradually vanishes during convergence, whereas in IID settings, it tends to oscillate. In Section D, we provide details on experimental hyperparameters and report supplementary results.

# A    LLM Usage

We used an LLM-based writing assistant solely for grammar and typographical corrections to improve the clarity of this paper. All outputs were carefully reviewed and revised by the authors to ensure technical accuracy and consistency with the intended scientific meaning. The intellectual contributions, methodological advances, and scientific insights are entirely original and author-driven.

# B    Related Work

**Federated Zeroth-Order Optimization.** Zeroth-order optimization (Malladi et al., 2023; Zhang et al., 2024a) has gained increasing attention in federated learning (Fang et al., 2022; Zhang et al., 2021), particularly for addressing challenges in training costs, privacy, and communication overhead. Fine-Grained  (Chen et al., 2024) demonstrates how clients can reduce upload overhead by sending estimated gradients rather than full model parameters to the server, though download costs remain significant due to complete model weight transfers. DeComFL (Li et al., 2024) further advances this approach by using gradient scalars for both uploads and downloads, substantially reducing bidirectional communication costs. However, it does not address the challenges posed by data heterogeneity (Non-IID) in federated learning. The integration of AirComp wireless technology enables direct over-the-air aggregation of model updates (Fang et al., 2022). In black-box settings where pre-trained language model parameters are inaccessible, FedBPT (Sun et al., 2023) employs ZO to optimize prompt vectors, achieving efficient distributed optimization with reduced computational and communication overhead. FedMeZO (Ling et al., 2024) analyzes the convergence properties of ZO for federated LLM fine-tuning.

**Sparsity in LLM.** Current research on sparsity in LLMs explores techniques such as pruning, contextual sparsity prediction, and structured sparsity (Wang et al., 2019; Liu et al., 2023b;a; Huang et al., 2024a; Lu et al., 2024; Shao et al., 2024; Wu et al., 2025; Xu et al., 2024; Su et al., 2024; Zhang et al., 2024b; Zheng et al., 2024; Zhou et al., 2024). These methods enhance both training and inference by improving computational efficiency, reducing memory usage, and enabling deployment in resource-constrained environments. Sparsity has also proven particularly effective in zeroth-order (ZO) optimization (Guo et al., 2024; Liu et al., 2024), especially when combined with weight quantization for fine-tuning LLMs. Building on this, our work investigates the role of sparsity in resource-frugal federated fine-tuning of LLMs.

**High-Frequency Communication for Non-IID Federated Learning.** Data heterogeneity across clients is a major challenge in Federated Learning, significantly degrading performance compared to IID settings. Increasing communication frequency, by reducing local training steps per round, is explored as a strategy to mitigate this issue. Early work showed that merely reducing local steps had limited improvements in extreme non-IID scenarios (Zhao et al., 2018). Theoretical analysis later confirmed that smaller local training steps can improve convergence speed under Non-IID conditions, but at the cost of increased communication budget, highlighting a critical trade-off (Li et al., 2020b). To effectively handle challenges arising from non-IID data that often necessitate higher communication, various algorithms have been proposed: SCAFFOLD (Karimireddy et al., 2021) highlights the 'client-drift' problem in FedAvg, noting it's exacerbated by increased local training steps (reduced communication frequency), and proposes using control variates to mitigate this drift, enabling improved convergence; FedDyn (Acar et al., 2021) guarantees consistent convergence to the global optimum even with a larger number of local training steps (lower communication frequency). This overcomes the limitation of traditional methods where high communication frequency is needed to compensate for local-global optimum inconsistency. Empirical studies further demonstrate that performance is highly sensitive to the number of local training steps under different non-IID distributions, and the optimal communication frequency depends on the specific data heterogeneity (Li et al., 2021). FedSA-LoRA (Guo et al., 2025) tackles Non-IID heterogeneity in federated LoRA by showing that the two low-rank matrices play asymmetric roles—A learns global shared knowledge while B captures client-specific patterns—and then aggregating only the A matrices on the server while keeping the B matrices local, which reduces aggregation distortion and cross-client knowledge contamination under skewed client distributions. FedAvgM (Hsu et al., 2019) directly studies FedAvg under synthetic label-skewed Dirichlet partitions and proposes adding server-side momentum to mitigate Non-IID client drift: by accumulating past aggregated updates, the server smooths noisy, biased client gradients and recovers much higher test accuracy and more stable training even when client

label distributions are highly skewed and only a small fraction of clients participate in each round. Stochastic controlled averaging with compression (Huang et al., 2024b) addresses Non-IID data in compressed FL by building SCALLION and SCAFCOM on a simplified SCAFFOLD backbone: control variates and local momentum keep local updates aligned with the global direction, and the authors prove (and verify empirically) that these algorithms remain robust to arbitrary client heterogeneity and partial participation even under aggressive communication compression.These works underscore the complex interplay between data heterogeneity, local computation, and communication frequency. This complexity motivates the development of algorithmic solutions to improve efficiency and robustness in FL under Non-IID settings.

## C  THEORETICAL AND ALGORITHM ANALYSIS

### C.1  NOTATIONS AND DEFINITIONS

In this subsection, we formally define the assumptions, notations and concepts used in the convergence analysis of MEERKAT and MEERKAT-VP. Table 2 summarizes the key symbols.

Table 2: Notations used in our theoretical analysis.

| Notation | Meaning |
|---|---|
| $\mathbf{w}$ | global model parameter |
| $K$ | total number of clients in the federated system |
| $p_k$ | probability or weight assigned to client $k$ |
| $f_k(\mathbf{w})$ | total loss computed over all data samples of the client k. |
| $f(\mathbf{w})$ | global loss function evaluated by the global model over all data |
| $T$ | number of local update steps per communication round |
| $r$ | communication round |
| $t$ | local update time step |
| $\eta$ | local learning rate |
| $\epsilon$ | perturbation magnitude in ZO estimation |
| $\mathbf{z}_k^t$ | standard Gaussian vector for client $k$ at local step $t$ from $\mathcal{N}(0, I_d)$ |
| $\mathbf{m}$ | binary sparse mask vector ($\mathbf{m} \in \{0,1\}^d$) |
| $d$ | model dimension |
| $R$ | federated learning training round |
| $u$ | sparsity ratio |
| $c$ | gradient coverage |
| $g_k^t$ | projected gradient estimate for client $k$ at local step $t$ |
| $\hat{\nabla} f_k^t$ | zeroth-order gradient of client $k$ at local step $t$ |
| $L$ | Lipschitz smoothness (Assumption 1) |
| $\mu$ | PL inequality (Assumption 2) |
| $f^*$ | minimal global loss achieved by optimizing the global model |
| $f_k^*$ | minimal client loss achieved by optimizing the local model on client $k$ |
| $c_h$ and $\sigma_h^2$ | heterogeneity-induced variance (Assumption 4) |
| $\|\cdot\|_{\text{op}}$ | operator norm of a matrix |
| $\sigma^2$ | variance of the sparse ZO gradient estimator(Assumption 6) |
| $\gamma$ | The clients with balanced data distributions contribute to the global model during training. |

### C.2  ASSUMPTIONS

We introduce the assumptions used in the convergence analysis of MEERKAT and MEERKAT-VP.

**Assumption C.1** ( Lipschitz smoothness). We assume that each client $k$'s local objective function $f_k(\mathbf{w})$ is differentiable and has $L$-Lipschitz continuous gradients:

$$\|\nabla f_k(\mathbf{w}_1) - \nabla f_k(\mathbf{w}_2)\| \leq L\|\mathbf{w}_1 - \mathbf{w}_2\|, \quad \forall \mathbf{w}_1, \mathbf{w}_2 \in \mathbb{R}^d.$$

Consequently, the global loss $f(\mathbf{w}) = \sum_{k=1}^K p_k f_k(\mathbf{w})$ is also $L$-smooth.

**Assumption C.2** (PL inequality). We assume that $f(\mathbf{w})$ satisfies the Polyak-Łojasiewicz (PL) condition:

$$f(\mathbf{w}) - f^* \leq \frac{1}{2\mu}\|\nabla f(\mathbf{w})\|^2, \quad \forall \mathbf{w} \in \mathbb{R}^d,$$

$\mu > 0$ is the PL constant. This condition holds for a broad class of non-convex objectives and is commonly used in analyzing convergence of gradient-based and zeroth-order methods.

**Assumption C.3** (Global–Local Disparities in Non-i.i.d. Setting). For any $\theta \in \mathbb{R}^d$, the discrepancy between the local and global gradient is bounded by

$$\left\|\nabla f(\theta) - \nabla f_i(\theta)\right\|^2 \leq c_h \left\|\nabla f(\theta)\right\|^2 + \sigma_h^2,$$

where $c_h > 0$ and $\sigma_h^2 \geq 0$ are constants, and $\theta$ is the global model parameter broadcast to all clients at the start of each round. We further assume $c_h \in (0, 1)$. In particular,

- A smaller $c_h$ corresponds to lower *data heterogeneity*: local gradient deviations from the global gradient are small, indicating that client data distributions are nearly i.i.d.

- A larger $c_h$ signals stronger non-i.i.d data distribution effects, with greater variation between each client's gradient and the global gradient.

**Assumption C.4** (Bounded stochastic gradient variance). For any sample $(x, y) \sim \mathcal{D}$ and any $\mathbf{w} \in \mathbb{R}^d$, denote $f(\mathbf{w}; (x, y))$ as the loss on that single data point, and let $\bar{f}(\mathbf{w}) := \mathbb{E}_{(x,y)\sim\mathcal{D}}\left[f(\mathbf{w}; (x, y))\right]$ be the average full-batch loss. We assume

$$\left\|\nabla f(\mathbf{w}; (\mathbf{x}, y)) - \nabla \bar{f}(\mathbf{w})\right\|_2^2 \leq \sigma^2.$$

**Assumption C.5** (Local–Global Optimality Gap). For each client $k$, define the local–global optimality gap as

$$\Delta_k = \left\|\mathbf{w}_k^* - \mathbf{w}^*\right\|_2^2,$$

where $\mathbf{w}_k^*$ is the local optimal model on client $k$ and $\mathbf{w}^*$ is the global optimal model.

**Assumption C.6** (Sensitive parameters are sparse). At each local step $t$ (and for every client $k$), there exists a binary mask $\mathbf{m} \in \{0, 1\}^d$ with exactly $u$ non-zero entries and a constant $c \in [0, 1]$ such that

$$\left\|\mathbf{m} \odot \nabla f_k\left(\mathbf{w}_k^t; (\mathbf{x}_t, \mathbf{y}_t)\right)\right\|^2 = c \left\|\nabla f_k\left(\mathbf{w}_k^t; (\mathbf{x}_t, \mathbf{y}_t)\right)\right\|^2.$$

We further assume $c \gg \frac{u}{d}$, meaning this small subset of "sensitive" parameters captures a disproportionately large fraction of the gradient norm.

These assumptions are standard and foundational in optimization and FL literature (Bottou et al., 2018; Li et al., 2020a;b; Wang et al., 2021; Guo et al., 2024)

We start by formulating the expectation of the sensitive sparse ZO surrogate gradient norm square in terms of its corresponding stochastic gradient norm square.

**Lemma C.7** (Sensitive sparse ZO surrogate gradient norm square).

$$\mathbb{E}_{\bar{z}}\left\|\hat{\nabla}f\left(w_t, (x_t, y_t), \bar{z}_t\right)\right\|^2 = (2 + u)c \left\|\nabla f\left(w_t; (x_t, y_t)\right)\right\|^2.$$

*Proof.* Our masked perturbation $\bar{\mathbf{z}}$ is sampled as $\bar{\mathbf{z}} \sim \mathcal{N}\left(0, \tilde{I}_{d,\mathbf{m}}\right)$, where $\tilde{I}_{d,\mathbf{m}}$ equals the identity matrix $I_d$ with its main diagonal masked by $\mathbf{m}$.

We expand the sensitive sparse ZO surrogate–gradient covariance matrix:

$$
\begin{aligned}
&\mathbb{E}_{\bar{z}}\hat{\nabla}f(w, (x, y), \bar{z})\hat{\nabla}f(w, (x, y), \bar{z})^\top \\
&= \mathbb{E}_{\bar{z}}[\bar{z}\bar{z}^\top((\mathbf{m} \odot \nabla f(w; (x, y)))(\mathbf{m} \odot \nabla f(w; (x, y)))^\top)\bar{z}\bar{z}^\top] \\
&= 2((\mathbf{m} \odot \nabla f(w; (x, y)))(\mathbf{m} \odot \nabla f(w; (x, y)))^\top) + \|\mathbf{m} \odot \nabla f(w; (x, y))\|^2 \tilde{I}_{d,\mathbf{m}}
\end{aligned}
$$

The above expected squared norm is obtained by summing the diagonal elements of this covariance matrix:

$$\mathbb{E}_{\bar{z}}\left\|\hat{\nabla}f(w_t, x_t, \bar{z}_t)\right\|^2 = \left(\mathrm{diag}[\mathbb{E}_{\bar{z}}\hat{\nabla}f(w, (x,y), \bar{z})\hat{\nabla}f(w, (x,y), \bar{z})^\top]\right)^2$$

$$= 2c\left\|\nabla f(w_t; (x_t, y_t))\right\|^2 + uc\left\|\nabla f(w_t; (x_t, y_t))\right\|^2$$

$$= (2+u)c\left\|\nabla f(w_t; (x_t, y_t))\right\|^2.$$

$\square$

**Lemma C.8** (Unbiasedness of Masked Sparse ZO Surrogate Gradient).

$$\mathbb{E}_{\bar{z}}\left[\hat{\nabla}f_k(\mathbf{w}_k^t, \bar{z})\right] = \mathbf{m}\odot\nabla f_k(\mathbf{w}_k^t), \quad where \; \bar{z} = z\odot\mathbf{m}. \tag{4}$$

*Proof.* First, consider the estimator defined as:

$$\hat{\nabla}f_k(\mathbf{w}_k^t, z) = \frac{f_k(\mathbf{w}_k^t + \epsilon(z\odot\mathbf{m})) - f_k(\mathbf{w}_k^t - \epsilon(z\odot\mathbf{m}))}{2\epsilon}\cdot(z\odot\mathbf{m}).$$

To proceed, we apply a first-order Taylor expansion of $f_k$ around $\mathbf{w}_k^t$ for small $\epsilon$:

$$f_k(\mathbf{w}_k^t \pm \epsilon(z\odot\mathbf{m})) = f_k(\mathbf{w}_k^t) \pm \epsilon\langle\nabla f_k(\mathbf{w}_k^t), z\odot\mathbf{m}\rangle + \mathcal{O}(\epsilon^2).$$

Substitute these expansions into the numerator of the estimator:

$$f_k\left(\mathbf{w}_k^t + \epsilon(z\odot\mathbf{m})\right) - f_k\left(\mathbf{w}_k^t - \epsilon(z\odot\mathbf{m})\right)$$

$$= \left[f_k(\mathbf{w}_k^t) + \epsilon\langle\nabla f_k(\mathbf{w}_k^t), z\odot\mathbf{m}\rangle\right]$$

$$- \left[f_k(\mathbf{w}_k^t) - \epsilon\langle\nabla f_k(\mathbf{w}_k^t), z\odot\mathbf{m}\rangle\right] + \mathcal{O}(\epsilon^2).$$

Simplify the expression:

$$f_k(\mathbf{w}_k^t + \epsilon(z\odot\mathbf{m})) - f_k(\mathbf{w}_k^t - \epsilon(z\odot\mathbf{m})) = 2\epsilon\langle\nabla f_k(\mathbf{w}_k^t), z\odot\mathbf{m}\rangle + \mathcal{O}(\epsilon^2).$$

Thus, the estimator becomes:

$$\hat{\nabla}f_k(\mathbf{w}_k^t, z) = \frac{2\epsilon\langle\nabla f_k(\mathbf{w}_k^t), z\odot\mathbf{m}\rangle + \mathcal{O}(\epsilon^2)}{2\epsilon}\cdot(z\odot\mathbf{m}) = \left[\langle\nabla f_k(\mathbf{w}_k^t), z\odot\mathbf{m}\rangle + \mathcal{O}(\epsilon)\right](z\odot\mathbf{m}).$$

As $\epsilon \to 0$, the $\mathcal{O}(\epsilon)$ term disappears, yielding the approximation:

$$\hat{\nabla}f_k(\mathbf{w}_k^t, z) \approx \langle\nabla f_k(\mathbf{w}_k^t), z\odot\mathbf{m}\rangle\cdot(z\odot\mathbf{m}).$$

Next, compute the expectation $\mathbb{E}_z\left[\hat{\nabla}f_k(\mathbf{w}_k^t, z)\right]$. Since the estimator is a vector, consider its $j$-th component:

$$\left[\hat{\nabla}f_k(\mathbf{w}_k^t, z)\right]_j \approx \langle\nabla f_k(\mathbf{w}_k^t), z\odot\mathbf{m}\rangle\cdot(z_j m_j).$$

Express the inner product explicitly:

$$\langle\nabla f_k(\mathbf{w}_k^t), z\odot\mathbf{m}\rangle = \sum_{i=1}^{d}(\nabla f_k(\mathbf{w}_k^t))_i z_i m_i.$$

Thus, the $j$-th component is:

$$\left[\hat{\nabla}f_k(\mathbf{w}_k^t, z)\right]_j \approx \left(\sum_{i=1}^{d}(\nabla f_k(\mathbf{w}_k^t))_i z_i m_i\right)z_j m_j.$$

Now, take the expectation over $z \sim \mathcal{N}(0, \mathbf{I}_d)$, where $z_i$ are independent standard normal variables:

$$\mathbb{E}_z \left[ \left( \sum_{i=1}^{d} (\nabla f_k(\mathbf{w}_k^t))_i z_i m_i \right) z_j m_j \right] = \sum_{i=1}^{d} (\nabla f_k(\mathbf{w}_k^t))_i m_i m_j \mathbb{E}[z_i z_j].$$

Since $\mathbb{E}[z_i z_j] = \delta_{ij}$ (1 if $i = j$, 0 otherwise), the sum reduces to:

$$(\nabla f_k(\mathbf{w}_k^t))_j m_j^2 \mathbb{E}[z_j^2].$$

Given $m_j^2 = m_j$ (as $m_j = 0$ or 1) and $\mathbb{E}[z_j^2] = 1$, this becomes:

$$(\nabla f_k(\mathbf{w}_k^t))_j m_j.$$

Thus, for each component $j$:

$$\mathbb{E}_z \left[ \left[ \hat{\nabla} f_k(\mathbf{w}_k^t, z) \right]_j \right] \approx m_j (\nabla f_k(\mathbf{w}_k^t))_j.$$

This implies:

$$\mathbb{E}_z \left[ \hat{\nabla} f_k(\mathbf{w}_k^t, z) \right] \approx \mathbf{m} \odot \nabla f_k(\mathbf{w}_k^t).$$

Finally, as $\epsilon \to 0$, the higher-order terms in the Taylor expansion vanish, making the approximation exact:

$$\mathbb{E}_{\bar{z}} \left[ \hat{\nabla} f_k(\mathbf{w}_k^t, \bar{z}) \right] = \mathbf{m} \odot \nabla f_k(\mathbf{w}_k^t).$$

$\square$

### C.3 MEERKAT CONVERGENCE ANALYSIS

We consider the **federated zeroth-order optimization problem**, where the objective is to minimize the global loss function(Ling et al., 2024):

$$\min_{\mathbf{w}} f(\mathbf{w}) = \sum_{k=1}^{K} p_k f_k(\mathbf{w})$$

Each client performs $T$ local steps:

$$\mathbf{w}_k^{t+1} = \mathbf{w}_k^t - \eta \nabla f_k^t(\mathbf{w}), \quad t = 0, 1, \dots, T-1$$

starting from the global model $\mathbf{w}_k^0 = \mathbf{w}^r$. After clients finish local updates, the server performs weighted aggregation of their model updates.

$$\mathbf{w}^{r+1} = \sum_{k=1}^{K} p_k \mathbf{w}_k^r.$$

**Theorem C.9.** *[Client Local ZO Update Convergence] Let $f_k$ be $L$-smooth and $\hat{\nabla} f_k^t$ be an unbiased sparse zeroth-order gradient estimator with variance bounded by $\sigma^2$. Then we have*

*If we set constant learning rate $\eta = \frac{1}{L(u+2)}$ and $T$ local steps, the output of client $k$ satisfies:*

$$\frac{1}{T} \sum_{t=0}^{T-1} \mathbb{E} \| \nabla f_k(\mathbf{w}_k^t) \|^2 \leq \mathcal{O} \left( \frac{1}{T} \right) + \mathcal{O}(\sigma^2). \tag{5}$$

*Proof.* We start by proving Theorem C.9 euqation 5 that each client achieves local convergence during training with sparse zeroth-order finetuning. Next, we demonstrate that server-side aggregation also converge. Finally, by leveraging the PL inequality, we prove that MEERKAT exhibits linear convergence to global minimum.

**Part 1: Client Local ZO Update Convergence**

We analyze the effect of one local step of MEERKAT under sparse zeroth-order updates. Let client $k$ perform the local update:

$$\mathbf{w}_k^{t+1} = \mathbf{w}_k^t - \eta\hat{\nabla}f_k^t,$$

where the stochastic sparse zeroth-order gradient estimator is defined as:

$$g_k^t = \frac{f_k(\mathbf{w}_k^t + \epsilon(\mathbf{z}_k^t \odot \mathbf{m})) - f_k(\mathbf{w}_k^t - \epsilon(\mathbf{z}_k^t \odot \mathbf{m}))}{2\epsilon}.$$

$$\hat{\nabla}f_k^t = g_k^t \cdot (\mathbf{z}_k^t \odot \mathbf{m})$$

**Descent via Lipschitz smoothness.** Since $f_k(\mathbf{w})$ is Lipschitz smoothness:

$$f_k(\mathbf{w}_k^{t+1}) \leq f_k(\mathbf{w}_k^t) + \langle \nabla f_k(\mathbf{w}_k^t), \mathbf{w}_k^{t+1} - \mathbf{w}_k^t \rangle + \frac{L}{2}\|\mathbf{w}_k^{t+1} - \mathbf{w}_k^t\|^2.$$

Substituting the update $\mathbf{w}_k^{t+1} - \mathbf{w}_k^t = -\eta\hat{\nabla}f_k^t$, we obtain:

$$f_k(\mathbf{w}_k^{t+1}) \leq f_k(\mathbf{w}_k^t) - \eta\langle \nabla f_k(\mathbf{w}_k^t), \hat{\nabla}f_k^t(\mathbf{w}, \bar{\mathbf{z}}_t) \rangle + \frac{L\eta^2}{2}\|\hat{\nabla}f_k^t(\mathbf{w}, \bar{\mathbf{z}}_t)\|^2.$$

Taking expectation, we have:

$$\mathbb{E}_{\bar{\mathbf{z}}}[f_k(\mathbf{w}_k^{t+1})] \leq \mathbb{E}_{\bar{\mathbf{z}}}[f_k(\mathbf{w}_k^t)] - \eta\mathbb{E}_{\bar{\mathbf{z}}}\|\mathbf{m} \odot \nabla f_k(\mathbf{w}_k^t)\|^2 + \frac{L\eta^2}{2}\mathbb{E}_{\bar{\mathbf{z}}}\|\hat{\nabla}f_k(\mathbf{w}_k^t, \bar{\mathbf{z}}_t)\|^2.$$

$$\mathbb{E}_{\bar{\mathbf{z}}}[f_k(\mathbf{w}_k^{t+1})] \leq \mathbb{E}_{\bar{\mathbf{z}}}[f_k(\mathbf{w}_k^t)] - c\eta\mathbb{E}_{\bar{\mathbf{z}}}\|\nabla f_k(\mathbf{w}_k^t)\|^2 + \frac{L\eta^2}{2}(2+u)c\mathbb{E}_{\bar{\mathbf{z}}}\|\nabla f_k(\mathbf{w}_k^t)\|^2.$$

$$\mathbb{E}_{\bar{\mathbf{z}}}f_k(\mathbf{w}_k^{t+1}) \leq \mathbb{E}_{\bar{\mathbf{z}}}f_k(\mathbf{w}_k^t) - \left(c\eta_t - \frac{L\eta_t^2}{2}c(u+2)\right)\|\nabla_{\mathbf{w}}f_k(\mathbf{w}_k^t)\|^2 + \frac{L\eta_t^2}{2}c(u+2)\sigma^2.$$

Denote $\alpha = Lc(u+2)$, we can rewrite as:

$$\mathbb{E}_{\bar{\mathbf{z}}}f_k(\mathbf{w}_k^{t+1}) \leq \mathbb{E}_{\bar{\mathbf{z}}}\left\{f_k(\mathbf{w}_k^t) - \eta_t\left(c - \frac{\alpha}{2}\eta_t\right)\|\nabla_{\mathbf{w}}f_k(\mathbf{w}_k^t)\|^2\right\} + \frac{\alpha}{2}\sigma^2\eta_t^2.$$

From the above inequality, we get $\eta < \frac{2c}{\alpha}$. Suppose we use a constant learning rate $\eta_t = \eta = \frac{c}{\alpha} = \frac{1}{L(u+2)}$, we get:

$$\mathbb{E}_{\bar{\mathbf{z}}}f_k(\mathbf{w}_k^{t+1}) \leq \mathbb{E}_{\bar{\mathbf{z}}}\left\{f_k(\mathbf{w}_k^t) - \frac{c\eta}{2}\|\nabla_{\mathbf{w}}f_k(\mathbf{w}_k^t)\|^2\right\} + \frac{\alpha}{2}\sigma^2\eta^2. \tag{6}$$

**Accumulating over $T$ steps.** Summing equation 6 over $t = 0$ to $T-1$, we get:

$$\begin{aligned}
\frac{1}{T}\sum_{t=0}^{T-1}\mathbb{E}_{\bar{\mathbf{z}}}\|\nabla f_k(\mathbf{w}_k^t)\|^2 &\leq \frac{2}{c\eta T}(f_k(\mathbf{w}_k^0) - f_k^*) + \frac{1}{T}\sum_{t=0}^{T-1}\frac{\alpha}{2c\eta}\sigma^2\eta^2 \\
&= \frac{2L(u+2)}{cT}(f_k(\mathbf{w}_k^0) - f_k^*) + \sigma^2 \\
&= \mathcal{O}\left(\frac{u}{T}(f_k(\mathbf{w}_k^0) - f_k^*)\right) + \mathcal{O}(1).
\end{aligned} \tag{7}$$

$\square$

## C.4 MEERKAT CONVERGENCE ANALYSIS

We now proceed to analyze the convergence of the global model in our federated learning framework. Having established the convergence properties of local client updates, we demonstrate how these results extend to guarantee the convergence of the server-aggregated global model.

*Proof.* We approach this proof systematically by analyzing how the local convergence properties of clients extend to the global model through the aggregation process.

**Global Model Update Representation.** First, the global model update can be represented as:

$$\mathbf{w}^{r+1} - \mathbf{w}^r = \sum_{k=1}^{K} p_k (\mathbf{w}_k^T - \mathbf{w}^r)$$

where each client $k$ starts from the global model $\mathbf{w}^r$ and performs $T$ local updates to reach $\mathbf{w}_k^T$.

**Client Local Update Accumulation** For any client $k$, the accumulated local updates can be expressed as:

$$\mathbf{w}_k^{r,T} - \mathbf{w}^r = -\eta \sum_{t=0}^{T-1} \hat{\nabla} f_k^t$$

**Global Loss Descent Analysis** By the $L$-smoothness property (Assumption C.1), we have:

$$f(\mathbf{w}^{r+1}) \leq f(\mathbf{w}^r) + \langle \nabla f(\mathbf{w}^r), \mathbf{w}^{r+1} - \mathbf{w}^r \rangle + \frac{L}{2} \|\mathbf{w}^{r+1} - \mathbf{w}^r\|^2 \tag{8}$$

For the inner product we can get:

$$\langle \nabla f(\mathbf{w}^r), \mathbf{w}^{r+1} - \mathbf{w}^r \rangle = \sum_{k=1}^{K} p_k \langle \nabla f(\mathbf{w}^r), \mathbf{w}_k^{r,T} - \mathbf{w}^r \rangle$$

Accoding to the client local update process, we have:

$$\sum_{k=1}^{K} p_k \langle \nabla f_k(\mathbf{w}^r), \mathbf{w}_k^{r,T} - \mathbf{w}^r \rangle = -\eta \sum_{k=1}^{K} p_k \langle \nabla f(\mathbf{w}^r), \sum_{t=0}^{T-1} \hat{\nabla} f_k(\mathbf{w}^{r,t}, \bar{\mathbf{z}}_t) \rangle$$
$$= -\eta \sum_{k=1}^{K} p_k \sum_{t=0}^{T-1} \langle \nabla f(\mathbf{w}^r), \hat{\nabla} f_k(\mathbf{w}^{r,t}, \bar{\mathbf{z}}_t) \rangle$$

We assume that each client's weight is equal $p_k = 1/K$, by substituting it into the above inequality, we have:

$$\sum_{k=1}^{K} p_k \langle \nabla f_k(\mathbf{w}^r), \mathbf{w}_k^{r,T} - \mathbf{w}^r \rangle = -\frac{\eta}{K} \sum_{k=1}^{K} \sum_{t=0}^{T-1} \langle \nabla f(\mathbf{w}^r), \hat{\nabla} f_k(\mathbf{w}^{r,t}, \bar{\mathbf{z}}_t) \rangle. \tag{9}$$

Based on the equation 9 and $\hat{\nabla} f_k^t$ is unbiased, we have:

$$\sum_{k=1}^{K} \sum_{t=0}^{T-1} \langle \nabla f(w^r), \hat{\nabla} f_k(\mathbf{w}^{r,t}, \bar{\mathbf{z}}_t) \rangle = \sum_{k=1}^{K} \sum_{t=0}^{T-1} \langle \nabla f(w^r), \mathbb{E}_{\bar{z}}[\hat{\nabla} f_k(\mathbf{w}^{r,t}, \bar{\mathbf{z}}_t)] \rangle$$

We substitute the equation 4 and get:

$$\sum_{k=1}^{K} \sum_{t=0}^{T-1} \langle \nabla f(w^r), \mathbb{E}_{\bar{z}}[\hat{\nabla} f_k(\mathbf{w}^{r,t}, \bar{\mathbf{z}}_t)] \rangle = \sum_{k=1}^{K} \sum_{t=0}^{T-1} \langle \nabla f(w^r), \mathbf{m} \odot \nabla f_k(w^{r,t}) \rangle.$$

Under the Cauchy–Schwarz inequality, we have:

$$\left\langle \nabla f(\mathbf{w}^r),\ \mathbf{m} \odot \nabla f_k(\mathbf{w}^{r,t}) \right\rangle \leq \|\nabla f(\mathbf{w}^r)\| \, \|\mathbf{m} \odot \nabla f_k(\mathbf{w}^{r,t})\|$$

We substitute Assumption C.6 get:

$$\|\nabla f(\mathbf{w}^r)\| \, \|\mathbf{m} \odot \nabla f_k(\mathbf{w}^{r,t})\| = \sqrt{c}\, \|\nabla f(\mathbf{w}^r)\| \, \|\nabla f_k(\mathbf{w}^{r,t})\|.$$

Thus we get:

$$\left\langle \nabla f(\mathbf{w}^r),\ \mathbf{m} \odot \nabla f_k(\mathbf{w}^{r,t}) \right\rangle \leq \sqrt{c}\, \|\nabla f(\mathbf{w}^r)\| \, \|\nabla f_k(\mathbf{w}^{r,t})\|.$$

By the triangle inequality, we have

$$\|\nabla f_k(\mathbf{w}^{r,t})\| \leq \|\nabla f(\mathbf{w}^r)\| + \|\nabla f_k(\mathbf{w}^{r,t}) - \nabla f(\mathbf{w}^r)\|$$

We substitute Assumption C.3 and use the properties of square roots we get:

$$\|\nabla f(\mathbf{w}^r)\| + \|\nabla f_k(\mathbf{w}^{r,t}) - \nabla f(\mathbf{w}^r)\|$$
$$\leq \|\nabla f(\mathbf{w}^r)\| + \sqrt{c_h \|\nabla f(\mathbf{w}^r)\|^2 + \sigma_h^2}$$
$$\leq (1 + \sqrt{c_h}) \|\nabla f(\mathbf{w}^r)\| + \sigma_h.$$

Using the bound $\langle \nabla f(\mathbf{w}^r),\ m \odot \nabla f_k(\mathbf{w}^{r,t}) \rangle \leq \sqrt{c}\, \|\nabla f(\mathbf{w}^r)\| \, \|\nabla f_k\|$ from Cauchy–Schwarz and Assumption C.6, and then plugging in the above, we obtain

$$\langle \nabla f(\mathbf{w}^r),\ m \odot \nabla f_k(\mathbf{w}^{r,t}) \rangle \leq \sqrt{c}\, \|\nabla f(\mathbf{w}^r)\| \left[ (1 + \sqrt{c_h}) \|\nabla f(\mathbf{w}^r)\| + \sigma_h \right]$$
$$\leq \sqrt{c}\,(1 + \sqrt{c_h}) \|\nabla f(\mathbf{w}^r)\|^2 + \sqrt{c}\, \sigma_h \|\nabla f(\mathbf{w}^r)\|.$$

Recall that the server update inner product is

$$\left\langle \nabla f(w^r),\ w^{r+1} - w^r \right\rangle = -\frac{\eta}{K} \sum_{k=1}^{K} \sum_{t=0}^{T-1} \langle \nabla f,\ m \odot \nabla f_k \rangle.$$

Substituting the bound to equation 9. We have:

$$\left\langle \nabla f(w^r),\ w^{r+1} - w^r \right\rangle \geq -\eta\, T \sqrt{c}\,(1 + \sqrt{c_h}) \|\nabla f(w^r)\|^2 - \eta\, T \sqrt{c}\, \sigma_h \|\nabla f(w^r)\|. \tag{10}$$

Substituting this inequality to equation 8, we have:

$$f(w^{r+1}) \leq f(w^r) - \eta\, T \sqrt{c}\,(1 + \sqrt{c_h}) \|\nabla f(w^r)\|^2$$
$$- \eta\, T \sqrt{c}\, \sigma_h \|\nabla f(w^r)\| + \frac{L}{2} \|w^{r+1} - w^r\|^2 \tag{11}$$

Applying Jensen's inequality, the last term of the equation 11 will be:

$$\|\mathbf{w}^{r+1} - \mathbf{w}^r\|^2 \leq \eta^2 \sum_{k=1}^{K} p_k \| \sum_{t=0}^{T-1} \hat{\nabla} f_k^{r,t} \|^2$$

And then we apply Cauchy-Schwarz inequality, the last term of the equation 11 will be:

$$\|\mathbf{w}^{r+1} - \mathbf{w}^r\|^2 \leq \eta^2 T \sum_{k=1}^{K} p_k \sum_{t=0}^{T-1} \| \hat{\nabla} f_k^{r,t} \|^2$$

Substitute this inequaltiy to equation 11 We get:

$$f(w^{r+1}) \leq f(w^r) - \eta\, T \sqrt{c}\,(1 + \sqrt{c_h}) \|\nabla f(w^r)\|^2 - \eta\, T \sqrt{c}\, \sigma_h \|\nabla f(w^r)\|$$
$$+ \frac{L}{2} \eta^2\, T \sum_{k=1}^{K} p_k \sum_{t=0}^{T-1} \| \hat{\nabla} f_k^{r,t} \|^2.$$

Taking Expectation and lemma C.7:

$$
\mathbb{E}_{\bar{z}}\, f(\mathbf{w}^{r+1}) \leq \mathbb{E}_{\bar{z}}\, f(\mathbf{w}^r) - \eta\, T\, \sqrt{c}\,(1 + \sqrt{c_h})\,\|\nabla f(\mathbf{w}^r)\|^2
$$

$$
- \eta\, T\, \sqrt{c}\, \sigma_h\, \|\nabla f(\mathbf{w}^r)\| + \frac{L\,\eta^2\, T\,(2 + u)\, c}{2K} \sum_{k=1}^{K} \sum_{t=0}^{T-1} \|\nabla f_k(\mathbf{w}^{r,t})\|^2 \,.
$$

According to the equation 7, we know that the client-average squared gradient has upper bound. We substitute the equation 7 to the above inequality last term we get:

$$
\mathbb{E}_{\bar{z}}\, f(w^{r+1}) \leq \mathbb{E}_{\bar{z}}\, f(w^r) - \eta\, T\, \sqrt{c}\,(1 + \sqrt{c_h})\,\|\nabla f(w^r)\|^2 - \eta\, T\, \sqrt{c}\, \sigma_h\, \|\nabla f(w^r)\|
$$

$$
+ \frac{L\,\eta^2\, T\,(2 + u)\, c}{2K} \sum_{k=1}^{K} \left[ \frac{2\, L\,(u + 2)}{c} \big(f_k(w^r) - f_k^*\big) + T\, \sigma^2 \right]
$$

$$
\leq \mathbb{E}_{\bar{z}}\, f(w^r) - \eta\, T\, \sqrt{c}\,(1 + \sqrt{c_h})\,\|\nabla f(w^r)\|^2 - \eta\, T\, \sqrt{c}\, \sigma_h\, \|\nabla f(w^r)\|
$$

$$
+ \frac{L^2\,\eta^2\, T\,(2 + u)\,(u + 2)}{K} \sum_{k=1}^{K} \big(f_k(w^r) - f_k^*\big) + \frac{L\,\eta^2\, T^2\,(2 + u)\, c}{2}\, \sigma^2 \qquad (12)
$$

**Accumulating Over $R$ Rounds.** Summing equation 12 over $r = 0$ to $R - 1$, we get:

$$
\mathbb{E}_{\bar{z}}\big[f(w^R)\big] - \mathbb{E}_{\bar{z}}\big[f(w^0)\big] \leq - \eta\, T\, \sqrt{c}\,(1 + \sqrt{c_h}) \sum_{r=0}^{R-1} \big\|\nabla f(w^r)\big\|^2
$$

$$
- \eta\, T\, \sqrt{c}\, \sigma_h \sum_{r=0}^{R-1} \big\|\nabla f(w^r)\big\|
$$

$$
\qquad\qquad (13)
$$

$$
+ \frac{L^2\,\eta^2\, T\,(2 + u)\,(u + 2)}{K} \sum_{r=0}^{R-1} \sum_{k=1}^{K} \big(f_k(w^r) - f_k^*\big)
$$

$$
+ \frac{L\,\eta^2\, T^2\,(2 + u)\, c}{2}\, \sigma^2\, R \,.
$$

From the accumulated global descent inequality over $R$ rounds:

First we set

$$
S = \sum_{r=0}^{R-1} \|\nabla f(w^r)\|^2.
$$

This represents the sum of squared gradient norms over $R$ rounds. The second term in the inequality involves $\sum_{r=0}^{R-1} \|\nabla f(w^r)\|$, and we apply the Cauchy-Schwarz inequality to it. For the sequence $a_r = \|\nabla f(w^r)\|$ (with $r = 0, 1, \ldots, R-1$), we consider it as a vector in $\mathbb{R}^R$ along with a vector of ones:

$$
\sum_{r=0}^{R-1} \|\nabla f(w^r)\| = \sum_{r=0}^{R-1} \|\nabla f(w^r)\| \cdot 1 \leq \sqrt{\sum_{r=0}^{R-1} \|\nabla f(w^r)\|^2} \cdot \sqrt{\sum_{r=0}^{R-1} 1^2}.
$$

Since $\sum_{r=0}^{R-1} 1^2 = R$, we obtain:

$$
\sum_{r=0}^{R-1} \|\nabla f(w^r)\| \leq \sqrt{\sum_{r=0}^{R-1} \|\nabla f(w^r)\|^2} \cdot \sqrt{R} = \sqrt{R}\sqrt{S} = \sqrt{RS}.
$$

Substituting this into the second term, we have:

$$\eta T \sqrt{c} \sigma_h \sum_{r=0}^{R-1} \|\nabla f(w^r)\| \leq \eta T \sqrt{c} \sigma_h \sqrt{RS}.$$

Thus, the inequality becomes:

$$\mathbb{E}_{\bar{z}}[f(w^R)] - \mathbb{E}_{\bar{z}}[f(w^0)] \leq -\eta T \sqrt{c}(1 + \sqrt{c_h})S + \eta T \sqrt{c} \sigma_h \sqrt{RS}$$
$$+ \frac{L^2 \eta^2 T(2+u)(u+2)}{K} \sum_{r=0}^{R-1} \sum_{k=1}^{K} (f_k(w^r) - f_k^*)$$
$$+ \frac{L \eta^2 T^2 (2+u)c}{2} \sigma^2 R.$$

Second, we focus on the term $\eta T \sqrt{c} \sigma_h \sqrt{RS}$ and apply Young's Inequality with $\delta > 0$ and non-negative real numbers $x$ and $y$,

$$xy \leq \frac{x^2}{2\delta} + \frac{y^2 \delta}{2}.$$

We identify $x = \sqrt{S}$ and $y = \eta T \sqrt{c} \sigma_h \sqrt{R}$, since:

$$\eta T \sqrt{c} \sigma_h \sqrt{RS} = (\eta T \sqrt{c} \sigma_h \sqrt{R}) \cdot \sqrt{S}.$$

Applying Young's Inequality:

$$\sqrt{S} \cdot (\eta T \sqrt{c} \sigma_h \sqrt{R}) \leq \frac{(\sqrt{S})^2}{2\delta} + \frac{(\eta T \sqrt{c} \sigma_h \sqrt{R})^2 \delta}{2}.$$

Therefore:

$$\eta T \sqrt{c} \sigma_h \sqrt{RS} \leq \frac{S}{2\delta} + \frac{\eta^2 T^2 c \sigma_h^2 R \delta}{2}.$$

$$-\eta T \sqrt{c} \sigma_h \sqrt{RS} \leq \frac{S}{2\delta} + \frac{\eta^2 T^2 c \sigma_h^2 R \delta}{2}.$$

Finally we replace the second term in the inequality with the above result:

$$\mathbb{E}_{\bar{z}}[f(w^R)] - \mathbb{E}_{\bar{z}}[f(w^0)] \leq -\eta T \sqrt{c}(1 + \sqrt{c_h})S + \left( \frac{S}{2\delta} + \frac{\eta^2 T^2 c \sigma_h^2 R \delta}{2} \right)$$
$$+ \frac{L^2 \eta^2 T(2+u)^2}{K} \sum_{r=0}^{R-1} \sum_{k=1}^{K} (f_k(w^r) - f_k^*)$$
$$+ \frac{L \eta^2 T^2 (2+u)c}{2} \sigma^2 R.$$

This inequality now depends on $\delta$.

$$\left( \eta T \sqrt{c} \left( 1 + \sqrt{c_h} \right) - \frac{1}{2\delta} \right) \sum_{r=0}^{R-1} \|\nabla f(w^r)\|^2 \leq \mathbb{E}_{\bar{z}}[f(w^0) - f(w^R)] + \eta^2 T^2 c \sigma_h^2 R \delta 2$$
$$+ \frac{L^2 \eta^2 T (2+u)^2}{K} \sum_{r=0}^{R-1} \sum_{k=1}^{K} (f_k(w^r) - f_k^*) \quad (14)$$
$$+ \frac{L \eta^2 T^2 (2+u)c \sigma^2 R}{2} .$$

According to Assumption C.1, we have:

$$f_k(\mathbf{w}^*) \leq f_k(\mathbf{w}_k^*) + \langle \nabla f_k(\mathbf{w}_k^*), \mathbf{w}^* - \mathbf{w}_k^* \rangle + \frac{L}{2} \|\mathbf{w}^* - \mathbf{w}_k^*\|_2^2.$$

Since $\mathbf{w}_k^*$ is the minimizer of $f_k(\mathbf{w})$, the gradient at the local optimum must be zero:

$$\nabla f_k(\mathbf{w}_k^*) = 0.$$

Substituting this into the inner product term:

$$\langle \nabla f_k(\mathbf{w}_k^*), \mathbf{w}^* - \mathbf{w}_k^* \rangle = \langle 0, \mathbf{w}^* - \mathbf{w}_k^* \rangle = 0.$$

Thus, the inner product term disappears because the gradient at $\mathbf{w}_k^*$ is zero, making the inner product with any vector (including $\mathbf{w}^* - \mathbf{w}_k^*$) equal to zero.

With the inner product term vanishing, the inequality simplifies to:

$$f_k(\mathbf{w}^*) \le f_k(\mathbf{w}_k^*) + \frac{L}{2}\Delta_k.$$

This provides an upper bound on $f_k(\mathbf{w}^*)$ in terms of the local optimal loss $f_k^*$ and the optimality gap $\Delta_k$.

The global optimal loss is defined as:

$$f^* = f(\mathbf{w}^*) = \sum_{k=1}^{K} p_k f_k(\mathbf{w}^*).$$

Using the bound derived for each local loss:

$$f_k(\mathbf{w}^*) \le f_k^* + \frac{L}{2}\Delta_k,$$

we substitute this into the expression for $f^*$:

$$f^* = \sum_{k=1}^{K} p_k f_k(\mathbf{w}^*) \le \sum_{k=1}^{K} p_k \left( f_k^* + \frac{L}{2}\Delta_k \right).$$

Expanding the right-hand side:

$$f^* \le \sum_{k=1}^{K} p_k f_k^* + \frac{L}{2} \sum_{k=1}^{K} p_k \Delta_k.$$

From the above equation, we have:

$$f^* - \frac{L}{2} \sum_{k=1}^{K} p_k \Delta_k \le \sum_{k=1}^{K} p_k f_k^*.$$

$$-\frac{1}{K} \sum_{k=1}^{K} f_k^* \le -f^* + \frac{L}{2K} \sum_{k=1}^{K} \Delta_k.$$

From the equation 14, we have the term:

$$\frac{L^2 \eta^2 T (2+u)^2}{K} \sum_{r=0}^{R-1} \sum_{k=1}^{K} \left( f_k(w^r) - f_k^* \right).$$

First, we express the double sum as:

$$\sum_{r=0}^{R-1} \sum_{k=1}^{K} \left( f_k(w^r) - f_k^* \right) = \sum_{r=0}^{R-1} \left( \sum_{k=1}^{K} f_k(w^r) - \sum_{k=1}^{K} f_k^* \right).$$

Since $p_k = \frac{1}{K}$, we have:

$$\sum_{k=1}^{K} f_k(w^r) = K f(w^r),$$

where $f(w^r) = \sum_{k=1}^{K} p_k f_k(w^r) = \frac{1}{K} \sum_{k=1}^{K} f_k(w^r)$. Therefore:

$$\sum_{r=0}^{R-1} \sum_{k=1}^{K} (f_k(w^r) - f_k^*) = \sum_{r=0}^{R-1} \left( K f(w^r) - \sum_{k=1}^{K} f_k^* \right).$$

From the earlier derivation, we have the inequality:

$$-\frac{1}{K} \sum_{k=1}^{K} f_k^* \leq -f^* + \frac{L}{2K} \sum_{k=1}^{K} \Delta_k.$$

Substituting this into the expression above:

$$\sum_{r=0}^{R-1} \sum_{k=1}^{K} (f_k(w^r) - f_k^*) \leq \sum_{r=0}^{R-1} \left( K f(w^r) - \left( K f^* - \frac{L}{2} \sum_{k=1}^{K} \Delta_k \right) \right).$$

Thus:

$$\sum_{r=0}^{R-1} \sum_{k=1}^{K} (f_k(w^r) - f_k^*) \leq \sum_{r=0}^{R-1} \left( K f(w^r) - K f^* + \frac{L}{2} \sum_{k=1}^{K} \Delta_k \right).$$

Since $\Delta_k$ is constant across iterations, we can factor it out:

$$K \sum_{r=0}^{R-1} (f(w^r) - f^*) + \frac{L}{2} \sum_{r=0}^{R-1} \sum_{k=1}^{K} \Delta_k = K \sum_{r=0}^{R-1} (f(w^r) - f^*) + \frac{LR}{2} \sum_{k=1}^{K} \Delta_k.$$

Now, multiply by the coefficient:

$$\frac{L^2 \eta^2 T (2+u)^2}{K} \sum_{r=0}^{R-1} \sum_{k=1}^{K} (f_k(w^r) - f_k^*) \leq \frac{L^2 \eta^2 T (2+u)^2}{K} \left[ K \sum_{r=0}^{R-1} (f(w^r) - f^*) + \frac{LR}{2} \sum_{k=1}^{K} \Delta_k \right].$$

Simplifying:

$$L^2 \eta^2 T (2+u)^2 \sum_{r=0}^{R-1} (f(w^r) - f^*) + \frac{L^3 \eta^2 T (2+u)^2 R}{2K} \sum_{k=1}^{K} \Delta_k.$$

Substituting this result into the original target inequality, we get:

$$\left( \eta T \sqrt{c} \left( 1 + \sqrt{c_h} \right) - \frac{1}{2\delta} \right) \sum_{r=0}^{R-1} \|\nabla f(w^r)\|^2 \leq \mathbb{E}_{\bar{z}} \left[ f(w^0) - f(w^R) \right] + \frac{\eta^2 T^2 c \sigma_h^2 R \delta}{2}$$

$$+ L^2 \eta^2 T (2+u)^2 \sum_{r=0}^{R-1} (f(w^r) - f^*)$$

$$+ \frac{L^3 \eta^2 T (2+u)^2 R}{2K} \sum_{k=1}^{K} \Delta_k$$

$$+ \frac{L \eta^2 T^2 (2+u) c \sigma^2 R}{2}.$$

According to the Assumption C.2 we have:

$$2\mu (f(\mathbf{w}^r) - f^*) \leq \|\nabla f(\mathbf{w}^r)\|^2, \quad \forall \mathbf{w}^r \in \mathbb{R}^d,$$

$$2\mu \sum_{r=0}^{R-1} (f(\mathbf{w}^r) - f^*) \leq \sum_{r=0}^{R-1} \|\nabla f(\mathbf{w}^r)\|^2, \quad \forall \mathbf{w}^r \in \mathbb{R}^d,$$

We let $\eta T \sqrt{c}\left(1 + \sqrt{c_h}\right) - \frac{1}{2\delta} > 0$ and substitute the above inequality, we have:

$$2\mu(\eta T \sqrt{c}\left(1 + \sqrt{c_h}\right) - \frac{1}{2\delta}) \sum_{r=0}^{R-1}(f(\mathbf{w}^r) - f^*) \leq \mathbb{E}_{\bar{z}}\left[f(w^0) - f(w^R)\right] + \frac{\eta^2 T^2 c \sigma_h^2 R \delta}{2}$$

$$+ L^2 \eta^2 T (2+u)^2 \sum_{r=0}^{R-1}(f(w^r) - f^*)$$

$$+ \frac{L^3 \eta^2 T (2+u)^2 R}{2K} \sum_{k=1}^{K} \Delta_k$$

$$+ \frac{L \eta^2 T^2 (2+u) c \sigma^2 R}{2}.$$

$$\sum_{r=0}^{R-1}\left(f(w^r) - f^*\right) \leq \frac{\mathbb{E}_{\bar{z}}\left[f(w^0) - f(w^R)\right]}{2\mu\left(\eta T \sqrt{c}(1 + \sqrt{c_h}) - \frac{1}{2\delta}\right) - L^2 \eta^2 T (2+u)^2}$$

$$+ \frac{\eta^2 T^2 c \sigma_h^2 R \delta}{2\left[2\mu\left(\eta T \sqrt{c}(1 + \sqrt{c_h}) - \frac{1}{2\delta}\right) - L^2 \eta^2 T (2+u)^2\right]}$$

$$+ \frac{L^3 \eta^2 T (2+u)^2 R \sum_{k=1}^{K} \Delta_k}{2K\left[2\mu\left(\eta T \sqrt{c}(1 + \sqrt{c_h}) - \frac{1}{2\delta}\right) - L^2 \eta^2 T (2+u)^2\right]}$$

$$+ \frac{L \eta^2 T^2 (2+u) c \sigma^2 R}{2\left[2\mu\left(\eta T \sqrt{c}(1 + \sqrt{c_h}) - \frac{1}{2\delta}\right) - L^2 \eta^2 T (2+u)^2\right]}. \tag{15}$$

$$\frac{1}{R}\sum_{r=0}^{R-1}\left(f(w^r) - f^*\right) \leq \frac{1}{R} \frac{\mathbb{E}_{\bar{z}}\left[f(w^0) - f(w^R)\right]}{2\mu\left(\eta T \sqrt{c}(1 + \sqrt{c_h}) - \frac{1}{2\delta}\right) - L^2 \eta^2 T (2+u)^2}$$

$$+ \frac{\eta^2 T^2 c \sigma_h^2 \delta}{2\left[2\mu\left(\eta T \sqrt{c}(1 + \sqrt{c_h}) - \frac{1}{2\delta}\right) - L^2 \eta^2 T (2+u)^2\right]}$$

$$+ \frac{L^3 \eta^2 T (2+u)^2 \sum_{k=1}^{K} \Delta_k}{2K\left[2\mu\left(\eta T \sqrt{c}(1 + \sqrt{c_h}) - \frac{1}{2\delta}\right) - L^2 \eta^2 T (2+u)^2\right]}$$

$$+ \frac{L \eta^2 T^2 (2+u) c \sigma^2}{2\left[2\mu\left(\eta T \sqrt{c}(1 + \sqrt{c_h}) - \frac{1}{2\delta}\right) - L^2 \eta^2 T (2+u)^2\right]}. \tag{16}$$

We select $\delta = \frac{1}{\eta T \sqrt{c}(1 + \sqrt{c_h})}$, which leads to:

$$\frac{1}{2\delta} = \frac{\eta T \sqrt{c}(1 + \sqrt{c_h})}{2}$$

Substituting into the denominator:

$$2\mu\left(\eta T \sqrt{c}(1 + \sqrt{c_h}) - \frac{\eta T \sqrt{c}(1 + \sqrt{c_h})}{2}\right) = \mu \eta T \sqrt{c}(1 + \sqrt{c_h})$$

With the chosen $\delta$, we have:

$$
\begin{aligned}
\frac{1}{R} \sum_{r=0}^{R-1} (f(w^r) - f^*) \leq {} & \frac{1}{R} \cdot \frac{\mathbb{E}_{\bar{z}} \left[ f(w^0) - f(w^R) \right]}{\mu \eta T \sqrt{c}(1 + \sqrt{c_h}) - L^2 \eta^2 T (2 + u)^2} \\
& + \frac{\sqrt{c} \sigma_h^2}{2(1 + \sqrt{c_h}) \left[ \mu \sqrt{c}(1 + \sqrt{c_h}) - L^2 \eta (2 + u)^2 \right]} \\
& + \frac{L^3 \eta (2 + u)^2 \sum_{k=1}^K \Delta_k}{2K \left[ \mu \sqrt{c}(1 + \sqrt{c_h}) - L^2 \eta (2 + u)^2 \right]} \\
& + \frac{L \eta T (2 + u) c \sigma^2}{2 \left[ \mu \sqrt{c}(1 + \sqrt{c_h}) - L^2 \eta (2 + u)^2 \right]},
\end{aligned}
\tag{17}
$$

where the step-size $\eta$ must satisfy: $\eta < \frac{\mu \sqrt{c}(1+\sqrt{c_h})}{L^2(2+u)^2}$ to ensure denominator positivity.

Plugging in a constant learning rate $\eta = \min \left\{ \frac{1}{L(u+2)}, \frac{\mu \sqrt{c}\left(1+\sqrt{c_h}\right)}{2 L^2(2+u)^2} \right\}$. We substitute this $\eta$ to equation 17 and get:

$$
\begin{aligned}
\frac{1}{R} \sum_{r=0}^{R-1} (f(w^r) - f^*) \leq {} & \frac{4L^2(2+u)^2}{\mu^2 c\left(1+\sqrt{c_h}\right)^2 TR} \, \mathbb{E}_{\bar{z}}[f(w^0) - f^*] \\
& + \frac{\sigma_h^2}{\mu\left(1+\sqrt{c_h}\right)^2} + \frac{L}{K} \sum_{k=1}^K \Delta_k + \frac{T c \sigma^2}{2L\,(2+u)} \, .
\end{aligned}
$$

$$
\frac{1}{R} \sum_{r=0}^{R-1} (f(w^r) - f^*) \leq \mathcal{O}\left( \frac{(2+u)^2}{TR} \cdot \mathbb{E}[f(w^0) - f(w^R)] \right) + \mathcal{O}\left( \frac{T}{2+u} \right) + \mathcal{O}(1).
\tag{18}
$$

$\square$

## C.5 MEERKAT-VP CONVERGENCE ANALYSIS

We propose a Virtual Path Client Selection (MEERKAT-VP) mechanism that identifies clients with highly heterogeneous data distributions based on their optimization trajectories. Instead of excluding them, MEERKAT-VP applies early stopping to these clients to limit their adverse influence on global model updates while still preserving their participation.

*Proof.* **Motivation for Early Stopping:** In federated learning, clients perform local updates starting from the global model $w^r$. For $T > 1$, clients may drift towards their local optima, introducing bias into the global update due to data heterogeneity. By identifying "bad" clients and limiting them to one update step, we reduce their drift and align their contributions more closely with the global gradient.

We divide the $K$ clients into two groups:

- **Balanced-distribution clients** ($K_g$): Perform $T$ local step updates.
- **Skewed-distribution clients** ($K_b$): Perform only 1 local step update.

The global model update becomes:

$$
w^{r+1} = w^r + \frac{1}{K} \sum_{k \in K_g} (w_k^{r,T} - w^r) + \frac{1}{K} \sum_{k \in K_b} (w_k^{r,1} - w^r)
$$

where:

$$
w_k^{r,T} - w^r = -\eta \sum_{t=0}^{T-1} \hat{\nabla} f_k(w^{r,t}), \quad w_k^{r,1} - w^r = -\eta \hat{\nabla} f_k(w^r)
$$

**Loss Descent Analysis**   Using the $L$-smoothness property:

$$f(w^{r+1}) \le f(w^r) + \langle \nabla f(w^r), w^{r+1} - w^r \rangle + \frac{L}{2} \| w^{r+1} - w^r \|^2$$

We analyze the inner product term:

$$\langle \nabla f(w^r), w^{r+1} - w^r \rangle = \sum_{k=1}^{K} p_k \langle \nabla f(\mathbf{w}^r), \mathbf{w}_k^{r,T} - \mathbf{w}^r \rangle$$

$$\sum_{k=1}^{K} p_k \langle \nabla f(\mathbf{w}^r), \mathbf{w}_k^{r,T} - \mathbf{w}^r \rangle = -\eta \sum_{k=1}^{K} p_k \langle \nabla f(\mathbf{w}^r), \sum_{t=0}^{T-1} \hat{\nabla} f_k(\mathbf{w}^{r,t}, \bar{\mathbf{z}}_t) \rangle$$

$$= -\eta \sum_{k=1}^{K} p_k \sum_{t=0}^{T-1} \langle \nabla f(\mathbf{w}^r), \hat{\nabla} f_k(\mathbf{w}^{r,t}, \bar{\mathbf{z}}_t) \rangle$$

Since we have balanced-distribution clients and skewed-distribution clients:

$$\langle \nabla f(w^r), w^{r+1} - w^r \rangle = \frac{1}{K} \sum_{k \in K_g} \langle \nabla f(w^r), w_k^{r,T} - w^r \rangle + \frac{1}{K} \sum_{k \in K_b} \langle \nabla f(w^r), w_k^{r,1} - w^r \rangle$$

$$\langle \nabla f(w^r), w^{r+1} - w^r \rangle = -\frac{\eta}{K} \sum_{k \in K_g} \sum_{t=0}^{T-1} \langle \nabla f(w^r), \hat{\nabla} f_k(w^{r,t}) \rangle$$
$$- \frac{\eta}{K} \sum_{k \in K_b} \langle \nabla f(w^r), \hat{\nabla} f_k(w^r) \rangle \tag{19}$$

Since $\hat{\nabla} f_k^t$ is unbiased, we have:

$$\sum_{k=1}^{K} \sum_{t=0}^{T-1} \langle \nabla f(w^r), \hat{\nabla} f_k(\mathbf{w}^{r,t}, \bar{\mathbf{z}}_t) \rangle = \sum_{k=1}^{K} \sum_{t=0}^{T-1} \langle \nabla f(w^r), \mathbb{E}_{\bar{z}}[\hat{\nabla} f_k(\mathbf{w}^{r,t}, \bar{\mathbf{z}}_t)] \rangle$$

We substitute the equation 4 and get:

$$\sum_{k=1}^{K} \sum_{t=0}^{T-1} \langle \nabla f(w^r), \mathbb{E}_{\bar{z}}[\hat{\nabla} f_k(\mathbf{w}^{r,t}, \bar{\mathbf{z}}_t)] \rangle = \sum_{k=1}^{K} \sum_{t=0}^{T-1} \langle \nabla f(w^r), \mathbf{m} \odot \nabla f_k(w^{r,t}) \rangle.$$

Thus taking expectation of equation 19, we can get:

$$\mathbb{E}_{\bar{z}} \langle \nabla f(\mathbf{w}^r), \mathbf{w}^{r+1} - \mathbf{w}^r \rangle = -\frac{\eta}{K} \Bigg( \sum_{k \in K_g} \sum_{t=0}^{T-1} \langle \nabla f(\mathbf{w}^r), \mathbf{m} \odot \nabla f_k(\mathbf{w}^{r,t}) \rangle$$
$$+ \sum_{k \in K_b} \langle \nabla f(\mathbf{w}^r), \mathbf{m} \odot \nabla f_k(\mathbf{w}^r) \rangle \Bigg) \tag{20}$$

Under the Cauchy–Schwarz inequality, we have:

$$\langle \nabla f(\mathbf{w}^r), \ \mathbf{m} \odot \nabla f_k(\mathbf{w}^{r,t}) \rangle \ \le \ \| \nabla f(\mathbf{w}^r) \| \ \| \mathbf{m} \odot \nabla f_k(\mathbf{w}^{r,t}) \|$$

We substitute Assumption C.6 get:

$$\|\nabla f(\mathbf{w}^r)\| \, \|\mathbf{m} \odot \nabla f_k(\mathbf{w}^{r,t})\| \;=\; \sqrt{c}\, \|\nabla f(\mathbf{w}^r)\| \, \|\nabla f_k(\mathbf{w}^{r,t})\|.$$

Thus we get:

$$\langle \nabla f(\mathbf{w}^r), \; \mathbf{m} \odot \nabla f_k(\mathbf{w}^{r,t}) \rangle \;\leq\; \sqrt{c}\, \|\nabla f(\mathbf{w}^r)\| \, \|\nabla f_k(\mathbf{w}^{r,t})\|.$$

By the triangle inequality, we have

$$\|\nabla f_k(\mathbf{w}^{r,t})\| \;\leq\; \|\nabla f(\mathbf{w}^r)\| + \|\nabla f_k(\mathbf{w}^{r,t}) - \nabla f(\mathbf{w}^r)\|$$

We substitute Assumption C.3 and use the properties of square roots we get:

$$\|\nabla f(\mathbf{w}^r)\| + \|\nabla f_k(\mathbf{w}^{r,t}) - \nabla f(\mathbf{w}^r)\| \leq \|\nabla f(\mathbf{w}^r)\| + \sqrt{c_h \|\nabla f(\mathbf{w}^r)\|^2 + \sigma_h^2}$$

$$\leq (1 + \sqrt{c_h})\, \|\nabla f(\mathbf{w}^r)\| + \sigma_h.$$

Using the bound $\langle \nabla f(\mathbf{w}^r), \; m \odot \nabla f_k(\mathbf{w}^{r,t}) \rangle \leq \sqrt{c}\, \|\nabla f(\mathbf{w}^r)\| \, \|\nabla f_k\|$ from Cauchy–Schwarz and Assumption C.6, and then plugging in the above, we obtain

$$\langle \nabla f(\mathbf{w}^r), \; m \odot \nabla f_k(\mathbf{w}^{r,t}) \rangle \leq \sqrt{c}\, \|\nabla f(\mathbf{w}^r)\| \big[ (1 + \sqrt{c_h})\, \|\nabla f(\mathbf{w}^r)\| + \sigma_h \big]$$

$$\leq \sqrt{c}\, (1 + \sqrt{c_h})\, \|\nabla f(\mathbf{w}^r)\|^2 \;+\; \sqrt{c}\, \sigma_h \|\nabla f(\mathbf{w}^r)\|.$$

Since this bound holds uniformly for all $k$ and $t$, and based on the equation 20 we get:

$$\sum_{k \in K_g} \sum_{t=0}^{T-1} \langle \nabla f(\mathbf{w}^r), \mathbf{m} \odot \nabla f_k(\mathbf{w}^{r,t}) \rangle + \sum_{k \in K_b} \langle \nabla f(\mathbf{w}^r), \mathbf{m} \odot \nabla f_k(\mathbf{w}^r) \rangle$$

$$\leq (|K_g|T + |K_b|) \left[ \sqrt{c}(1 + \sqrt{c_h}) \|\nabla f(\mathbf{w}^r)\|^2 + \sqrt{c}\sigma_h \|\nabla f(\mathbf{w}^r)\| \right].$$

We get:

$$\mathbb{E}_{\bar{z}}[f(w^{r+1})] \leq \mathbb{E}_{\bar{z}}[f(w^r)] - \frac{\eta \sqrt{c}\alpha}{K}(1 + \sqrt{c_h}) \|\nabla f(w^r)\|^2$$

$$- \frac{\eta \sqrt{c}\alpha}{K} \sigma_h \|\nabla f(w^r)\| + \frac{L}{2} \mathbb{E}_{\bar{z}} \|w^{r+1} - w^r\|^2 \tag{21}$$

where $\alpha = |K_g|T + |K_b|$.

Since the global model update is given by:

$$w^{r+1} = w^r + \frac{1}{K} \sum_{k \in K_g} (w_k^{r,T} - w^r) + \frac{1}{K} \sum_{k \in K_b} (w_k^{r,1} - w^r)$$

We substitute the local updates and the squared norm is:

$$\|w^{r+1} - w^r\|^2 = \frac{\eta^2}{K^2} \left\| \sum_{k \in K_g} \sum_{t=0}^{T-1} \hat{\nabla} f_k(w^{r,t}) + \sum_{k \in K_b} \hat{\nabla} f_k(w^r) \right\|^2$$

Define the update contribution per client:

$$\hat{\Delta}_k = \begin{cases} -\eta \sum_{t=0}^{T-1} \hat{\nabla} f_k(w^{r,t}) & \text{if } k \in K_g, \\ -\eta \hat{\nabla} f_k(w^r) & \text{if } k \in K_b. \end{cases}$$

Then:

$$w^{r+1} - w^r = \frac{1}{K} \sum_{k=1}^{K} \hat{\Delta}_k$$

$$\|w^{r+1} - w^r\|^2 = \frac{1}{K^2} \left\| \sum_{k=1}^{K} \hat{\Delta}_k \right\|^2$$

Using the Cauchy-Schwarz inequality:

$$\left\| \sum_{k=1}^{K} \hat{\Delta}_k \right\|^2 \leq K \sum_{k=1}^{K} \|\hat{\Delta}_k\|^2, \quad \text{where } \hat{\Delta}_k \text{ denotes the actual model update on client } k.$$

So:

$$\|w^{r+1} - w^r\|^2 \leq \frac{1}{K} \sum_{k=1}^{K} \|\hat{\Delta}_k\|^2$$

Now compute $\|\hat{\Delta}_k\|^2$:

$$\|\hat{\Delta}_k\|^2 = \eta^2 \left\| \sum_{t=0}^{T-1} \hat{\nabla} f_k(w^{r,t}) \right\|^2 \quad \text{for } k \in K_g,$$

$$\|\hat{\Delta}_k\|^2 = \eta^2 \left\| \hat{\nabla} f_k(w^r) \right\|^2 \quad \text{for } k \in K_b.$$

Thus:

$$\|w^{r+1} - w^r\|^2 \leq \frac{\eta^2}{K} \left( \sum_{k \in K_g} \left\| \sum_{t=0}^{T-1} \hat{\nabla} f_k(w^{r,t}) \right\|^2 + \sum_{k \in K_b} \left\| \hat{\nabla} f_k(w^r) \right\|^2 \right)$$

We take the expectation:

$$\mathbb{E}_{\bar{z}} \|w^{r+1} - w^r\|^2 \leq \frac{\eta^2}{K} \left( \sum_{k \in K_g} \mathbb{E}_{\bar{z}} \left\| \sum_{t=0}^{T-1} \hat{\nabla} f_k(w^{r,t}) \right\|^2 + \sum_{k \in K_b} \mathbb{E}_{\bar{z}} \left\| \hat{\nabla} f_k(w^r) \right\|^2 \right)$$

For $k \in K_b$:

$$\mathbb{E}_{\bar{z}} \left\| \hat{\nabla} f_k(w^r) \right\|^2 = (2 + u)c \left\| \nabla f_k(w^r) \right\|^2$$

For $k \in K_g$:

$$\mathbb{E}_{\bar{z}} \left\| \sum_{t=0}^{T-1} \hat{\nabla} f_k(w^{r,t}) \right\|^2$$

Using the Cauchy-Schwarz inequality:

$$\mathbb{E}_{\bar{z}} \left\| \sum_{t=0}^{T-1} \hat{\nabla} f_k(w^{r,t}) \right\|^2 \leq T \sum_{t=0}^{T-1} \mathbb{E}_{\bar{z}} \left\| \hat{\nabla} f_k(w^{r,t}) \right\|^2$$

According to the lemma C.7:

$$\mathbb{E}_{\bar{z}} \left\| \hat{\nabla} f_k(w^{r,t}) \right\|^2 = (2 + u)c \left\| \nabla f_k(w^{r,t}) \right\|^2$$

So:

$$\mathbb{E}_{\bar{z}} \left\| \sum_{t=0}^{T-1} \hat{\nabla} f_k(w^{r,t}) \right\|^2 \leq T(2 + u)c \sum_{t=0}^{T-1} \left\| \nabla f_k(w^{r,t}) \right\|^2$$

Combine the terms we get:

$$\mathbb{E}_{\bar{z}} \|w^{r+1} - w^r\|^2 \leq \frac{\eta^2 (2 + u)c}{K} \left( T \sum_{k \in K_g} \sum_{t=0}^{T-1} \left\| \nabla f_k(w^{r,t}) \right\|^2 + \sum_{k \in K_b} \left\| \nabla f_k(w^r) \right\|^2 \right)$$

We substitute this inequality to the equation 21.

$$\begin{aligned}
\mathbb{E}_{\bar{z}}[f(w^{r+1})] \leq{}& \mathbb{E}_{\bar{z}}[f(w^r)] - \frac{\eta\sqrt{c}\alpha}{K}(1 + \sqrt{c_h}) \left\| \nabla f(w^r) \right\|^2 \\
& - \frac{\eta\sqrt{c}\alpha}{K} \sigma_h \left\| \nabla f(w^r) \right\| \\
& + \frac{\eta^2 (2 + u)cL}{2K} \left( T \sum_{k \in K_g} \sum_{t=0}^{T-1} \left\| \nabla f_k(w^{r,t}) \right\|^2 + \sum_{k \in K_b} \left\| \nabla f_k(w^r) \right\|^2 \right)
\end{aligned} \qquad (22)$$

$$\mathbb{E}_{\bar{z}}\big[f(w^{r+1})\big] \leq \mathbb{E}_{\bar{z}}\big[f(w^r)\big] - \frac{\eta\sqrt{c}\,\alpha}{K}\,(1+\sqrt{c_h})\,\big\|\nabla f(w^r)\big\|^2 - \frac{\eta\sqrt{c}\,\alpha}{K}\,\sigma_h\big\|\nabla f(w^r)\big\|$$

$$+ \frac{\eta^2(2+u)\,c\,L\,T}{2K}\sum_{k\in K_g}\sum_{t=0}^{T-1}\big\|\nabla f_k(w^{r,t})\big\|^2 + \frac{\eta^2(2+u)\,c\,L}{2K}\sum_{k\in K_b}\big\|\nabla f_k(w^r)\big\|^2$$

According to the equation 7, we know that the client-average squared gradient has upper bound.

$$\mathbb{E}_{\bar{z}}\big[f(w^{r+1})\big] \leq \mathbb{E}_{\bar{z}}\big[f(w^r)\big] - \frac{\eta\sqrt{c}\,\alpha}{K}\,(1+\sqrt{c_h})\,\big\|\nabla f(w^r)\big\|^2 - \frac{\eta\sqrt{c}\,\alpha}{K}\,\sigma_h\,\big\|\nabla f(w^r)\big\|$$

$$+ \frac{\eta^2(2+u)\,c\,L\,T}{2K}\sum_{k\in K_g}\Big[\frac{2L(2+u)}{c}\big(f_k(w_k^{0,r})-f_k^*\big)+T\,\sigma^2\Big] \tag{23}$$

$$+ \frac{\eta^2(2+u)\,c\,L}{2K}\sum_{k\in K_b}\big\|\nabla f_k(w^r)\big\|^2.$$

Using Assumption C.3, which states that for any $\theta \in \mathbb{R}^d$,

$$\big\|\nabla f(\theta) - \nabla f_i(\theta)\big\|^2 \leq c_h\,\big\|\nabla f(\theta)\big\|^2 + \sigma_h^2,$$

we can bound the squared norm of the local gradient $\big\|\nabla f_k(w^r)\big\|^2$. Specifically, by the inequality $(x+y)^2 \leq 2x^2 + 2y^2$, we have:

$$\big\|\nabla f_k(w^r)\big\|^2 = \big\|\nabla f(w^r) + (\nabla f_k(w^r) - \nabla f(w^r))\big\|^2 \leq 2\,\big\|\nabla f(w^r)\big\|^2 + 2\,\big\|\nabla f_k(w^r) - \nabla f(w^r)\big\|^2.$$

Then, applying Assumption C.3 with $\theta = w^r$ and $i = k$:

$$\big\|\nabla f_k(w^r) - \nabla f(w^r)\big\|^2 \leq c_h\,\big\|\nabla f(w^r)\big\|^2 + \sigma_h^2.$$

Therefore,

$$\big\|\nabla f_k(w^r)\big\|^2 \leq 2\,\big\|\nabla f(w^r)\big\|^2 + 2\,\Big(c_h\,\big\|\nabla f(w^r)\big\|^2 + \sigma_h^2\Big) = (2+2c_h)\,\big\|\nabla f(w^r)\big\|^2 + 2\sigma_h^2.$$

Thus, we obtain the bound:

$$\big\|\nabla f_k(w^r)\big\|^2 \leq (2+2c_h)\,\big\|\nabla f(w^r)\big\|^2 + 2\sigma_h^2.$$

We substitute the bound to the inequality 23, according to the Assumption C.3, we substitute the last term:

$$\mathbb{E}_{\bar{z}}\big[f(w^{r+1})\big] \leq \mathbb{E}_{\bar{z}}\big[f(w^r)\big] - \frac{\eta\sqrt{c}\,\alpha}{K}\,(1+\sqrt{c_h})\,\|\nabla f(w^r)\|^2 - \frac{\eta\sqrt{c}\,\alpha}{K}\,\sigma_h\,\|\nabla f(w^r)\|$$

$$+ \frac{\eta^2(2+u)\,c\,L\,T}{2K}\sum_{k\in K_g}\Big[\frac{2L(2+u)}{c}\big(f_k(w_k^{0,r})-f_k^*\big)+T\,\sigma^2\Big]$$

$$+ \frac{\eta^2(2+u)\,c\,L}{2K}\sum_{k\in K_b}\Big[(2+2c_h)\,\|\nabla f(w^r)\|^2 + \sigma_h^2\Big].$$

$$\mathbb{E}_{\bar{z}}\big[f(w^{r+1})\big] \leq \mathbb{E}_{\bar{z}}\big[f(w^r)\big] - \frac{\eta\sqrt{c}\,\alpha}{K}\,(1+\sqrt{c_h})\,\|\nabla f(w^r)\|^2 - \frac{\eta\sqrt{c}\,\alpha}{K}\,\sigma_h\,\|\nabla f(w^r)\|$$

$$+ \frac{\eta^2(2+u)\,c\,L\,T}{2K}\sum_{k\in K_g}\Big[\frac{2L(2+u)}{c}\big(f_k(w_k^{0,r})-f_k^*\big)+T\,\sigma^2\Big]$$

$$+ \frac{\eta^2(2+u)\,c\,L\,|K_b|\,(1+c_h)}{K}\,\|\nabla f(w^r)\|^2 + \frac{\eta^2(2+u)\,c\,L\,|K_b|\,\sigma_h^2}{K}.$$

$$\mathbb{E}_{\bar{z}}\big[f(w^{r+1})\big] \leq \mathbb{E}_{\bar{z}}\big[f(w^r)\big]$$

$$+ \frac{\eta^2(2+u)\,c\,L\,K_b\,(1+c_h) - \eta\sqrt{c}\,\alpha}{K}\,\big\|\nabla f(w^r)\big\|^2$$

$$- \frac{\eta\sqrt{c}\,\alpha\,\sigma_h}{K}\,\big\|\nabla f(w^r)\big\| \tag{24}$$

$$+ \frac{\eta^2(2+u)^2\,L^2 T}{K}\sum_{k \in K_g}\big(f_k(w_k^{0,r}) - f_k^*\big)$$

$$+ \frac{\eta^2(2+u)\,c\,L}{2K}\Big(T^2\,\sigma^2\,K_g + 2\,K_b\sigma_h^2\Big).$$

**Accumulating Over $R$ Rounds.** Summing equation 24 over $r = 0$ to $R-1$,

$$\mathbb{E}_{\bar{z}}\big[f(w^R)\big] - \mathbb{E}_{\bar{z}}\big[f(w^0)\big] \leq \frac{\eta^2(2+u)\,c\,L\,K_b\,(1+c_h) - \eta\sqrt{c}\,\alpha}{K}\sum_{r=0}^{R-1}\big\|\nabla f(w^r)\big\|^2$$

$$- \frac{\eta\sqrt{c}\,\alpha\,\sigma_h}{K}\sum_{r=0}^{R-1}\big\|\nabla f(w^r)\big\| \tag{25}$$

$$+ \frac{\eta^2(2+u)^2\,L^2\,T}{K}\sum_{r=0}^{R-1}\sum_{k \in K_g}\big(f_k(w_k^{0,r}) - f_k^*\big)$$

$$+ \frac{\eta^2(2+u)\,c\,L\,R}{2K}\Big(T^2\,\sigma^2\,K_g + 2\,K_b\,\sigma^2\Big).$$

According to our previous derivation, we know that:

$$\sum_{r=0}^{R-1}\big\|\nabla f(w^r)\big\| \leq \sqrt{R}\sqrt{\sum_{r=0}^{R-1}\big\|\nabla f(w^r)\big\|^2}. \tag{26}$$

Apply Young's inequality with $\delta > 0$ and nonnegative real numbers $x$ and $y$,

$$xy \leq \frac{x^2}{2\delta} + \frac{y^2\delta}{2}.$$

$$\frac{\eta\sqrt{c}\,\sigma_h\alpha}{K}\sum_{r=0}^{R}\big\|\nabla f(w^r)\big\| \leq \frac{\eta\sqrt{c}\,\alpha\sigma_h\sqrt{R}}{K}\sqrt{\sum_{r=0}^{R}\big\|\nabla f(w^r)\big\|^2}$$

$$\leq \frac{1}{2\delta}\sum_{r=0}^{R}\big\|\nabla f(w^r)\big\|^2 + \frac{\eta^2\,c\,\alpha^2\sigma_h^2\,R\,\delta}{2\,K^2}$$

$$-\frac{\eta\sqrt{c}\,\sigma_h\alpha}{K}\sum_{r=0}^{R}\big\|\nabla f(w^r)\big\| \leq \frac{1}{2\delta}\sum_{r=0}^{R}\big\|\nabla f(w^r)\big\|^2 + \frac{\eta^2\,c\,\alpha^2\sigma_h^2\,R\,\delta}{2\,K^2}.$$

We substitute this to the equation 25.

$$\mathbb{E}_{\bar{z}}\big[f(w^R)\big] \; - \; \mathbb{E}_{\bar{z}}\big[f(w^0)\big] \leq \left( \frac{\eta^2(2+u)\,c\,L\,K_b\,(1+c_h) \; - \; \eta\sqrt{c}\,\alpha}{K} \; + \; \frac{1}{2\delta} \right) \sum_{r=0}^{R-1} \|\nabla f(w^r)\|^2$$

$$+ \; \frac{\eta^2\,c\,\alpha^2\sigma_h^2\,R\,\delta}{2\,K^2} + \frac{\eta^2(2+u)^2\,L^2\,T}{K} \sum_{r=0}^{R} \sum_{k\in K_g} \big(f_k(w_k^{0,r}) - f_k^*\big)$$

$$+ \; \frac{\eta^2(2+u)\,c\,L\,R}{2\,K} \Big(T^2\,\sigma^2\,K_g + 2\,K_b\,\sigma_h^2\Big). \tag{27}$$

Given that $w_k^{0,r} = w^r$, this term is equivalent to $\sum_{r=0}^{R} \sum_{k\in K_g} \big(f_k(w^r) - f_k^*\big)$.

From our previous discussion, we have the inequality for a single round $r$:

$$\sum_{k\in K_g} \big(f_k(w^r) - f_k^*\big) \leq \sum_{k=1}^{K} \big(f_k(w^r) - f_k^*\big)$$

and the inequality used in Part 2 of the proof:

$$\sum_{r=0}^{R-1} \sum_{k=1}^{K} \big(f_k(w^r) - f_k^*\big) \leq \sum_{r=0}^{R-1} \left( K f(w^r) - K f^* + \frac{L}{2} \sum_{k=1}^{K} \Delta_k \right).$$

Combining these two inequalities, we obtain a bound for the sum over the set $K_g$:

We set $\gamma \leq 1$ which means that the subset clients the effect to the global:

$$\sum_{r=0}^{R-1} \sum_{k\in K_g} \big(f_k(w^r) - f_k^*\big) \; \leq \; \gamma \sum_{r=0}^{R-1} \Big( K \big(f(w^r) - f^*\big) + \frac{L}{2} \sum_{k=1}^{K} \Delta_k \Big)$$

We substitute this to the above inequality get:

$$\mathbb{E}_{\bar{z}}\big[f(w^R)\big] - \mathbb{E}_{\bar{z}}\big[f(w^0)\big] \leq \left( \frac{\eta^2(2+u)\,c\,L\,K_b\,(1+c_h)}{K} - \frac{\eta\sqrt{c}\,\alpha}{K} + \frac{1}{2\delta} \right) \sum_{r=0}^{R-1} \|\nabla f(w^r)\|^2$$

$$+ \; \frac{\eta^2\,c\,\alpha^2\,\sigma_h^2\,R\,\delta}{2\,K^2} + \eta^2(2+u)^2\,L^2\,T\,\gamma \sum_{r=0}^{R-1} \big(f(w^r) - f^*\big)$$

$$+ \; \frac{\eta^2(2+u)^2\,L^3\,T\,R\,\gamma}{2\,K} \sum_{k=1}^{K} \Delta_k$$

$$+ \; \frac{\eta^2(2+u)\,c\,L\,R}{2\,K} \Big(T^2\,\sigma^2\,K_g + 2\,K_b\,\sigma_h^2\Big).$$

We substitute $\alpha$:

$$\mathbb{E}_{\bar{z}}\big[f(w^R)\big] - \mathbb{E}_{\bar{z}}\big[f(w^0)\big] \leq \left( \frac{\eta^2(2+u)cLK_b(1+c_h)}{K} - \frac{\eta\sqrt{c}(K_gT+K_b)}{K} + \frac{1}{2\delta} \right) \sum_{r=0}^{R-1} \|\nabla f(w^r)\|^2$$

$$+ \; \frac{\eta^2 c(K_g^2T^2 + 2K_gTK_b + K_b^2)\sigma_h^2 R\delta}{2K^2} + \eta^2(2+u)^2L^2T\gamma \sum_{r=0}^{R-1} (f(w^r) - f^*)$$

$$+ \; \frac{\eta^2(2+u)^2L^3TR\gamma}{2K} \sum_{k=1}^{K} \Delta_k + \frac{\eta^2(2+u)cLR}{2K} \Big(T^2\sigma^2 K_g + 2K_b\sigma_h^2\Big).$$

To simplify the inequality, we solve for $\delta$:

$$\frac{1}{2\delta} = -\frac{\eta\sqrt{c}(K_gT + K_b)}{2K} - \frac{\eta^2(2+u)cLK_b(1+c_h)}{K} + \frac{\eta\sqrt{c}(K_gT + K_b)}{K},$$

$$\frac{1}{2\delta} = \frac{\eta\sqrt{c}(K_gT + K_b)}{2K} - \frac{\eta^2(2+u)cLK_b(1+c_h)}{K},$$

$$\delta = \frac{K}{\eta\sqrt{c}(K_gT + K_b) - 2\eta^2(2+u)cLK_b(1+c_h)}.$$

For $\delta > 0$, the denominator must be positive:

$$\eta\sqrt{c}(K_gT + K_b) - 2\eta^2(2+u)cLK_b(1+c_h) > 0,$$

yielding the condition:

$$\eta < \frac{\sqrt{c}(K_gT + K_b)}{2(2+u)cLK_b(1+c_h)}.$$

Substitute $\delta$:

$$\mathbb{E}_{\bar{z}}\big[f(w^R)\big] - \mathbb{E}_{\bar{z}}\big[f(w^0)\big] \leq -\frac{\eta\sqrt{c}(K_gT + K_b)}{2K}\sum_{r=0}^{R-1}\|\nabla f(w^r)\|^2$$
$$+ \frac{\eta^2 c(K_gT + K_b)^2\sigma_h^2 R}{2K\left(\eta\sqrt{c}(K_gT + K_b) - 2\eta^2(2+u)cLK_b(1+c_h)\right)}$$
$$+ \eta^2(2+u)^2L^2T\gamma\sum_{r=0}^{R-1}(f(w^r) - f^*)$$
$$+ \frac{\eta^2(2+u)^2L^3TR\gamma}{2K}\sum_{k=1}^{K}\Delta_k$$
$$+ \frac{\eta^2(2+u)cLR}{2K}\left(T^2\sigma^2 K_g + 2K_b\sigma_h^2\right).$$

According to the Assumption C.2 we have:

$$2\mu(f(\mathbf{w}^r) - f^*) \leq \|\nabla f(\mathbf{w}^r)\|^2, \quad \forall \mathbf{w}^r \in \mathbb{R}^d,$$

$$2\mu\sum_{r=0}^{R-1}(f(\mathbf{w}^r) - f^*) \leq \sum_{r=0}^{R-1}\|\nabla f(\mathbf{w}^r)\|^2, \quad \forall \mathbf{w}^r \in \mathbb{R}^d,$$

Combine the PL inequality to the above function we get:

$$\frac{\eta\sqrt{c}(K_gT + K_b)}{2K}\sum_{r=0}^{R-1}\|\nabla f(w^r)\|^2 \leq \mathbb{E}_{\bar{z}}\big[f(w^0)\big] - \mathbb{E}_{\bar{z}}\big[f(w^R)\big]$$
$$+ \frac{\eta^2 c(K_gT + K_b)^2\sigma_h^2 R}{2K\left(\eta\sqrt{c}(K_gT + K_b) - 2\eta^2(2+u)cLK_b(1+c_h)\right)}$$
$$+ \eta^2(2+u)^2L^2T\gamma\sum_{r=0}^{R-1}(f(w^r) - f^*)$$
$$+ \frac{\eta^2(2+u)^2L^3TR\gamma}{2K}\sum_{k=1}^{K}\Delta_k$$
$$+ \frac{\eta^2(2+u)cLR}{2K}\left(T^2\sigma^2 K_g + 2K_b\sigma_h^2\right).$$

Let $S_E = \sum_{r=0}^{R-1} \mathbb{E}[f(w^r) - f^*]$, $D_\delta = \eta\sqrt{c}(K_g T + K_b) - 2\eta^2(2+u)cLK_b(1+c_h)$. We require $D_\delta > 0$.

Substituting this back into the original inequality:

$$
\begin{aligned}
\mathbb{E}[f(w^R)] - \mathbb{E}[f(w^0)] \leq & -\frac{\eta\mu\sqrt{c}(K_g T + K_b)}{K} S_E + \eta^2(2+u)^2 L^2 T\gamma S_E \\
& + \frac{\eta^2 c(K_g T + K_b)^2 \sigma_h^2 R}{2KD_\delta} \\
& + \frac{\eta^2(2+u)^2 L^3 T R\gamma}{2K} \sum_{k=1}^{K} \Delta_k \\
& + \frac{\eta^2(2+u)cLR}{2K}\left(T^2\sigma^2 K_g + 2K_b\sigma_h^2\right).
\end{aligned}
$$

Collecting terms involving $S_E$:

$$
\mathbb{E}[f(w^R)] - \mathbb{E}[f(w^0)] \leq \left(\eta^2(2+u)^2 L^2 T\gamma - \frac{\eta\mu\sqrt{c}(K_g T + K_b)}{K}\right) S_E + \text{other terms.}
$$

Moving $S_E$ to the left side:

$$
\begin{aligned}
\left(\frac{\eta\mu\sqrt{c}(K_g T + K_b)}{K} - \eta^2(2+u)^2 L^2 T\gamma\right) S_E \leq & \; \mathbb{E}[f(w^0)] - \mathbb{E}[f(w^R)] \\
& + \frac{\eta^2 c(K_g T + K_b)^2 \sigma_h^2 R}{2KD_\delta} + \frac{\eta^2(2+u)^2 L^3 T R\gamma}{2K} \sum_{k=1}^{K} \Delta_k \\
& + \frac{\eta^2(2+u)cLR}{2K}\left(T^2\sigma^2 K_g + 2K_b\sigma_h^2\right). \quad (28)
\end{aligned}
$$

Since $\mathbb{E}[f(w^R)] \geq f^*$ (typically $f^*$ is the minimum), we have $\mathbb{E}[f(w^0)] - \mathbb{E}[f(w^R)] \leq \mathbb{E}[f(w^0)] - f^*$. Let $f_0^* = \mathbb{E}[f(w^0)] - f^*$ (the initial expected suboptimality). Let the coefficient of $S_E$ be $C_S' = \frac{\eta\mu\sqrt{c}(K_g T + K_b)}{K} - \eta^2(2+u)^2 L^2 T\gamma$. To ensure $C_S' > 0$, we need $\eta$ sufficiently small such that $\eta < \frac{\mu\sqrt{c}(K_g T + K_b)}{K(2+u)^2 L^2 T\gamma}$. Then:

$$
\begin{aligned}
C_S' S_E \leq & \; f_0^* + \frac{\eta^2 c(K_g T + K_b)^2 \sigma_h^2 R}{2KD_\delta} + \frac{\eta^2(2+u)^2 L^3 T R\gamma}{2K} \sum_{k=1}^{K} \Delta_k \\
& + \frac{\eta^2(2+u)cLR}{2K}\left(T^2\sigma^2 K_g + 2K_b\sigma_h^2\right).
\end{aligned}
$$

Our goal is $\frac{1}{R} S_E = \frac{1}{R}\sum_{r=0}^{R-1} \mathbb{E}[f(w^r) - f^*]$. Dividing both sides by $R$:

$$
\begin{aligned}
C_S' \frac{1}{R}\sum_{r=0}^{R-1} \mathbb{E}[f(w^r) - f^*] \leq & \; \frac{f_0^*}{R} + \frac{\eta^2 c(K_g T + K_b)^2 \sigma_h^2}{2KD_\delta} \\
& + \frac{\eta^2(2+u)^2 L^3 T\gamma}{2K} \sum_{k=1}^{K} \Delta_k \\
& + \frac{\eta^2(2+u)cL}{2K}\left(T^2\sigma^2 K_g + 2K_b\sigma_h^2\right).
\end{aligned}
$$

Finally, dividing both sides by $C_S'$ (assuming $C_S' > 0$):

$$
\begin{aligned}
\frac{1}{R}\sum_{r=0}^{R-1} \mathbb{E}_{\bar{z}}[f(w^r) - f^*] \leq & \; \frac{1}{C_S'}\left[\frac{f_0^*}{R} + \frac{\eta^2 c(K_g T + K_b)^2 \sigma_h^2}{2K\left(\eta\sqrt{c}(K_g T + K_b) - 2\eta^2(2+u)cLK_b(1+c_h)\right)}\right. \\
& \left. + \frac{\eta^2(2+u)^2 L^3 T\gamma}{2K}\sum_{k=1}^{K}\Delta_k + \frac{\eta^2(2+u)cL}{2K}\left(T^2\sigma^2 K_g + 2K_b\sigma_h^2\right)\right],
\end{aligned}
$$

where

$$C'_S = \frac{\eta\,\mu\sqrt{c}\,(K_g T + K_b)}{K} - \eta^2(2+u)^2 L^2 T\,\gamma, \qquad D_\delta = \eta\sqrt{c}\,(K_g T + K_b) - 2\eta^2(2+u)cL\,K_b(1+c_h).$$

To ensure both $C'_S > 0$ and $D_\delta > 0$, we require

$$\eta < \underbrace{\frac{\sqrt{c}(K_g T + K_b)}{2(2+u)cL\,K_b(1+c_h)}}_{=\bar\eta_\delta}, \qquad \eta < \underbrace{\frac{\mu\sqrt{c}(K_g T + K_b)}{K(2+u)^2 L^2 T\,\gamma}}_{=\bar\eta_S}.$$

Let

$$\eta_{\max} = \min\{\bar\eta_\delta,\ \bar\eta_S\}, \qquad \theta \in \left(0, \tfrac{1}{2}\right].$$

Choosing $\theta = \frac{1}{2}$ gives

$$\eta = \tfrac{1}{2}\,\eta_{\max} = \frac{\mu\sqrt{c}\,(K_g T + K_b)}{2\,K\,(2+u)^2 L^2 T\,\gamma}.$$

We select

$$\eta = \frac{\mu\sqrt{c}(K_g T + K_b)}{2K(2+u)^2 L^2 T\gamma}$$

And from previous client convergence conclusion, we pick a constant local learning rate

$$\eta_{\text{client}} = \frac{c}{\alpha} = \frac{1}{L\,(u+2)} < \frac{2c}{\alpha}$$

Substituting the learning rate $\eta = \min\left\{\frac{1}{L\,(u+2)},\ \frac{\mu\,\sqrt{c}\,(K_g\,T + K_b)}{2\,K\,(2+u)^2\,L^2\,T\,\gamma}\right\}$, since $\eta$ is a small value, we neglect $\eta^2$.

$$\frac{1}{R}\sum_{r=0}^{R-1}\mathbb{E}_{\bar z}\big[f(w^r) - f^*\big] \leq \frac{4K^2(2+u)^2 L^2 T\gamma\,\mathbb{E}[f(w^0) - f^*]}{\mu^2\,c\,(K_g T + K_b)^2\,R} + \frac{\sigma_h^2}{2} + \frac{(2+u)\,L}{4K}\sum_{k=1}^{K}\Delta_k$$

$$+ \frac{c}{4K(2+u)L\,T\,\gamma}\Big(T^2\sigma^2 K_g + 2K_b\sigma_h^2\Big).$$

$$\frac{1}{R}\sum_{r=0}^{R-1}\mathbb{E}_{\bar z}\big[f(w^r) - f^*\big] \leq O\!\left(\frac{K^2\,(2+u)^2\,\gamma\,T}{c\,(K_g T + K_b)^2\,R}\right)$$

$$+ O\!\left(\frac{1+u}{K}\Big(\sum_{k=1}^{K_g}\Delta_{kg} + \sum_{k=1}^{K_b}\Delta_{kb}\Big)\right) \tag{29}$$

$$+ O\!\left(\frac{c\,T\,K_g}{K(1+u)\,\gamma}\right)$$

$$+ O\!\left(\frac{c\,K_b\,\sigma_h^2}{K(1+u)\,T\,\gamma}\right) + O(1).$$

$\square$

Define the error upper-bounds for MEERKAT-VP and the baseline MEERKAT as follows:

$$E_{\text{MEERKAT-VP}} = \underbrace{\frac{4K^2(2+u)^2 L^2 T\gamma}{\mu^2\,c\,(K_g T + K_b)^2}\,\frac{\mathbb{E}[f(w^0) - f^*]}{R}}_{\text{(I) Transient term}} + \underbrace{\left[\frac{\sigma_h^2}{2} + \frac{(2+u)L}{4K}\sum_{k=1}^{K}\Delta_k + \frac{c\big(T^2\sigma^2 K_g + 2K_b\sigma_h^2\big)}{4K(2+u)L\,T\,\gamma}\right]}_{\text{(II) Steady-state term}},$$

$$E_{\text{MEERKAT}} = \underbrace{\frac{4L^2(2+u)^2}{\mu^2\,c\,(1+\sqrt{c_h})^2\,T}\,\frac{\mathbb{E}[f(w^0) - f^*]}{R}}_{\text{(I') Transient term}} + \underbrace{\left[\frac{\sigma_h^2}{\mu\,(1+\sqrt{c_h})^2} + \frac{L}{K}\sum_{k=1}^{K}\Delta_k + \frac{T\,c\,\sigma^2}{2L\,(2+u)}\right]}_{\text{(II') Steady-state term}}.$$

- **Transient term ratio:**

$$\frac{(I)}{(I')} \approx \gamma\left(1 + \sqrt{c_h}\right)^2 < 1, \quad \text{and as } c_h \to 1, \ \gamma(1 + \sqrt{c_h})^2 \to 0.$$

- **Noise term ratio:**

$$\frac{\sigma_h^2/2}{\sigma_h^2/(\mu\left(1 + \sqrt{c_h}\right)^2)} = \frac{\mu\left(1 + \sqrt{c_h}\right)^2}{2}, \quad \text{which is } < 1 \text{ when } \mu\left(1 + \sqrt{c_h}\right)^2 < 2.$$

Empirically $\mu < 1$, thus $\mu\left(1 + \sqrt{c_h}\right)^2 < 2$ is True. Additionally, VPCS includes an extra term $\frac{cK_b\sigma_h^2}{2K(2+u)LT\gamma}$, which decays as $\frac{1}{T}$ and becomes negligible for large $T$.

- **Heterogeneity and variance terms:**

$$\frac{(2+u)L}{4K}\sum_{k=1}^{K}\Delta_k < \frac{L}{K}\sum_{k=1}^{K}\Delta_k, \quad \text{and the extra variance term decays as } 1/K.$$

Therefore, under the same $T$ and $R$, $E_{\text{MEERKAT-VP}} < E_{\text{MEERKAT}}$ and this gap widens as data heterogeneity $c_h$ increases.

REMARKS

The analysis of the upper bound in Equation 17 reveals how the local training step $T$, density level $u$, and communication rounds $R$ collectively influence the optimization dynamics through a balance of convergence rate, bias–variance trade-offs, and steady-state error control:

- **Impact of Local Update Steps $T$:** A smaller $T$ amplifies the term $\mathcal{O}\left(\frac{(2+u)^2}{TR} \cdot \mathbb{E}[f(w^0) - f(w^R)]\right)$, increasing the average optimality gap after $R$ communication rounds when $R$ is fixed. However, this effect can be mitigated by increasing $R$, as the scaling factor $\frac{1}{R}$ reduces the term's impact. Conversely, reducing $T$ diminishes the variance term $\mathcal{O}\left(\frac{T}{2+u}\right)$, leading to a smaller steady-state error. Thus, a smaller $T$ may prolong the transient phase but ultimately achieves a tighter optimality gap relative to $f^*$ after sufficient rounds.
- **Density Level $u$.** Reducing $u$ (i.e., increasing sparsity) quadratically benefits the transient term, yet it also inflates the steady-state term through the denominator $2 + u$. Choosing $u$ therefore amounts to balancing communication savings against the plateau error; aggressive sparsification should be coupled with smaller $T$ to avoid performance degradation.
- **MEERKAT-VP Client Selection Strategy:** By early-stopping extreme data-imbalance clients with a single local training step, MEERKAT-VP effectively reduces Non-IID drift in zeroth-order federated llm fine-tuning. This strategy lowers the coefficient of the transient term and further reduces heterogeneity- and variance-induced steady-state error. Under fixed $T$ and $R$, these effects yield strictly faster convergence and a tighter optimality gap in Non-IID settings.

These conclusions illustrate how tuning $T$, $R$, $u$, and the MEERKAT-VP client selection strategy can optimize performance in federated, sparse, and Non-IID learning scenarios.

C.6 EMPIRICAL ANALYSIS OF THE GRADIP PHENOMENON

By Lemma C.8, the masked sparse zeroth-order (ZO) surrogate gradient is an *unbiased* estimator of the masked first-order gradient. Building on this fact, we define the vector $g_c(w; x, y)$ is obtained by computing the gradient of the cross-entropy loss for a single sample with respect to a small subset of parameters selected by a mask.

From logits to Softmax Probabilities we have:

- The model's final layer outputs a *logit* for each class:

$$h(x; w) = (h_1, \ldots, h_C) \in \mathbb{R}^C.$$

- The softmax probabilities are given by:

$$p_j(x; w) = \frac{e^{h_j}}{\sum_{r=1}^{C} e^{h_r}}.$$

The cross-entropy loss for a single sample is:

$$\ell(w; x, y) = -\log p_y(x; w), \quad \text{where } y \in \{1, \ldots, C\}.$$

For each logit $h_j$, the partial derivative is:

$$\frac{\partial \ell}{\partial h_j} = p_j - \mathbf{1}_{\{y=j\}} = p_j - (e_y)_j,$$

where $e_y$ is the one-hot vector with 1 in the $y$-th component.

Since we are only interested in the sensitive parameters selected by the mask $m$, the gradient with respect to the parameters can be written as:

$$g_c(w; x, y) = \nabla_{w_m} \ell(w; x, y)$$
$$= \sum_{j=1}^{C} \frac{\partial \ell}{\partial h_j} \nabla_{w_m} h_j(x; w)$$
$$= \left( p(x; w) - e_y \right)^\top \nabla_{w_m} h(x; w).$$

Here:

- $\nabla_{w_m} h_j(x; w)$ is the gradient/Jacobian of the logit $h_j$ with respect to the masked parameter $w_m$.
- By collecting the coefficients $p_j - \mathbf{1}_{y=j}$ into a vector, we obtain the compact form:

$$g_c(w; x, y) = (p - e_y)^\top \nabla_{w_m} h(x; w).$$

In our existing local client convergence inequality and from the assumption C.4, we can empirically write the key constant estimator variance:

$$\sigma_k^2 = \frac{1}{d} \text{Var}_{(x,y) \sim D_k}[g_c(w; x, y)].$$

We write $g_c$ in matrix form: Define:

$$\mathbf{J}(x; w) = \nabla_{w_m} h(x; w) \in \mathbb{R}^{d_m \times C}, \quad \mathbf{a}(x, y; w) = p(x; w) - \mathbf{e}_y \in \mathbb{R}^C.$$

Thus:

$$g_c(w; x, y) = \mathbf{J}^\top(x; w) \, \mathbf{a}(x, y; w) \in \mathbb{R}^{d_m}.$$

We substitute this equation to the above estimator variance:

$$\sigma_k^2 = \frac{1}{d_m} \underbrace{\mathbb{E}_{(x,y)} \big\| g(w; x, y) - \nabla f_k(w) \big\|^2}_{\text{total variance}} = \frac{1}{d_m} \text{tr} \left( \mathbf{J}^\top \underbrace{\text{Cov}_{(x,y)} \big[ \mathbf{a}(x, y; w) \big]}_{\Sigma_a} \mathbf{J} \right). \quad (1)$$

Note:

- $\Sigma_a \in \mathbb{R}^{C \times C}$ is determined solely by the **label distribution and prediction probabilities**.
- $\mathbf{J}$ reflects the network structure and influences only a similarity coefficient.

**Analysis of Extreme Non-IID (Single Label $y^\dagger$):**

- The label is fixed, so $\mathbf{1}_{y=j}$ is constant.
- If the model is mostly correct: $p \approx \mathbf{e}_{y^\dagger}$, then $\mathbf{a}(x, y; w) \approx \mathbf{0}$, yielding:

$$\Sigma_a \approx \mathbf{0} \implies \sigma_{\text{non}}^2 \approx \frac{1}{d_m} \text{tr}(\mathbf{0}) = 0.$$

**Analysis of Approximate IID (Balanced Multi-Label)**

- The label $y$ varies across $\{1, \ldots, C\}$.

- Even as the loss decreases, $p_j$ differs across classes. The covariance is:

$$(\Sigma_a)_{rs} = \mathbb{E}\left[(p_r - \mathbf{1}_{y=r})(p_s - \mathbf{1}_{y=s})\right] - \underbrace{(\mathbb{E}[p_r - \mathbf{1}_{y=r}])}_{=0}\underbrace{(\mathbb{E}[p_s - \mathbf{1}_{y=s}])}_{=0}.$$

This matrix has diagonal elements $\mathbb{E}\left[(p_r - \mathbf{1}_{y=r})^2\right] > 0$, making $\Sigma_a$ positive definite or semi-definite but non-zero. Thus:

$$\sigma_{\text{iid}}^2 = \frac{1}{d_m}\,\text{tr}\left(\mathbf{J}^\top \Sigma_a \mathbf{J}\right) > 0.$$

Our local convergence bound is:

$$\frac{1}{T}\sum_{t=0}^{T-1}\mathbb{E}\|\nabla f_k(w_k^t)\|^2 \le O\left(\frac{1}{T}\right) + \sigma_k^2,$$

which indicates that in the steady state, the upper bound of the gradient norm is determined by $\sigma_k^2$. Therefore,

$$\sigma_{\text{iid}}^2 \gg \sigma_{\text{non-iid}}^2 \approx 0 \quad \implies \quad \begin{cases} \text{IID clients: Gradient Norm oscillates significantly;} \\ \text{Non-IID clients: Gradient Norm decreases monotonically and approaches 0.} \end{cases}$$

REMARKS

In summary, by substituting the explicit form of the cross-entropy gradient into our sparse ZO convergence formula, we can empirically explain that due to the variance differences caused by label distributions, the Gradient Norms of IID clients maintain significant fluctuations, while that of extremely Non-IID clients rapidly decays and converges to zero.

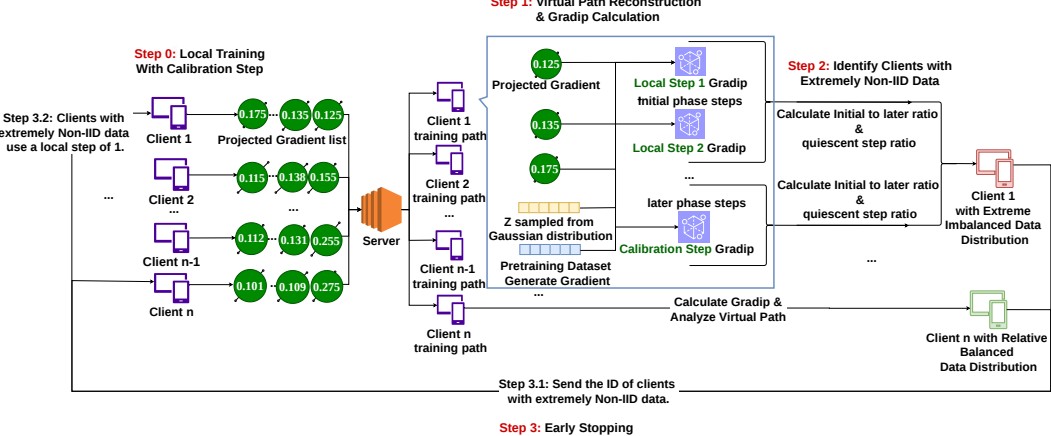

Figure 5: MEERKAT-VP: Each client locally trains with a prescribed number of logging steps, yielding a sequence of projected gradients. The server leverages a randomly sampled vector $z_k^t$ from the Gaussian distribution $\mathcal{N}(\mathbf{0}, \mathbf{I}_d)$ to reconstruct $\nabla f_k^t$, and then computes GradIP (see Definition 2.3) at every local training step. By analyzing the resulting GradIP values across all clients, the system distinguishes those clients with extremely Non-IID data from those that are relatively balanced. For the definition of parameters **later phase steps**, **initial phase steps**, **quiescent step ratio**, and **initial to later ratio**, please refer to Table 3 in Appendix D.1

## D   MORE EXPERIMENTAL DETAILS

### D.1   ADDITIONAL EXPERIMENTAL SETTINGS

**Testbed.** All experiments are run on servers with the following configurations: RTX A6000 Setup: Ubuntu 18.04.6 LTS with 2 NVIDIA RTX A6000 GPUs (each with 48GB GPU memory). GH200

---

**Algorithm 2** MEERKAT: Sparse Zeroth-Order Optimization for Federated LLM Fine-Tuning

---

**Input:** pre-trained weight $\mathbf{w}_0$, sparse mask $\mathbf{m}$, learning rate $\eta$, perturbation scale $\epsilon$, number of rounds $R$, total number of *clients* $K$ number of local steps $T$

*Server* initiates seed list $\{s_1^1, \cdots, s_1^T\}$

**for** Round $r = 1$ to $R$ **do**
   **Step 1. Local ZO update.**
   **for** each *client* $k$ **in parallel do**
      Download model from *server*: $\mathbf{w}_k \leftarrow \mathbf{w}_{r-1}$
      Download seed list $\{s_r^1, \cdots, s_r^T\}$ from *server*
      **for** local step $t = 1$ to $T$ **do**
         Initialize $\mathbf{z}_k^t$ with seed $s_r^t$.
         Sample a batch $\mathcal{B}$ on *client* dataset.
         $\tilde{\mathbf{w}}_k^t \leftarrow \mathbf{w}_k^t + \epsilon \cdot (\mathbf{z}_k^t \odot \mathbf{m})$
         Compute loss $f_+ \leftarrow f(\tilde{\mathbf{w}}_k^t; \mathcal{B})$
         $\tilde{\mathbf{w}}_k^t \leftarrow \mathbf{w}_k^t - 2\epsilon \cdot (\mathbf{z}_k^t \odot \mathbf{m})$
         Compute loss: $f_- \leftarrow f(\tilde{\mathbf{w}}_k^t; \mathcal{B})$
         Compute projected gradient:

$$g_k^t \leftarrow (f_+ - f_-)/2\epsilon$$

         Update *client* model:

$$\hat{\nabla} f_k^t \leftarrow g_k^t \cdot (\mathbf{z}_k^t \odot \mathbf{m})$$
$$\mathbf{w}_k^{t+1} \leftarrow \mathbf{w}_k^t - \eta \hat{\nabla} f_k^t$$

      **end for**
      Send projected gradients $\{g_k^1, g_k^2, \ldots, g_k^T\}$ to *server*.
   **end for**
   **Step 2.** *Server* **recovers each** *client's* **update with** *virtual path*.
   **for** $k = 1$ to $K$ **do**
      **for** local step $t = 1$ to $T$ **do**
         Generate $\mathbf{z}_k^t$ with seed $s_r^t$.
         Perform *virtual path*:

$$\hat{\nabla} f_k^t = g_k^t \cdot (\mathbf{z}_k^t \odot \mathbf{m})$$
$$\mathbf{w}_k^{t+1} \leftarrow \mathbf{w}_k^t - \eta \hat{\nabla} f_k^t$$

      Store recovered *client* model parameters $\mathbf{w}_k^T$
      **end for**
   **end for**
   **Step 3.** *Server* **aggregates reconstructed sparse model update.**

$$\mathbf{w}_r \leftarrow \frac{1}{K} \sum_{k=1}^{K} \mathbf{w}_k^T$$

   Generate new seed list $\{s_{r+1}^1, \cdots, s_{r+1}^T\}$.
**end for**
**Output:** $\mathbf{w}_R$

---

---

**Algorithm 3** MEERKAT with high frequency server-client synchronization

---

**Input:** Seed $s$ and projected gradients $g_k^t$ from all clients, learning rate $\eta$, number of clients $K$, sparse mask $\mathbf{m}$

**Aggregate projected gradients from all clients with same seed:**

$$g \leftarrow \frac{1}{K} \sum_{k=1}^{K} g_k$$

**Calculate Zeroth-Order Gradients:**

$$\hat{\nabla} f \leftarrow g \cdot (\mathbf{z} \odot \mathbf{m})$$

**Update global model parameters:**

$$\mathbf{w_{r+1}} \leftarrow \mathbf{w_r} - \eta \left( \hat{\nabla} f \odot \mathbf{m} \right)$$

**Generate new seed** $s\_new$
**Output: Send aggregated global projected gradients $g$ and seed $s\_new$ to all clients.**

---

Setup: Ubuntu 20.04 with 1 NVIDIA GH200 GPU (96GB GPU memory). A100 Setup: Ubuntu 22.04 with 1 NVIDIA A100 GPU (40GB GPU memory).

**Dataset.** We conducted experiments using datasets from the GLUE and SuperGLUE benchmarks, including SST2, AgNews, Yelp, BoolQ, RTE, WSC, and WiC. To create IID client data, we shuffle the entire dataset and evenly divide it among the clients. To create Non-IID client data, we split the data using a Dirichlet distribution. For all tasks, the Dirichlet $\alpha$ parameter is set to $0.5$ to control the degree of data heterogeneity.

**Evaluation metric.** In our experiments, test accuracy is used as the primary evaluation metric. Accuracy is computed as the proportion of correctly predicted labels across all evaluation samples. Additionally, we incorporate the GradIP score (see Definition 2.3) to analyze further the dynamics of local model training under IID and Non-IID client data settings. GradIP provides a metric to measure the quality of client training trajectories, particularly in heterogeneous data distributions.

**Notations.** We present the definitions of parameters used in MEERKAT-VP in Table 3.

Table 3: MEERKAT-VP Parameters Notation

| Term | Explanation |
|---|---|
| **calibration steps** $T_{\text{cali}}$ | Number of steps each client performs to measure GradIP. |
| **initial phase steps** $T_{\text{init}}$ | Number of earliest local steps used to measure the early-phase GradIP. |
| **later phase steps** $T_{\text{later}}$ | Number of latest local steps used to observe the late-phase GradIP. |
| **convergence threshold** $\sigma$ | Threshold indicating when GradIP is effectively zero. |
| **quiescent step ratio** $\rho_{\text{quie}}$ | Fraction of later phase where GradIP stays below threshold |
| **Initial to later ratio** $\rho_{\text{later}}$ | Ratio of average GradIP in the initial phase to average GradIP in the later phase. |

**Hyper-parameters.** We use the following hyper-parameters in our experiments; see Table 4

**MEERKAT-VP Hyperparameter Selection.** Below, we present the default hyperparameter values for MEERKAT-VP, task-specific adjustments, and the results of our hyperparameter sensitivity analysis to demonstrate the robustness of the method.

These default values work well for most tasks. However, some tasks benefit from task-specific tuning, particularly the RTE task which shows sensitivity to these parameters:

**Sensitivity Analysis.** We conducted sensitivity analysis to understand the robustness of our method to hyperparameter variations. Tables 7 and 8 show the performance stability across different parameter settings.

Table 4: Hyper-parameters used in our experiments.

| Parameter | Value |
|---|---|
| MEERKAT learning rate | [2e-4, 2e-8] |
| MEERKAT-VP learning rate | [2e-4, 2e-8] |
| LoRA-FedZO learning rate | [2e-4, 2e-8] |
| Full-FedZO learning rate | [2e-4, 2e-8] |
| Batch size | 16 |
| Dirichlet alpha | 0.5, 0.3, 0.1 |
| LoRA rank | 16 |
| LoRA alpha | 16 |
| initial phase steps | 20 |
| later phase steps | 20 |
| convergence threshold | 1 |
| quiescent step ratio | [0.4, 0.5, 0.7] |
| Initial to later ratio | [1.5, 2, 5, 10, 15] |
| calibration steps | 100 |
| Total clients | 10 |

Table 5: Default MEERKAT-VP Hyperparameter Values

| initial phase steps | later phase steps | convergence threshold | quiescent step ratio | Initial to later ratio |
|---|---|---|---|---|
| 20 | 20 | 1 | 0.5 | 5 |

Table 6: Task-Specific VPCS Hyperparameters for RTE Task

| Model | initial phase steps | later phase steps | convergence threshold | quiescent step ratio | Initial to later ratio |
|---|---|---|---|---|---|
| Gemma2-2B | 20 | 20 | 1 | 0.7 | 5 |
| LLaMA-3.2-1B | 20 | 20 | 0.5 | 0.7 | 5 |
| Qwen2-1.5B | 20 | 20 | 0.5 | 0.5 | 5 |

Table 7: Parameter Sensitivity Analysis for LLaMA-3.2-1B on SST-2 Task

| initial phase steps | later phase steps | convergence threshold | quiescent step ratio | Initial to later ratio | Performance |
|---|---|---|---|---|---|
| 20 | 20 | 1 | 0.5 | 3 | 0.922 |
| 20 | 20 | 1 | 0.5 | 5 | 0.922 |
| 20 | 20 | 1 | 0.5 | 7 | 0.922 |
| 20 | 20 | 1 | 0.5 | 10 | 0.922 |
| 20 | 20 | 1 | 0.5 | 12 | 0.922 |

Table 8: Parameter Sensitivity Analysis for RTE Task

| Model | initial phase steps | later phase steps | convergence threshold | quiescent step ratio | Initial to later ratio | Performance |
|---|---|---|---|---|---|---|
| LLaMA-3.2-1B | 20 | 20 | 0.8 | 0.5 | 7 | 0.617 |
| | 20 | 20 | 0.7 | 0.5 | 5 | 0.617 |
| Gemma2-2B | 20 | 20 | 1 | 0.5 | 3 | 0.657 |
| | 20 | 20 | 1 | 0.5 | 5 | 0.657 |

## D.2 ADDITIONAL EXPERIMENT RESULTS

In this section, we present additional experimental results to compare MEERKAT, MEERKAT-VP, Full-FedZO, and LoRA-FedZO under various settings. The results include multiple tables and figures, providing a detailed evaluation of performance across different models, datasets and experiment settings. Table 3 provides a description of the parameters used in MEERKAT-VP, and Table 4 lists the experiment parameters used in this experiment. Tables 5 and 6 list the hyperparameter values for MEERKAT-VP. Tables 7 and 8 demonstrate the robustness of the MEERKAT-VP parameter selection. Table 9 provides a quantitative analysis that demonstrates the significant disparity in gradient sensitivity across different parameter groups, thereby justifying our selection criteria. Table 11 shows that a domain-shifted calibration dataset can be used effectively to select sensitive model parameters. Furthermore, we designed an experiment where each client builds a local parameter

mask from its own dataset. The results demonstrate that aggregating these local masks into a union mask does not achieve better performance than using a single, globally unified mask. Table 13 compares MEERKAT and Full-FedZO on multiple tasks at the same communication frequency for Llama-3.2-1B, Qwen2-1.5B, and Gemma-2-2b models. Table 14 presents results in a Non-IID client data scenario, comparing MEERKAT-VP and MEERKAT under the same communication frequency and sparsity density, and demonstrating MEERKAT-VP improved performance. Table 15 investigates the robustness of MEERKAT by evaluating test accuracy with local step 1 across different sparsity densities. Table 16 compares MEERKAT, Full-FedZO and LoRA-FedZO under high communication frequency across IID and Non-IID client data settings. Table 23 details the number of training rounds required for convergence across different models and tasks. Table 24 benchmarks computational and communication efficiency, demonstrating that MEERKAT significantly reduces peak RAM usage and client download bandwidth compared to the Full-FedZO and LoRA-FedZO baselines. Table 25 shows that our MEERKAT-VP method achieves competitive performance against the back-propagation upper bound and substantially outperforms FedDYN (Acar et al., 2021). Figure 7 and Figure 9 further illustrate the phenomenon of GradIP under IID and Non-IID client data settings.

Table 9: Gradient Sensitivity Analysis for Qwen2-1.5B Model on C4 Dataset (Top 0.1% Parameters). To quantitatively analyze gradient sensitivity, we ranked all parameters by their average squared gradients from pre-training and divided them into four disjoint (non-overlapping) buckets: 0-0.1%, 0.1-1%, 1-10% and 10%-100%.

| Bucket / Metric | Top 0.1% | 0.1%-1% | 1%-10% | 10%-100% |
|---|---|---|---|---|
| Avg Gradient Square | $4.403 \times 10^{-3}$ | $8.536 \times 10^{-5}$ | $1.075 \times 10^{-5}$ | $1.764 \times 10^{-6}$ |
| Std Gradient Square | $8.094 \times 10^{-2}$ | $5.858 \times 10^{-5}$ | $6.255 \times 10^{-6}$ | $1.099 \times 10^{-6}$ |
| Max Gradient Square | $1.413 \times 10^{1}$ | $3.147 \times 10^{-4}$ | $3.505 \times 10^{-5}$ | $5.245 \times 10^{-6}$ |
| Min Gradient Square | $3.166 \times 10^{-4}$ | $3.529 \times 10^{-5}$ | $5.245 \times 10^{-6}$ | $1.025 \times 10^{-19}$ |

Table 10: Accuracy of MEERKAT vs. Random-Select (Qwen2-1.5B, 0.1% mask). Directly addressing the comparison with random selection, we ran a control experiment that shows our method is significantly better across all tasks. The local step is 10.

| Method | SST-2 | AGNews | Yelp | BoolQ | RTE | WSC | WIC | Avg |
|---|---|---|---|---|---|---|---|---|
| MEERKAT | **0.949** | **0.881** | **0.934** | **0.752** | **0.813** | **0.682** | **0.628** | **0.806** |
| Random Select | 0.821 | 0.543 | 0.852 | 0.667 | 0.711 | 0.663 | 0.539 | 0.685 |
| **Improvement** | +12.8% | +33.8% | +8.2% | +8.5% | +10.2% | +1.9% | +8.9% | +12.1% |

Table 11: Performance Comparison with Different Calibration Datasets and Methods. Our method does not require the original pre-training data. It uses a small sample (128 sequences) from any public, high-quality text corpus to create a transferable parameter mask. This table confirms MEERKAT's flexibility and transferability across different domains, including web-text, code, and medical data, consistently outperforming the Full-FedZO baseline. We also explore UnionMask, a client-specific mask aggregation approach: (1) Each client computes its own mask based on local data distribution; (2) Clients send masks to the server for aggregation into a union mask; (3) All clients use this union mask for ZO training; (4) The server uses the union mask for parameter updates. Results show that the specialized UnionMask performs similarly to our transferable mask, validating our universality approach. The local step is 10. Code data: microsoft/rStar-Coder. Medical data: FreedomIntelligence/medical-o1-reasoning-SFT.

| Method | SST-2 | AGNews | Yelp | BoolQ | RTE | WSC | WIC | Avg |
|---|---|---|---|---|---|---|---|---|
| Full-FedZO | 0.909 | 0.705 | 0.940 | 0.641 | 0.542 | 0.634 | 0.523 | 0.699 |
| *Web-Text Domain Calibration Data* | | | | | | | | |
| MEERKAT (C4, 0.1%) | **0.916** | **0.872** | **0.964** | 0.695 | **0.600** | **0.653** | **0.614** | **0.759** |
| MEERKAT (Wiki, 0.1%) | 0.913 | 0.855 | 0.952 | 0.646 | 0.582 | 0.634 | 0.567 | 0.736 |
| MEERKAT (ArXiv, 0.1%) | 0.901 | 0.851 | 0.949 | **0.714** | 0.573 | 0.644 | 0.562 | 0.742 |
| MEERKAT (FineWeb, 0.1%) | 0.902 | 0.846 | 0.958 | 0.695 | 0.584 | 0.634 | 0.561 | 0.740 |
| *Domain-Shifted Calibration Data* | | | | | | | | |
| MEERKAT (Code, 0.1%) | 0.915 | 0.843 | 0.956 | 0.695 | 0.551 | 0.612 | 0.602 | 0.739 |
| MEERKAT (Bio, 0.1%) | 0.912 | 0.850 | 0.956 | 0.694 | 0.560 | 0.625 | 0.595 | 0.742 |
| *Client-Specific Mask Aggregation* | | | | | | | | |
| UnionMask (per-client, C4, 0.1%) | 0.902 | 0.845 | 0.950 | 0.669 | 0.582 | 0.634 | 0.569 | 0.736 |

Table 12: Transferability of the sparse mask between legal-domain (MultiEURLEX) (Chalkidis et al., 2021) calibration datasets on LLaMA-3.2-1B.

| Mask domain | SST2 | AgNews | Yelp | BoolQ |
|---|---|---|---|---|
| Legal-domain (MultiEURLEX) | 0.912 | 0.845 | 0.948 | 0.703 |

Table 13: Performance comparison of MEERKAT and Full-FedZO on tasks SST-2, AgNews, Yelp, BoolQ, RTE, WSC, WIC under an IID client data setting. "Acc" is the average test accuracy across tasks. Bold numbers indicate the highest value in each row.

| | Methods | Local Step | SST-2 | AgNews | Yelp | BoolQ | RTE | WSC | WIC | Acc |
|---|---|---|---|---|---|---|---|---|---|---|
| **LLaMA-3.2-1B** | Full-FedZO | 10 | 0.913 | 0.700 | 0.938 | 0.646 | 0.537 | 0.634 | 0.540 | 0.701 |
| | MEERKAT | 10 | **0.925** | **0.881** | **0.964** | **0.751** | **0.684** | 0.634 | **0.648** | **0.784** |
| | Full-FedZO | 30 | 0.913 | 0.700 | 0.935 | 0.643 | 0.542 | 0.634 | 0.528 | 0.699 |
| | MEERKAT | 30 | **0.919** | **0.865** | **0.967** | **0.729** | **0.644** | **0.663** | **0.617** | **0.772** |
| | Full-FedZO | 50 | 0.913 | 0.698 | 0.939 | 0.641 | 0.520 | 0.634 | 0.539 | 0.698 |
| | MEERKAT | 50 | **0.920** | **0.871** | **0.966** | **0.734** | **0.648** | **0.653** | **0.614** | **0.772** |
| | Full-FedZO | 100 | 0.903 | 0.705 | 0.934 | 0.656 | 0.537 | 0.634 | 0.537 | 0.701 |
| | MEERKAT | 100 | **0.913** | **0.842** | **0.945** | **0.722** | **0.573** | 0.634 | **0.595** | **0.746** |
| **Qwen2-1.5b** | Full-FedZO | 10 | 0.891 | 0.701 | 0.931 | 0.696 | 0.800 | 0.682 | 0.579 | 0.754 |
| | MEERKAT | 10 | **0.944** | **0.889** | **0.942** | **0.788** | **0.817** | **0.700** | **0.656** | **0.819** |
| | Full-FedZO | 30 | 0.902 | 0.702 | 0.930 | 0.709 | 0.817 | 0.663 | 0.583 | 0.758 |
| | MEERKAT | 30 | **0.942** | **0.895** | **0.940** | **0.786** | **0.840** | **0.710** | **0.659** | **0.825** |
| | Full-FedZO | 50 | 0.902 | 0.705 | 0.929 | 0.701 | 0.808 | **0.663** | 0.590 | 0.757 |
| | MEERKAT | 50 | **0.942** | **0.885** | **0.934** | **0.784** | **0.840** | 0.634 | **0.637** | **0.808** |
| | Full-FedZO | 100 | 0.899 | 0.714 | 0.928 | 0.705 | **0.831** | **0.682** | 0.594 | 0.765 |
| | MEERKAT | 100 | **0.946** | **0.886** | **0.930** | **0.776** | 0.804 | 0.653 | **0.653** | **0.807** |
| **Gemma2-2b** | Full-FedZO | 10 | 0.87 | 0.732 | 0.944 | 0.717 | 0.564 | 0.634 | 0.592 | 0.723 |
| | MEERKAT | 10 | **0.943** | **0.892** | **0.97** | **0.817** | **0.724** | **0.653** | **0.636** | **0.805** |
| | Full-FedZO | 30 | 0.91 | 0.81 | 0.942 | 0.73 | 0.56 | 0.644 | 0.578 | 0.739 |
| | MEERKAT | 30 | **0.943** | **0.887** | **0.973** | **0.812** | **0.617** | **0.663** | **0.608** | **0.786** |
| | Full-FedZO | 50 | 0.911 | 0.812 | 0.942 | 0.735 | 0.551 | 0.634 | 0.572 | 0.737 |
| | MEERKAT | 50 | **0.94** | **0.873** | **0.964** | **0.812** | **0.604** | 0.634 | **0.617** | **0.778** |
| | Full-FedZO | 100 | 0.917 | 0.83 | 0.936 | 0.728 | 0.56 | **0.644** | 0.59 | 0.744 |
| | MEERKAT | 100 | **0.949** | **0.87** | **0.954** | **0.815** | **0.568** | 0.634 | **0.592** | **0.769** |

Table 14: Comparison of MEERKAT-VP and MEERKAT under Non-IID client data setting, with the same local step and sparsity. Tasks include SST-2, AgNews, Yelp, BoolQ, RTE, WSC, and WIC. "Acc" indicates the average test accuracy across all tasks. Bold numbers highlight the best result in each row.

| | Methods | Local Step | SST-2 | AgNews | Yelp | BoolQ | RTE | WSC | WIC | Acc |
|---|---|---|---|---|---|---|---|---|---|---|
| | MEERKAT-VP | 10 | **0.922** | 0.864 | 0.962 | **0.713** | **0.617** | 0.644 | 0.625 | **0.764** |
| | MEERKAT | 10 | 0.916 | **0.872** | **0.964** | 0.695 | 0.600 | **0.653** | **0.614** | 0.759 |
| | MEERKAT-VP | 30 | **0.919** | 0.825 | 0.963 | **0.685** | **0.595** | 0.634 | **0.631** | **0.750** |
| LLaMA-3.2-1B | MEERKAT | 30 | 0.897 | **0.862** | **0.965** | 0.646 | 0.577 | **0.644** | 0.583 | 0.739 |
| | MEERKAT-VP | 50 | 0.909 | **0.836** | 0.959 | **0.691** | 0.577 | 0.615 | **0.615** | **0.743** |
| | MEERKAT | 50 | 0.909 | 0.827 | **0.965** | 0.647 | **0.595** | **0.634** | 0.567 | 0.734 |
| | MEERKAT-VP | 100 | **0.904** | **0.824** | 0.962 | **0.684** | 0.577 | **0.653** | **0.630** | **0.747** |
| | MEERKAT | 100 | 0.896 | 0.777 | 0.961 | 0.658 | 0.577 | 0.644 | 0.573 | 0.726 |
| | MEERKAT-VP | 10 | 0.941 | **0.886** | **0.947** | 0.76 | **0.822** | 0.653 | **0.636** | **0.806** |
| | MEERKAT | 10 | **0.949** | 0.881 | 0.934 | 0.752 | 0.813 | **0.682** | 0.628 | 0.805 |
| | MEERKAT-VP | 30 | 0.935 | 0.876 | **0.953** | **0.759** | **0.822** | 0.653 | **0.626** | **0.803** |
| Qwen2-1.5b | MEERKAT | 30 | **0.944** | **0.878** | 0.928 | 0.734 | 0.800 | **0.663** | 0.624 | 0.795 |
| | MEERKAT-VP | 50 | 0.931 | **0.882** | **0.946** | **0.754** | **0.804** | 0.644 | **0.63** | **0.798** |
| | MEERKAT | 50 | **0.948** | 0.872 | 0.926 | 0.746 | 0.795 | **0.663** | 0.594 | 0.792 |
| | MEERKAT-VP | 100 | 0.935 | 0.874 | **0.947** | **0.751** | **0.817** | 0.653 | **0.644** | **0.803** |
| | MEERKAT | 100 | **0.936** | **0.878** | 0.925 | 0.741 | 0.795 | **0.663** | 0.61 | 0.792 |
| | MEERKAT-VP | 10 | **0.948** | **0.873** | **0.971** | 0.802 | **0.657** | **0.663** | 0.609 | **0.789** |
| | MEERKAT | 10 | 0.939 | 0.869 | 0.96 | **0.804** | 0.591 | 0.634 | 0.609 | 0.772 |
| | MEERKAT-VP | 30 | **0.948** | **0.86** | **0.974** | **0.799** | **0.6** | 0.634 | **0.619** | **0.776** |
| Gemma2-2b | MEERKAT | 30 | 0.94 | 0.855 | 0.947 | 0.734 | 0.568 | **0.644** | 0.601 | 0.755 |
| | MEERKAT-VP | 50 | **0.949** | 0.853 | **0.969** | **0.782** | 0.551 | 0.615 | 0.620 | 0.762 |
| | MEERKAT | 50 | 0.945 | **0.857** | 0.966 | 0.767 | **0.613** | **0.634** | **0.623** | **0.772** |
| | MEERKAT-VP | 100 | **0.944** | 0.812 | **0.97** | 0.733 | 0.551 | 0.634 | **0.634** | **0.754** |
| | MEERKAT | 100 | 0.94 | **0.851** | 0.951 | **0.745** | 0.551 | 0.634 | 0.574 | 0.749 |

Table 15: MEERKAT performance at local step = 1 with varying outlier percentages across the LLaMA-3.2-1B, Qwen2-1.5b, and Gemma2-2b models. We report test accuracy on SST-2, AgNews, Yelp, BoolQ, RTE, WSC, and WIC under both IID and Non-IID client data settings. Bold numbers indicate the highest value in each row.

| Model | Outlier Percentage | IID | | | | | | | Non-IID | | | | | | |
|---|---|---|---|---|---|---|---|---|---|---|---|---|---|---|---|
| | | SST-2 | AgNews | Yelp | BoolQ | RTE | WSC | WIC | SST-2 | AgNews | Yelp | BoolQ | RTE | WSC | WIC |
| | 5e-1 | 0.917 | 0.72 | 0.965 | 0.725 | 0.653 | 0.644 | 0.634 | 0.895 | 0.669 | 0.964 | 0.684 | 0.644 | 0.653 | 0.594 |
| | 5e-2 | 0.913 | 0.861 | 0.966 | 0.749 | 0.653 | 0.644 | 0.633 | 0.915 | 0.87 | **0.97** | 0.722 | 0.653 | 0.644 | 0.619 |
| LLaMA-3.2-1B | 5e-3 | 0.900 | **0.885** | **0.971** | 0.769 | 0.702 | 0.653 | 0.614 | **0.930** | 0.874 | 0.963 | **0.753** | 0.620 | 0.66 | 0.62 |
| | 5e-4 | 0.910 | 0.877 | 0.954 | **0.773** | **0.720** | **0.663** | 0.641 | 0.911 | **0.888** | 0.956 | 0.700 | 0.693 | **0.663** | 0.628 |
| | 5e-5 | **0.922** | 0.879 | 0.964 | 0.724 | 0.631 | 0.625 | **0.648** | 0.92 | 0.876 | 0.940 | 0.725 | 0.613 | **0.663** | 0.626 |
| | 5e-1 | 0.854 | 0.856 | 0.947 | 0.766 | 0.82 | 0.663 | 0.644 | 0.845 | 0.854 | 0.946 | 0.753 | 0.826 | 0.682 | 0.631 |
| | 5e-2 | 0.925 | 0.868 | 0.949 | 0.778 | 0.826 | 0.692 | 0.647 | 0.93 | 0.853 | 0.943 | 0.759 | 0.822 | 0.663 | 0.663 |
| Qwen2-1.5b | 5e-3 | **0.926** | 0.851 | 0.945 | 0.765 | **0.813** | **0.692** | **0.658** | 0.924 | **0.866** | 0.94 | 0.759 | 0.822 | **0.692** | 0.661 |
| | 5e-4 | 0.92 | 0.764 | 0.943 | 0.774 | 0.813 | 0.682 | 0.645 | 0.918 | 0.848 | 0.943 | **0.762** | 0.813 | 0.682 | 0.647 |
| | 5e-5 | 0.903 | 0.78 | 0.941 | 0.748 | 0.80 | 0.673 | 0.625 | 0.896 | 0.799 | 0.937 | 0.739 | 0.80 | 0.673 | 0.633 |
| | 5e-1 | 0.842 | 0.867 | 0.963 | 0.751 | 0.657 | **0.673** | 0.626 | 0.871 | 0.855 | 0.952 | 0.695 | 0.653 | **0.663** | 0.619 |
| | 5e-2 | 0.932 | **0.878** | **0.977** | 0.809 | 0.791 | 0.663 | 0.623 | 0.92 | **0.863** | 0.968 | 0.786 | 0.706 | 0.653 | 0.634 |
| Gemma2-2b | 5e-3 | **0.952** | 0.871 | 0.971 | **0.837** | **0.800** | 0.663 | **0.639** | **0.942** | 0.853 | **0.97** | 0.807 | 0.751 | 0.653 | **0.645** |
| | 5e-4 | 0.941 | 0.824 | 0.967 | 0.83 | 0.764 | 0.663 | 0.612 | 0.941 | 0.83 | 0.962 | **0.831** | 0.746 | 0.634 | 0.63 |
| | 5e-5 | 0.92 | 0.828 | 0.952 | 0.797 | 0.6 | 0.634 | 0.606 | 0.922 | 0.764 | 0.949 | 0.774 | 0.56 | 0.634 | 0.601 |

Table 16: Performance comparison of Full-FedZO, LoRA-FedZO, and MEERKAT under synchronous updates with $localstep = 1$, evaluated on both IID and Non-IID client data settings(**Dirichlet** $\alpha = 0.5$) across LLaMA-3.2-1B, Qwen2-1.5b, and Gemma2-2b. We report test accuracy on SST-2, AgNews, Yelp, BoolQ, RTE, WSC, and WIC. Bold numbers indicate the highest value in each row.

| Model | Method | SST-2 | AgNews | Yelp | BoolQ | RTE | WSC | WIC | Acc |
|---|---|---|---|---|---|---|---|---|---|
| | Full-FedZO | **0.918** | 0.801 | 0.937 | 0.686 | 0.54 | 0.625 | 0.58 | 0.726 |
| LLaMA-3.2-1B (IID) | LoRA-FedZO | 0.915 | 0.855 | 0.944 | 0.672 | 0.599 | **0.663** | 0.599 | 0.749 |
| | MEERKAT | 0.900 | **0.885** | **0.971** | **0.773** | **0.702** | 0.653 | **0.614** | **0.785** |
| | Full-FedZO | 0.911 | 0.831 | 0.937 | 0.672 | 0.528 | 0.587 | 0.567 | 0.719 |
| LLaMA-3.2-1B (Non-IID) | LoRA-FedZO | 0.8669 | 0.842 | 0.944 | 0.659 | 0.53 | 0.567 | 0.578 | 0.712 |
| | MEERKAT | **0.93** | **0.888** | **0.963** | **0.753** | **0.67** | **0.66** | **0.62** | **0.783** |
| | Full-FedZO | 0.9013 | 0.726 | 0.918 | 0.700 | 0.797 | **0.710** | 0.579 | 0.761 |
| Qwen2-1.5b (IID) | LoRA-FedZO | **0.935** | 0.752 | 0.925 | 0.686 | 0.794 | 0.673 | 0.606 | 0.767 |
| | MEERKAT | 0.926 | **0.851** | **0.945** | **0.778** | **0.813** | 0.692 | **0.658** | **0.809** |
| | Full-FedZO | 0.844 | 0.725 | 0.937 | 0.688 | 0.769 | 0.663 | 0.565 | 0.741 |
| Qwen2-1.5b (Non-IID) | LoRA-FedZO | **0.932** | 0.76 | **0.944** | 0.682 | 0.773 | 0.682 | 0.565 | 0.763 |
| | MEERKAT | 0.924 | **0.866** | 0.94 | **0.762** | **0.822** | **0.692** | **0.661** | **0.809** |
| | Full-FedZO | 0.934 | 0.84 | 0.953 | 0.774 | 0.542 | 0.644 | 0.606 | 0.756 |
| Gemma2-2b (IID) | LoRA-FedZO | 0.942 | 0.856 | 0.94 | 0.735 | 0.52 | 0.644 | 0.606 | 0.749 |
| | MEERKAT | **0.952** | **0.871** | **0.971** | **0.837** | **0.8** | **0.663** | **0.639** | **0.819** |
| | Full-FedZO | 0.93 | 0.824 | 0.95 | 0.744 | 0.56 | 0.625 | 0.575 | 0.744 |
| Gemma2-2b (Non-IID) | LoRA-FedZO | 0.9415 | 0.825 | 0.954 | 0.711 | 0.528 | 0.625 | 0.578 | 0.737 |
| | MEERKAT | **0.942** | **0.853** | **0.97** | **0.807** | **0.751** | **0.653** | **0.645** | **0.803** |

Table 17: Performance comparison of Full-FedZO, LoRA-FedZO, and MEERKAT under synchronous updates with $localstep = 1$, evaluated on Non-IID client data settings (**Dirichlet** $\alpha = 0.3$) across LLaMA-3.2-1B, Qwen2-1.5b, and Gemma2-2b. We report test accuracy on SST-2, AgNews, Yelp, BoolQ, RTE, WSC, and WIC. Bold numbers indicate the highest value in each row.

| Model | Method | SST-2 | AgNews | Yelp | BoolQ | RTE | WSC | WIC | Acc |
|---|---|---|---|---|---|---|---|---|---|
| | Full-FedZO | 0.891 | 0.759 | 0.94 | 0.623 | 0.528 | 0.644 | 0.551 | 0.705 |
| LLaMA-3.2-1B (Non-IID) | LoRA-FedZO | 0.915 | 0.866 | 0.952 | 0.646 | 0.586 | 0.653 | 0.554 | 0.739 |
| | MEERKAT | **0.918** | **0.843** | **0.97** | **0.761** | **0.626** | **0.653** | **0.609** | **0.769** |
| | Full-FedZO | 0.52 | 0.347 | 0.45 | 0.62 | 0.532 | 0.632 | 0.51 | 0.516 |
| Qwen2-1.5b (Non-IID) | LoRA-FedZO | 0.855 | 0.732 | 0.907 | 0.674 | 0.72 | 0.634 | 0.603 | 0.732 |
| | MEERKAT | **0.91** | **0.809** | **0.954** | **0.772** | **0.822** | **0.682** | **0.661** | **0.801** |
| | Full-FedZO | 0.881 | 0.761 | 0.94 | 0.688 | 0.552 | 0.613 | 0.603 | 0.720 |
| Gemma2-2b (Non-IID) | LoRA-FedZO | 0.922 | 0.826 | 0.921 | 0.681 | 0.52 | 0.625 | 0.606 | 0.729 |
| | MEERKAT | **0.942** | **0.873** | **0.97** | **0.806** | **0.688** | **0.634** | **0.615** | **0.79** |

Table 18: Performance comparison of Full-FedZO, LoRA-FedZO, and MEERKAT under synchronous updates with $localstep = 1$, evaluated on Non-IID client data settings (**Dirichlet** $\alpha = 0.1$) across LLaMA-3.2-1B, Qwen2-1.5b, and Gemma2-2b. We report test accuracy on SST-2, AgNews, Yelp, BoolQ, RTE, WSC, and WIC. Bold numbers indicate the highest value in each row.

| Model | Method | SST-2 | AgNews | Yelp | BoolQ | RTE | WSC | WIC | Acc |
|---|---|---|---|---|---|---|---|---|---|
| | Full-FedZO | 0.891 | 0.754 | 0.933 | 0.626 | 0.522 | 0.365 | 0.512 | 0.658 |
| LLaMA-3.2-1B (Non-IID) | LoRA-FedZO | 0.902 | 0.845 | 0.942 | 0.643 | 0.533 | 0.365 | 0.559 | 0.684 |
| | MEERKAT | **0.92** | **0.794** | **0.965** | **0.745** | **0.582** | **0.644** | **0.603** | **0.750** |
| | Full-FedZO | 0.49 | 0.247 | 0.44 | 0.62 | 0.528 | 0.634 | 0.5 | 0.494 |
| Qwen2-1.5b (Non-IID) | LoRA-FedZO | 0.848 | 0.735 | 0.92 | 0.67 | 0.746 | 0.548 | 0.601 | 0.724 |
| | MEERKAT | **0.889** | **0.78** | **0.944** | **0.732** | **0.822** | **0.634** | **0.637** | **0.777** |
| | Full-FedZO | 0.879 | 0.741 | 0.937 | 0.681 | 0.48 | 0.634 | 0.601 | 0.708 |
| Gemma2-2b (Non-IID) | LoRA-FedZO | 0.91 | 0.78 | 0.914 | 0.682 | 0.551 | 0.567 | 0.608 | 0.716 |
| | MEERKAT | **0.944** | **0.866** | **0.971** | **0.805** | **0.728** | **0.605** | **0.628** | **0.792** |

Table 19: Test accuracy of MEERKAT versus DecomFL on Qwen2-1.5b with a single local step under Non-IID data settings (Dirichlet $\alpha = 1$). Results are shown for SST-2, BoolQ, RTE, and WSC; bold indicates the best score in each row. Experiments use 8 clients in total, with 2 clients participating in each round, following the DecomFL configuration.

| Model | Method | SST-2 | BoolQ | RTE | WSC |
|---|---|---|---|---|---|
| Qwen2-1.5b | DecomFL | 0.868 | 0.674 | 0.773 | 0.653 |
| | MEERKAT | **0.918** | **0.734** | **0.817** | **0.682** |

Table 20: Performance comparison of Task-Mask, and MEERKAT under synchronous updates with $localstep = 1$, evaluated on IID client data settings across LLaMA-3.2-1B, Qwen2-1.5b, and Gemma2-2b. We report test accuracy on SST-2, AgNews, Yelp, BoolQ, RTE, WSC, and WIC. Bold numbers indicate the highest value in each row.

| Model | Method | SST-2 | AgNews | Yelp | BoolQ | RTE | WSC | WIC |
|---|---|---|---|---|---|---|---|---|
| LLaMA-3.2-1B (IID) | Task | **0.910** | 0.847 | 0.957 | 0.718 | 0.661 | 0.644 | **0.661** |
| | MEERKAT | 0.90 | **0.885** | **0.971** | **0.773** | **0.702** | **0.653** | 0.614 |
| Qwen2-1.5b (IID) | Task | 0.936 | 0.827 | 0.954 | 0.765 | 0.83 | **0.711** | **0.664** |
| | MEERKAT | 0.926 | **0.851** | 0.945 | **0.778** | 0.813 | 0.692 | 0.658 |
| Gemma2-2b (IID) | Task | 0.942 | 0.868 | **0.972** | 0.78 | 0.728 | 0.644 | 0.6 |
| | MEERKAT | **0.952** | **0.871** | 0.971 | **0.837** | **0.8** | **0.663** | **0.639** |

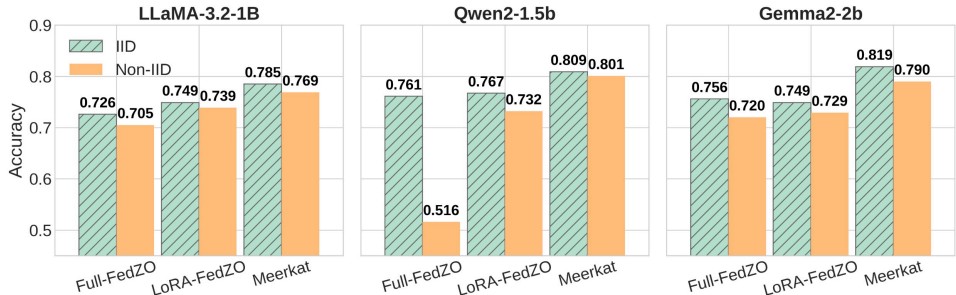

(a) This figure compares three methods—Full-FedZO, LoRA-FedZO, and MEERKAT—on three LLMs: LLaMA-3.2-1B, Qwen2-1.5b, and Gemma2-2b. The x-axis shows the different methods, and each method has two bars indicating performance under IID and Non-IID settings. The Non-IID results are obtained under a Dirichlet $\alpha = 0.3$ .The y-axis represents the average test accuracy across multiple downstream tasks—SST2, AgNews, Yelp, BoolQ, RTE, WSC, and WiC.

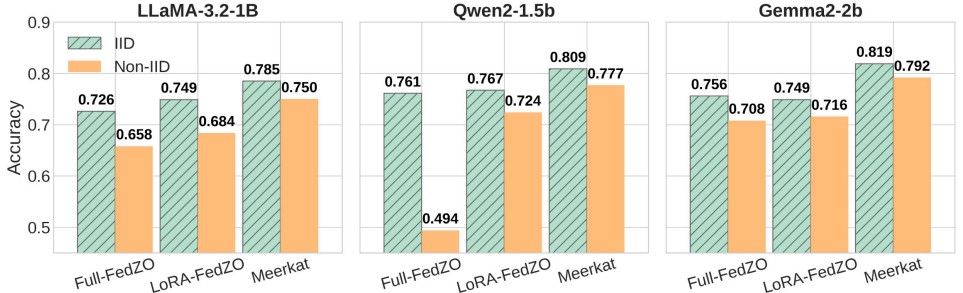

(b) This figure compares three methods—Full-FedZO, LoRA-FedZO, and MEERKAT—on three LLMs: LLaMA-3.2-1B, Qwen2-1.5b, and Gemma2-2b. The x-axis shows the different methods, and each method has two bars indicating performance under IID and Non-IID settings. The Non-IID results are obtained under a Dirichlet $\alpha = 0.1$ .The y-axis represents the average test accuracy across multiple downstream tasks—SST2, AgNews, Yelp, BoolQ, RTE, WSC, and WiC.

Figure 6: Comparison of Full-FedZO, LoRA-FedZO, and MEERKAT on LLaMA-3.2-1B, Qwen2-1.5b, and Gemma2-2b under IID and Non-IID settings with varying Dirichlet $\alpha$. Subfigure (a) presents results for Non-IID data generated with $\alpha = 0.3$, while Subfigure (b) shows results for Non-IID data with $\alpha = 0.1$.

Table 21: Performance comparison of Task-Mask, which uses downstream task data to select sensitive model parameters, and MEERKAT under synchronous updates with $localstep = 1$, evaluated on Non-IID client data settings (**Dirichlet** $\alpha = 0.5$) across LLaMA-3.2-1B, Qwen2-1.5b, and Gemma2-2b. We report test accuracy on SST-2, AgNews, Yelp, BoolQ, RTE, WSC, and WIC. Bold numbers indicate the highest value in each row.

| Model | Method | SST-2 | AgNews | Yelp | BoolQ | RTE | WSC | WIC |
|---|---|---|---|---|---|---|---|---|
| **LLaMA-3.2-1B (Non-IID)** | Task | 0.904 | 0.874 | 0.956 | 0.744 | 0.591 | 0.615 | 0.622 |
| | MEERKAT | **0.93** | **0.888** | **0.963** | **0.753** | **0.62** | **0.66** | **0.62** |
| **Qwen2-1.5b (Non-IID)** | Task | 0.938 | 0.863 | 0.956 | 0.779 | 0.817 | 0.692 | 0.65 |
| | MEERKAT | **0.924** | **0.866** | **0.94** | **0.762** | **0.822** | **0.692** | **0.661** |
| **Gemma2-2b (Non-IID)** | Task | 0.91 | 0.834 | 0.966 | 0.822 | 0.72 | 0.644 | 0.578 |
| | MEERKAT | **0.942** | **0.853** | **0.97** | **0.807** | **0.751** | **0.653** | **0.645** |

Table 22: Test accuracy of MEERKAT versus Task-Mask on Qwen2-1.5b with a 10 local step under Non-IID data settings (Dirichlet $\alpha = 0.5$). Results are shown for SST-2, BoolQ, RTE, and WSC; bold indicates the best score in each row. Experiments use 8 clients in total, with 2 clients participating in each round, following the DecomFL configuration.

| Model | Method | SST-2 | BoolQ | RTE | WSC |
|-------|--------|-------|-------|-----|-----|
| **Qwen2-1.5b** | Task | 0.932 | **0.784** | **0.823** | 0.681 |
| | MEERKAT | **0.944** | 0.752 | 0.813 | **0.682** |

Table 23: MEERKAT Convergence Rounds for the LLaMA-3.2-1B, Gemma2-2B, and Qwen2-1.5B models on the SST-2, AgNews, Yelp, and BoolQ tasks, with 10 local steps.

| Model | SST-2 | AgNews | Yelp | BoolQ |
|-------|-------|--------|------|-------|
| Gemma2-2B | 39 | 61 | 29 | 43 |
| Qwen2-1.5B | 51 | 75 | 36 | 70 |
| LLaMA-3.2-1B | 85 | 77 | 52 | 97 |

Table 24: Computation and Communication Efficiency Benchmark Shows MEERKAT's Superior Resource Usage over Baselines. We benchmarked resource usage on Qwen2-1.5B with 10 clients (FP16). Setting: Full-FedZO vs Meerkat vs LoRA-FedZO, where LoRA is configured with rank = 16, $\alpha$ = 16—the same setting used in Table 1.

| Method/Metrics | RAM (Peak) | Upload/Client | Download/Client |
|----------------|------------|---------------|-----------------|
| Full-FedZO | 12,600 MiB | 0.078 KB | 2.875 GB |
| LoRA-FedZO | 10,741 MiB | 0.078 KB | 35.22 MB |
| MEERKAT (0.1% mask) | **7,850 MiB** | 0.078 KB | **2.50 MB** |

Table 25: Performance comparison on LLaMA-3.2-1B under Non-IID Dirichlet partition ($\alpha = 0.5$) with $T = 10$ local steps. While ZO methods cannot match back-propagation due to gradient noise from limited sampling, MEERKAT-VP achieves competitive accuracy (0.764 avg) with significantly lower memory consumption, and outperforms several Non-IID FL baselines (FedDYN, FedAvgM, FedSA-LoRA, and stochastic controlled) under the same training setup. We adapt FedDYN following the original paper with $\alpha = 0.01$.

| Method | SST-2 | AGNews | Yelp | BoolQ | RTE | WSC | WIC | Avg |
|--------|-------|--------|------|-------|-----|-----|-----|-----|
| Back-propagation | 0.925 | 0.893 | 0.968 | 0.751 | 0.644 | 0.660 | 0.630 | 0.782 |
| Stochastic Controlled | 0.880 | 0.720 | 0.901 | 0.612 | 0.523 | 0.612 | 0.580 | 0.690 |
| FedAvgM | 0.901 | 0.821 | 0.941 | 0.629 | 0.580 | 0.613 | 0.600 | 0.726 |
| FedSA-LoRA | 0.905 | 0.832 | 0.920 | 0.630 | 0.570 | 0.622 | 0.570 | 0.721 |
| MEERKAT+FedDYN | 0.917 | 0.841 | 0.954 | 0.638 | 0.564 | 0.615 | 0.570 | 0.728 |
| MEERKAT-VP | 0.922 | 0.864 | 0.962 | 0.713 | 0.617 | 0.644 | 0.625 | 0.764 |

Table 26: Effect of different sparsity density ratios on LLaMA-3-8B under Non-IID Dirichlet $\alpha = 0.5$. We compare density ratios $10^{-3}$ and $10^{-4}$ using the same transferable mask construction pipeline. Both settings achieve strong performance.

| Density ratio | SST-2 | AGNews | Yelp | BoolQ | RTE | WSC | WIC | Avg |
|---------------|-------|--------|------|-------|-----|-----|-----|-----|
| $1 \times 10^{-3}$ | 0.950 | 0.851 | 0.954 | 0.831 | 0.755 | 0.674 | 0.660 | 0.811 |
| $1 \times 10^{-4}$ | 0.941 | 0.862 | 0.956 | 0.861 | 0.783 | 0.664 | 0.640 | 0.815 |

Table 27: Scalability of MEERKAT and MEERKAT-VP with respect to the number of clients on Qwen2-1.5B. Increasing the number of clients from 10 to 20 does not degrade performance; MEERKAT-VP-20 even slightly improves the average accuracy.

| Model | SST2 | AgNews | Yelp | BoolQ |
|-------|------|--------|------|-------|
| MEERKAT-VP (20 clients) | 0.951 | 0.885 | 0.936 | 0.756 |
| MEERKAT (20 clients) | 0.929 | 0.869 | 0.922 | 0.719 |
| MEERKAT (10 clients) | 0.949 | 0.881 | 0.934 | 0.752 |

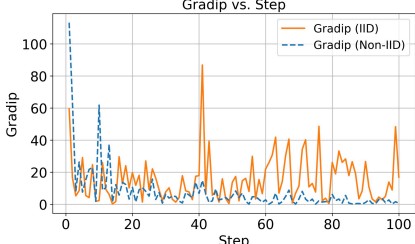

(a) The GradIP measured for IID and Non-IID client data under the WIC task using the Llama-3.2-1B model.

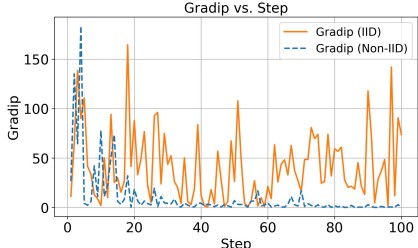

(b) The GradIP measured for IID and Non-IID client data under the AgNews task using the Llama-3.2-1B model.

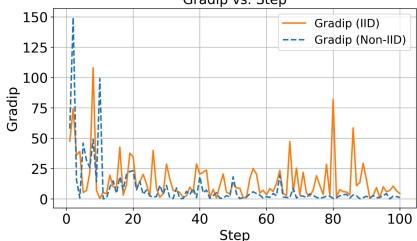

(c) The GradIP measured for IID and Non-IID client data under the Yelp task using the Llama-3.2-1B model.

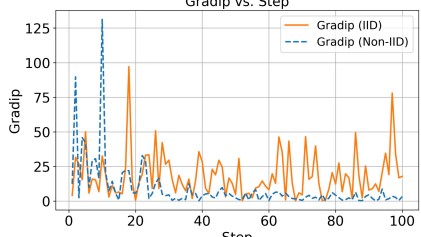

(d) The GradIP measured for IID and Non-IID client data under the BoolQ task using the Llama-3.2-1B model.

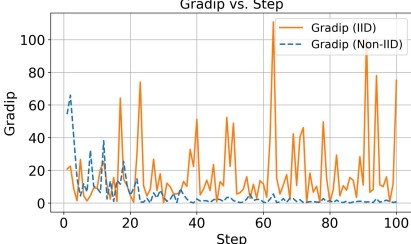

(e) The GradIP measured for IID and Non-IID client data under the RTE task using the Llama-3.2-1B model.

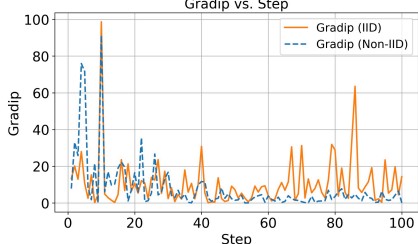

(f) The GradIP measured for IID and Non-IID client data under the WSC task using the Llama-3.2-1B model.

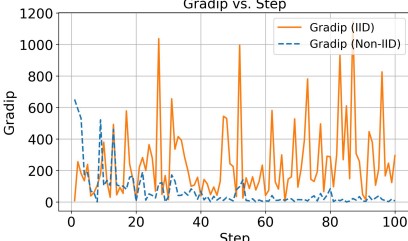

(g) The GradIP measured for IID and Non-IID client data under the BoolQ task using the Gemma-2-2b model.

Figure 7: These figures show GradIP (Definition 2.3) curves under IID and Non-IID settings, computed over 100 local training steps on six datasets (WSC, BoolQ, RTE, WIC, AgNews, Yelp) using the Llama-3.2-1B model with density level $5 \times 10^{-3}$. An extra BoolQ result is shown for the Gemma-2-2B model.

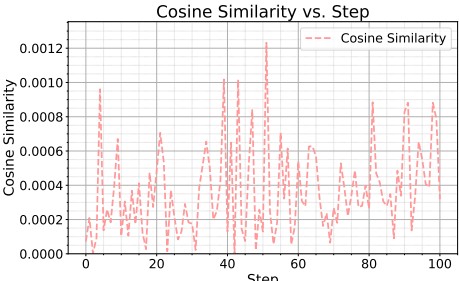
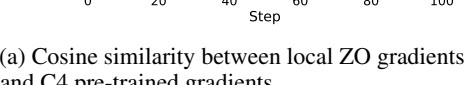
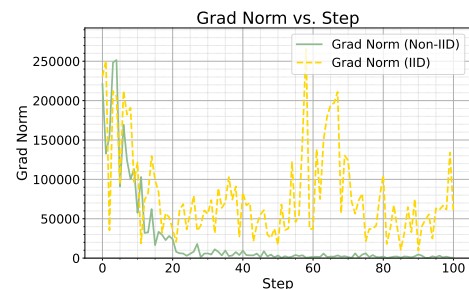

(a) Cosine similarity between local ZO gradients and C4 pre-trained gradients.

(b) Gradient norm from local ZO training under Non-IID and IID data distribution.

Figure 8: The left panel shows the cosine similarity between locally computed ZO gradients and gradients from the C4-pre-trained data, illustrating that the two gradient vectors remain nearly orthogonal throughout training. The right panel presents the norm of local ZO gradients over training steps, showing a consistent decay and convergence in magnitude under Non-IID and IID data distribution. These observations are obtained under density level of $5 \times 10^{-3}$.

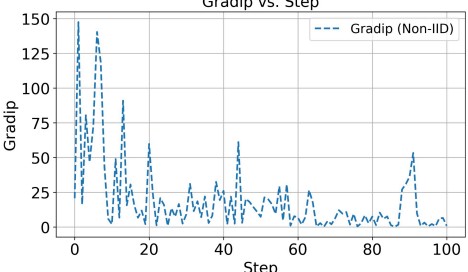
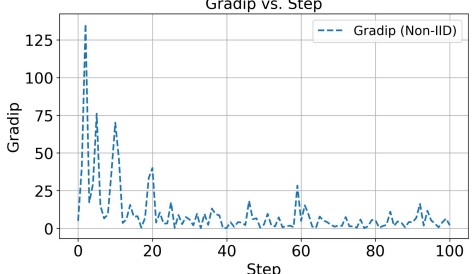

(a) GradIP for Non-IID clients on the AgNews task, where the two classes have a highly imbalanced ratio (5 vs. 89 samples).

(b) GradIP for Non-IID client data on the BoolQ task, where the two classes have a highly imbalanced ratio (6 vs. 190 samples).

Figure 9: These subfigures show GradIP (see Definition 2.3) for LLaMA-3.2-1B under Non-IID client data with 100 local training steps. Subfigure (a) uses AgNews (5 vs. 89), while Subfigure (b) uses BoolQ (6 vs. 190).

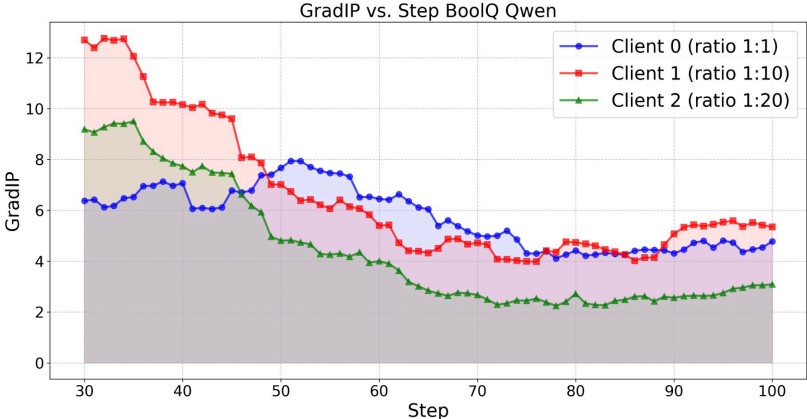

(a) The experiments, conducted using the Qwen2-1.5B model on the BoolQ dataset, reveal that under Non-IID settings—especially with a 1:20 class imbalance—there is a pronounced decline in GradIP between the early and later stages of training. In the extreme Non-IID case, the GradIP values in the later stages tend to approach zero.

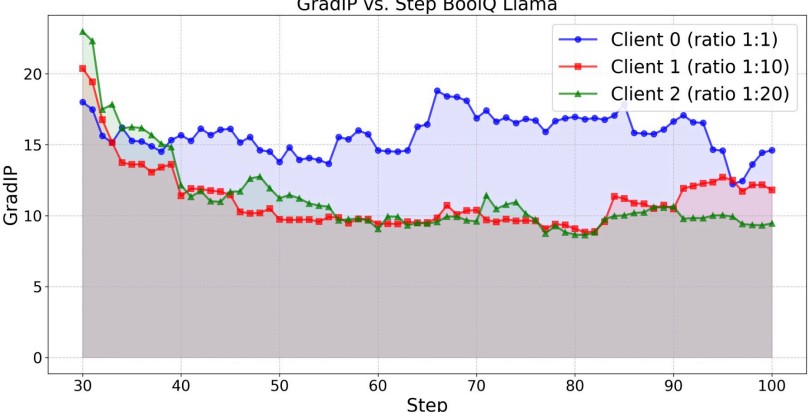

(b) The experiments, conducted using the Llama-3.2-1B model on the BoolQ dataset, reveal that under Non-IID settings—especially with a 1:20 class imbalance—there is a pronounced decline in GradIP between the early and later stages of training. In the extreme Non-IID case, the GradIP values in the later stages tend to approach zero.

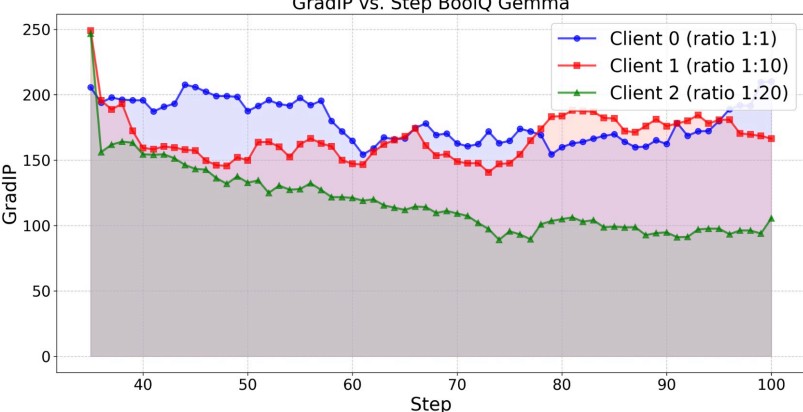

(c) The experiments, conducted using the Gemma-2-2B model on the BoolQ dataset, reveal that under Non-IID settings—especially with a 1:20 class imbalance—there is a pronounced decline in GradIP between the early and later stages of training. In the extreme Non-IID case, the GradIP values in the later stages tend to approach zero.

Figure 10: GradIP analysis for different models on the BoolQ dataset under Non-IID and IID conditions: As the class imbalance ratio increases, GradIP in the later training stages tends to approach zero. This decline is more pronounced under Non-IID settings, where the gap between initial and final GradIP values is larger than in the IID case. All trends are visualized using a moving average for clarity.

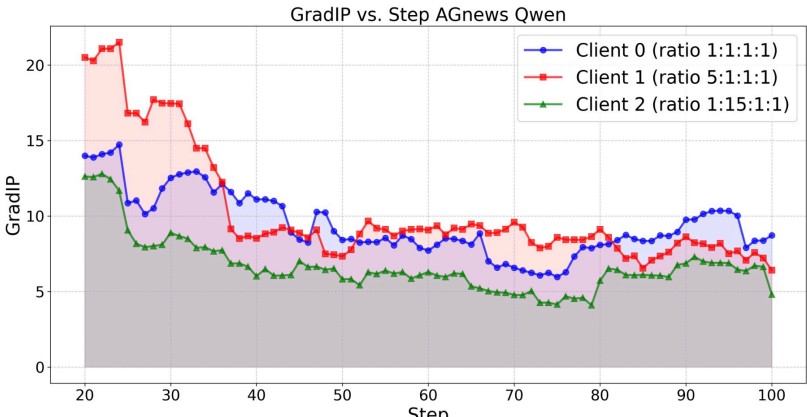

(a) The experiments, conducted using the Qwen2-1.5B model on the AGNews dataset, reveal that under Non-IID settings—especially with a 1:15:1:1 class imbalance—there is a pronounced decline in GradIP between the early and later stages of training. In the extreme Non-IID case, the GradIP values in the later stages tend to approach zero.

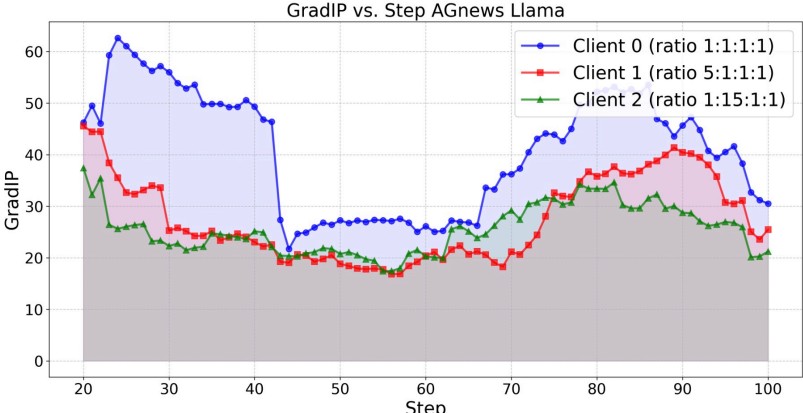

(b) The experiments, conducted using the Llama-3.2-1B model on the AGNews dataset, reveal that under Non-IID settings—especially with a 1:15:1:1 class imbalance—there is a pronounced decline in GradIP between the early and later stages of training. In the extreme Non-IID case, the GradIP values in the later stages tend to approach zero.

Figure 11: GradIP analysis for different models on the AGNews dataset under Non-IID and IID conditions: As the class imbalance ratio increases, GradIP in the later training stages tends to approach zero. This decline is more pronounced under Non-IID settings, where the gap between initial and final GradIP values is larger than in the IID case. All trends are visualized using a moving average for clarity; consequently, the plotted lines do not begin at step zero, as the initial data points are used to compute the first averaged value. This is an intentional effect of the visualization, not an error or a result of missing data.

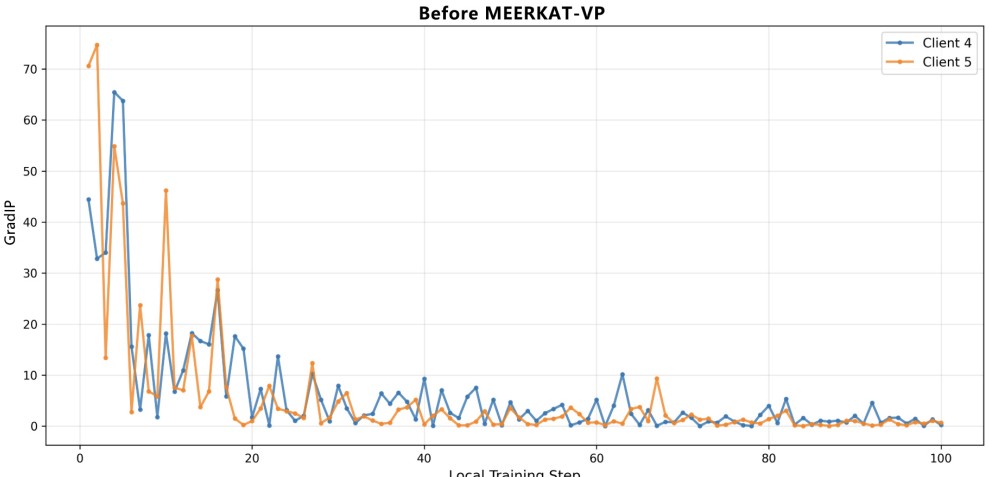

(a) GradIP trajectories for the two extreme Non-IID clients on the SST2 task with Qwen2-1.5B, before MEERKAT-VP training.

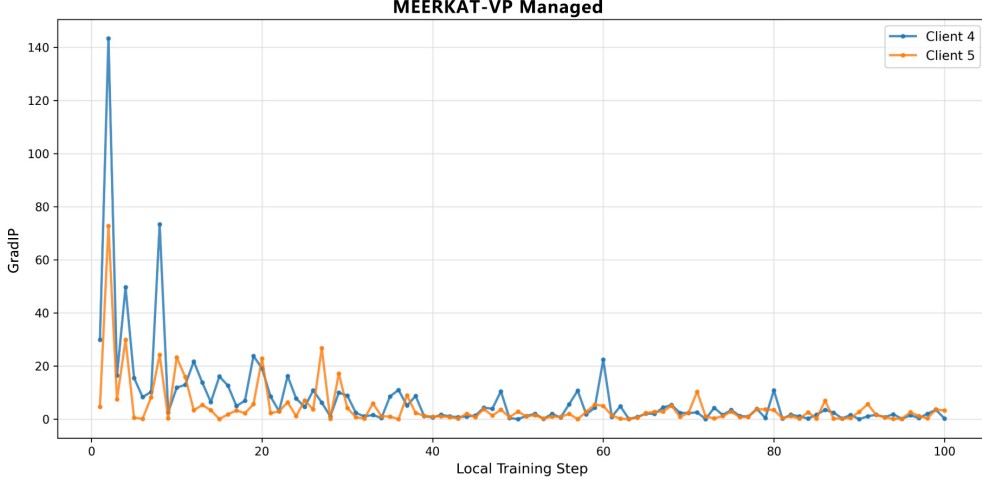

(b) GradIP trajectories for the same two extreme Non-IID clients on SST2 *after MEERKAT-VP training has converged* (global validation accuracy $\approx 90\%$).

Figure 12: GradIP trajectories (Definition 2.3) for Qwen2-1.5B on SST2 with 6 clients (2 extreme Non-IID, 4 IID). Subfigure (a) shows the GradIP trajectories of the two extreme Non-IID clients at initialization, while Subfigure (b) shows the trajectories for the same clients after MEERKAT-VP training has converged. The shape of the trajectories remains similar before and after training, supporting our claim that GradIP is primarily a data-distribution-driven signal rather than a direct reflection of the global model state.

