# OpenReview forum: "Mitigating Non-IID Drift in Zeroth-Order Federated LLM Fine-Tuning with Transferable Sparsity"
_ICLR.cc/2026/Conference — ICLR 2026 Poster_

### Official Review · Reviewer_46ze · 2025-10-31

**Soundness:** 4
**Presentation:** 3
**Contribution:** 3
**Rating:** 6
**Confidence:** 4

**Summary:**

This paper proposes MEERKAT, a novel framework for federated fine-tuning of LLMs that combines Zeroth-Order optimization with extreme sparsity. The core idea is that this combination is extremely memory-efficient and communication-efficient. The authors claim this efficiency enables high-frequency client-server synchronization, which in turn is the key to mitigating Non-IID client drift. Furthermore, the paper introduces a "virtual path" mechanism, where the server can reconstruct client update trajectories, and use"GradIP", a metric based on the inner product of client and pre-training gradients, that can identify clients with extreme Non-IID data. The authors then propose MEERKAT-VP, an extension that uses this signal to apply early stopping to these problematic clients, further improving global model performance.

**Strengths:**

1. **Strong Empirical Performance:** The method is well-validated empirically. Empirical results clearly show that MEERKAT consistently and significantly outperforms full-parameter ZO and LoRA-based ZO baselines, especially in Non-IID settings.
2. **Comprehensive Analysis:** The paper provides strong support for its claims through both theoretical convergence analysis and extensive empirical ablations. The validation is structured clearly around three main claims, and each is convincingly argued with evidence.
3. **Novel Mechanism for Heterogeneity:** The discovery of the "GradIP" phenomenon is a novel contribution. Using this signal, which is enabled by the virtual path, to identify and manage extreme Non-IID clients (MEERKAT-VP) is a clever and effective FL-native strategy for handling heterogeneity.

**Weaknesses:**

The paper's primary strategy for handling extreme Non-IID clients (MEERKAT-VP) is purely defensive. It aims to "mitigate their adverse impact" by identifying them via GradIP and applying early stopping. This implies that the base MEERKAT method is still vulnerable to significant client drift from these clients. This strategy is inherently limited: it focuses on reducing the participation of "bad" clients rather than developing a mechanism to effectively learn from their (potentially valuable) information.

**Questions:**

The core optimization (sparse ZO on a static mask) seems to be a general technique, not specialized for FL. Are there any other properties of this sparse ZO method that make it uniquely suited for the federated setting, beyond the communication efficiency gains and the virtual path mechanisms they enable?

---

> ### Author Response · Authors · 2025-11-18
>
> We thank the reviewer for the comments and suggestions to improve our paper.
>
> ### [W1 – Concern about MEERKAT-VP being a purely defensive strategy – Our clarification on preserving client information while preventing drift amplification]
>
> **(1) All client data are fully used.** Each client maintains a data pointer and resumes from the next unprocessed batch in the following round. Even when a client is early-stopped to \(T = 1\), its entire dataset is eventually consumed (Claim 3). No samples—minority or otherwise—are removed.
>
> **(2) Early stopping prevents drift amplification, not “discarding” client information.** Under ZO updates, each local step follows the client’s own update direction. For extreme Non-IID clients, this direction is computed from a very narrow and skewed slice of the data distribution, and is often strongly misaligned with the global descent direction (as approximated by pre-training gradients). Allowing many local steps lets this misaligned direction accumulate and move the local model far away from the global trajectory; at aggregation time, such updates act as high-variance noise on the global model. By restricting these clients to \(T = 1\), MEERKAT-VP still uses their updates every round, but limits how far they can drift, thereby reducing their ability to distort the global training path. This is consistent with our analysis in Appendix C.5, which shows that \(E_{\text{MEERKAT-VP}} < E_{\text{MEERKAT}}\) and that this gap widens as heterogeneity increases.
>
> **(3) Federated learning aims to train a global model that performs well on the joint data distribution, not to optimize each local model independently.**  When a few clients are extremely skewed (e.g., seeing only one class), aggressively fitting their local objectives can harm the global objective. Prior FL methods therefore limit the influence of problematic clients: Astraea [1] balances imbalanced clients, Krum [2] filters outlier gradients, FedCor [3] selects diverse clients based on gradient correlations, and Oort [4] downweights low-utility clients. MEERKAT-VP follows the same high-level principle of protecting the global model, but (i) all client data are still eventually used via the data pointer, and (ii) the decision is driven by GradIP, a ZO-specific misalignment signal rather than a heuristic rule.
>
> [1] Moming Duan et al., Astraea: Self-balancing Federated Learning for Improving Classification Accuracy of Mobile Deep Learning Applications.
>
> [2] Peva Blanchard et al., Machine Learning with Adversaries: Byzantine Tolerant Gradient Descent, NeurIPS.
>
> [3] Minxue Tang et al., FedCor: Correlation-Based Active Client Selection Strategy for Heterogeneous Federated Learning, CVPR.
>
> [4] Fan Lai et al., Oort: Efficient Federated Learning via Guided Participant Selection, OSDI.
>
> ### [Q1 – Why a static shared sparse ZO mask is particularly suited for FL, beyond communication and virtual paths]
>
> Thank you for this insightful question.
>
> **(1) Shared mask vs. client-specific masks (UnionMask ablation).**
> To test whether more FL-specific, client-dependent masks help, we implemented a UnionMask variant (Appendix Table 11), where each client first computes its own sensitivity mask and the server takes their union. This design is more locally adaptive but requires extra mask exchange and yields higher communication and computation costs. Empirically, it does not outperform our global static mask (e.g., 0.736 vs. 0.759 average accuracy in Table 11), while being more expensive. This suggests a shared sparse subspace is sufficient for FL.
>
> **(2) FL-specific properties of a static shared mask.**
> Beyond this ablation, the static mask has two properties that are particularly suited to FL:
> **Privacy.** The mask is computed once on public pre-training data (e.g., C4). No client data or client-specific gradients are ever exposed to calculate the mask.**Robustness under Non-IID.** The shared mask selects globally sensitive parameters that are broadly useful across clients, even when their local data distributions are heterogeneous. In contrast, client-specific masks (as in UnionMask) are heavily influenced by local Non-IID data distributions, and their union tends to include more client-specific directions, making the active parameter set denser without improving accuracy (Table 11). Using a single global mask keeps the sparse subspace focused on consensus-important coordinates and leads to more stable aggregation.Taken together, the **global, transferable static mask** in MEERKAT is particularly well suited to FL: it respects privacy constraints, focuses updates on a shared set of globally sensitive parameters across heterogeneous clients, and in practice outperforms more complex client-specific mask designs.

---

> > ### Comment · Reviewer_46ze · 2025-11-26
> >
> > Thank you for the clear response. Your clarifications address my concerns, and I will maintain my positive assessment and acceptance recommendation.

---

> > > ### Author Response · Authors · 2025-11-26
> > >
> > > Thank you for your response and for maintaining your positive assessment. We sincerely appreciate your understanding of our work and your support. Your constructive comments have been very valuable in strengthening our paper.

---

### Official Review · Reviewer_Gtsq · 2025-10-31

**Soundness:** 2
**Presentation:** 3
**Contribution:** 2
**Rating:** 2
**Confidence:** 3

**Summary:**

This paper proposes MEERKAT, a sparse zeroth-order optimization method for federated LLM fine-tuning. By updating a static, highly sparse subset of parameters, MEERKAT reduces communication costs and enables frequent synchronization to alleviate Non-IID drift. It also introduces a “virtual path” mechanism that uncovers the GradIP phenomenon—the inner product between server and client gradients varies across data heterogeneity. Building on this, MEERKAT-VP detects extreme Non-IID clients and mitigates their impact through early stopping. The paper provides theoretical convergence results and validates the method with extensive experiments.

**Strengths:**

* The paper is clearly presented, systematically proposing three distinct claims and validating them one by one.
* It includes a detailed convergence analysis for the proposed methods.
* The experimental evaluation is comprehensive, covering various benchmarks and recent LLMs.

**Weaknesses:**

* The use of sparse updates for communication efficiency in FL is a well-explored area, often framed as model compression. The paper should provide a more thorough comparison with this body of work to clarify the unique contributions of applying sparsity specifically to ZO-based FL fine-tuning. Applying sparsity to new FL models (here is federated fine-tuning) seems not a strong contribution.
* Methods for tackling the Non-IID issue in compressed FL have also been developed, such as the one in [a]. The paper would be stronger if it compared its GradIP-aware solution not just with other non-sparse fine-tuning methods, but also with compression-aware FL baselines that explicitly address Non-IID data, even if they are not designed for LLMs.
* The rationale for setting the number of local training steps to one for identified Non-IID clients is not fully convincing. These clients may hold important minority data that is crucial for preventing the global model from becoming biased. You should not move it only because it is minority and hard to converge into the global model. The paper should better justify why this aggressive early stopping is preferable to other strategies that might still leverage the clients' unique data more fully.

[a] Huang, Xinmeng, Ping Li, and Xiaoyun Li. "Stochastic Controlled Averaging for Federated Learning with Communication Compression." The Twelfth International Conference on Learning Representations.

**Questions:**

* The client identification mechanism in MEERKAT-VP uses an "initial phase" and a "later phase" of local training to analyze GradIP scores. What is the motivation for this two-phase design?
* Figure 3 shows the GradIP trajectories. To make the effect of the proposed solution more intuitive, could the authors add a plot showing the trajectory for a client being managed by MEERKAT-VP (i.e., after mitigating the global weight changes after the design)?

---

> ### Author Response · Authors · 2025-11-18
>
> We thank the reviewer for the comments and suggestions to improve our paper!
> ### [W1–Sparse updates are just model compression;applying sparsity to a new FL setting is not a strong contribution–Clarifying our study contribution]
>
> In our study,sparsity is used not only for communication efficiency but primarily for computational efficiency:we train only a sparse subset of parameters,yet still outperform Full-ZO. Beyond generic compression,our method combines three ZO-specific ingredients:(i)ZO-specific,transferable extreme sparsity (≤0.1%).We derive a static sensitivity mask from pre-training gradient squares so that ZO perturbs only the most sensitive weights.This mask transfers across domains and achieves higher accuracy than many method at the same communication frequency(Table 1,Table 10).(ii) Scalar-only projected-gradient communication enabling high-frequency synchronization.We show MEERKAT/MEERKAT-VP in Algorithm2 and Algorithm3.This differs from standard compressed-gradient FL and is tailored to ZO perturbations.(iii)Virtual-path GradIP and MEERKAT-VP for detecting and mitigating extreme Non-IID.Such virtual-path GradIP analysis is specific to ZO projected-scalar communication and is not available in first-order compressed-gradient FL.
>
> ### [W2–Insufficient benchmarks for Non-IID client drift – Clarifying our experimental choices and extensions]
>
> Thank you for your valuable suggestion.We benchmarked MEERKAT-VP against MEERKAT and Random Client Selection,and we also include a widely-used Non-IID mitigation method FedDYN (ICLR 2021)[1].The results presented in Figure 4 and Table 24 in appendix.Regarding [2],we adapted it to ZO LLM federated fine-tuning.Table 1 shows the comparison on LLaMA-3.2-1B(T=10,Dirichlet α=0.5):MEERKAT-VP achieves 10.7% higher accuracy than Stochastic Controlled(0.764vs0.69) and 5% over FedDYN(0.764vs0.728).This validates that our GradIP-based client selection is more effective for ZO FL.
>
> **Table1-MEERKAT-VP vs Stochastic Controlled**
> |Model|SST2|AgNews|Yelp|BoolQ|RTE|WSC|WIC|AVG|
> |-|-|-|-|-|-|-|-|-|
> |Stochastic Controlled|0.88|0.72|0.901|0.612|0.523|0.612|0.58|0.69|
> |MEERKAT-VP|0.922|0.864|0.962|0.713|0.617|0.644|0.625|0.764|
>
> [1]Durmus Alp Emre Acar et al,FEDERATED LEARNING BASED O DYNAMIC REGULARIZATION,ICLR
>
> [2]Huang Xinmeng et al,Stochastic Controlled Averaging for Federated Learning with Communication Compression.ICLR
>
> ### [W3–Rationale for early stopping extreme Non-IID clients–Data pointer ensures fairness and prevents discarding minority data]
>
> **(1)Data pointer ensures full data utilization.** Each client maintains a data pointer to resume from the next batch in the following round. No samples are skipped, and all minority data are eventually used(Claim 3).**(2)Theory supports that early stopping improves global convergence.** For extreme Non-IID clients,multi-step ZO updates amplify misaligned directions.Appendix C.5 shows that,under fixed local step T and global round R,E_VP<E_MEERKAT and the gap widens as heterogeneity increases,demonstrating that early stopping reduces drift.**(3)Early stopping for extreme Non-IID clients still works well even when all clients are extreme Non-IID.** If all clients are labeled as extreme Non-IID,the setting simply degenerates to high-frequency synchronization.We show in the appendix that even under a Dirichlet partition with alpha=0.1,our method still performs very well and outperforms both Full-FedZO and LoRA-FedZO.
>
> ### [Q1-What is the motivation for this two-phase design-Ensuring stable identification under ZO noise]
>
> The motivation for using two phases is that ZO fine-tuning exhibits substantial stochastic noise.A single GradIP value at any moment is therefore not reliable for determining whether a client is IID or extreme Non-IID.By comparing the initial-phase average to the later-phase average,MEERKAT-VP identifies clients based on stable trajectory patterns rather than instantaneous noisy values, avoiding false positives and ensuring reliable detection.
>
> ### [Q2–Visualizing GradIP after MEERKAT-VP–GradIP is a data-distribution-driven signal]
>
> GradIP trajectories serve as a signal for identifying extreme Non-IID clients and are primarily driven by the local data distribution rather than the global model state (Appendix C.6).To support this,we conduct an SST2 experiment with Qwen2-1.5B and 6 clients(2 extreme Non-IID,4 IID),and plot GradIP for the two extreme Non-IID clients both at initialization and after MEERKAT-VP training has converged(global validation accuracy≈90%). In both cases,the trajectories exhibit the same characteristic GradIP pattern.
> MEERKAT-VP training:https://anonymous.4open.science/r/ICLRREBUTTAL-7196/meerkat_vp_manage.pdf,
> Initial gradip:https://anonymous.4open.science/r/ICLRREBUTTAL-7196/initial_meerkat_vp.pdf
> In light of the additional experiments and clarifications above,we feel that a rating of 2 may be somewhat conservative.We would be very grateful if the reviewer could kindly reconsider the score.

---

> ### Author Response · Authors · 2025-11-24
>
> Dear Reviewer Gtsq,
>
> Thank you once again for the time and care you have devoted to reviewing our submission. We posted our author response on November 18th, including additional experiments and further clarifications regarding our use of sparsity and the MEERKAT-VP early stopping mechanism.
>
> If there are any remaining concerns or points that would benefit from further explanation, we would be very glad to provide additional clarification. Otherwise, if our response has addressed your main questions, we would be **sincerely grateful if you could kindly consider updating your assessment** at your convenience.
> Thank you again for your time and thoughtful feedback.
>
> Warm regards,
>
> 15272 Authors

---

### Official Review · Reviewer_TB4g · 2025-11-01

**Soundness:** 3
**Presentation:** 2
**Contribution:** 3
**Rating:** 6
**Confidence:** 4

**Summary:**

This paper introduces MEERKAT and its extension MEERKAT-VP, a novel methodology to tackle the twin challenges of high communication overhead and Non-IID data drift when fine-tuning LLMs in a Federated Learning setting. MEERKAT addresses efficiency by using a sparse Zeroth-Order Optimization method that limits fine-tuning to an extremely sparse (e.g., <0.1%), static subset of parameters, drastically reducing communication costs and enabling high-frequency server synchronization to mitigate drift. Building on this, MEERKAT-VP uses the concept of a virtual path based on local updates to detect the GradIP phenomenon in extreme Non-IID clients, subsequently applying early stopping to restrict these clients' local steps, thereby improving global model quality. The experiments confirm that MEERKAT, through its combined approach of extreme sparsity and strategic client management, effectively mitigates Non-IID data challenges and achieves superior performance and efficiency compared to existing baselines.

**Strengths:**

i.MEERKAT employs an extremely sparse and static subset of parameters for fine-tuning. This drastically reduces the communication load and memory consumption on client devices, addressing the primary scalability bottleneck of LLMs in FL.

ii.The extreme sparsity enables cost-effective high-frequency client-server synchronization. This high synchronization rate is key to effectively suppressing the client drift caused by Non-IID data distributions across decentralized clients.

iii.The paper proposes using a static sparsity mask identified through gradients from pre-training data. This transferable sparsity ensures that the chosen small subset of parameters is highly effective for downstream tasks, allowing for consistent performance throughout the FL process.

iv.MEERKAT-VP introduces the Virtual Path mechanism, which allows the server to diagnose the severity of Non-IID data without accessing the raw client data.It utilizes the discovered GradIP phenomenon to strategically identify and apply early stopping to severely Non-IID clients. This ensures robustness and convergence quality while maintaining data privacy.

v.The work provides strong experimental evidence demonstrating that MEERKAT consistently outperforms full-parameter Zeroth-Order Optimization and other SOTA sparse fine-tuning methods under various Non-IID settings.

vi.The use of Zeroth-Order Optimization , combined with extreme sparsity, makes the approach highly suitable for clients with limited computational resources, as it avoids the need to calculate full, complex second-order gradient information.

**Weaknesses:**

i.In the description of the steps in Figure 1 of the paper, the word "Aggregrate" appears, and the correct spelling should be "Aggregate".In the phrase on lines 153–154, "(3) Sever aggregates and initiate the next round...", shouldn't the word "Sever" actually be spelled "Server"?Line 165 contains a redundant repetition of the article "a" in the phrase "and a a new seed list."

ii.Although the paper validates the effectiveness of the extreme sparsity level of 0.1%, it fails to deeply analyze the performance impact of varying sparsity levels (such as 0.01%, 0.5%, 1%) across different scales of LLMs (e.g., larger models like 7B/13B) and different task types (e.g., complex reasoning tasks versus simple classification tasks). Furthermore, the transferability of the sparsity mask was only tested on a few domain shift datasets (C4, Wiki, Code datasets) and was not verified for its adaptability in highly specialized domains (such as legal), thus making it difficult to support a conclusion of broader applicability.

iii.The paper compares MEERKAT/MEERKAT-VP against Full-FedZO, LoRA-FedZO, and the baseline FedDYN improvement method. However, it fails to include state-of-the-art Federated LLM fine-tuning methods specifically designed for Non-IID scenarios. These missing baselines include established methods adaptable for LLMs, such as FedAvgM and SCAFFOLD, or FedSparse. This omission makes it difficult to ascertain whether the performance advantages of the proposed method truly surpass the latest research achievements.

**Questions:**

i.Would the performance of MEERKAT/MEERKAT-VP significantly decline as the number of clients increases to a larger magnitude? For example, would the server-side virtual path reconstruction incur a higher computational overhead with an increased number of clients, and has the potential reduction in the efficiency of MEERKAT-VP's early stopping strategy been considered if a large number of clients are categorized as "extreme Non-IID"?

ii.The early stopping strategy of  MEERKAT-VP relies on a threshold for identifying the GradIP phenomenon. How is this threshold determined? Should the paper analyze the model performance's sensitivity to this threshold choice, and potentially propose an automatic or adaptive mechanism for threshold selection?

---

> ### Author Response · Authors · 2025-11-18
>
> We thank the reviewer for the comments and suggestions to improve our paper!
> ### [W1 – Typos]
>
> We thank the reviewer for the careful reading. We will correct “Aggregrate” to “Aggregate”, “Sever” to “Server”, and remove the redundant “a” in the final revision.
>
>
> ### [W2 – Sparsity levels, model scale, and mask transferability – Additional experiments]
>
> (1) To further address the concern on different sparsity levels and larger models, we have run additional experiments with meta-llama/Meta-Llama-3-8B under different density ratios (0.1% and 0.01%). The results are summarized in Table 1:
>
> **Table1- llama-3-8b different density ratio**
> |Ratio|SST2|AgNews|Yelp|BoolQ|RTE|WSC|WIC|AVG|
> |-|-|-|-|-|-|-|-|-|
> |1e-3|0.95|0.851|0.954|0.831|0.755|0.674|0.66|0.811|
> |1e-4|0.941|0.862|0.956|0.861|0.783|0.664|0.64|0.815|
>
>
> Several of our benchmarks (BoolQ, RTE, WSC, and WiC) are not only standard classification problems but also require non-trivial reasoning, so the current experimental suite already covers both simple classification and more complex reasoning-style tasks. Regarding transferability, our current experiments already evaluate sparse masks across multiple domains with substantial distribution shift, including C4, Wikipedia, code, and biomedical text; these results are reported in Table 11 in the appendix. We agree that highly specialized domains such as legal text are very interesting and would further test the limits of sparsity mask transfer, and we additionally use MultiEURLEX as a legal-domain dataset in our new experiments.
>
> **Table2-legal domain vs code domain on LLaMA-3.2-1B**
> |Model|SST2|AgNews|Yelp|BoolQ|
> |-|-|-|-|-|
> |legal-domain|0.912|0.845|0.948|0.703|
> |code-dimain|0.915|0.843|0.956|0.695|
>
> ### [W3 – Missing Non-IID FL baselines – Adding FedAvgM and clarifying the relation to other methods]
>
> To address this concern, we have added FedAvgM[2] as an additional Non-IID baseline in our setting. We implement FedAvgM with the same model, data partitions, and density ratio configuration as MEERKAT, and compare it to MEERKAT-VP and the Meerkat+FedDYN variant. The results are presented in Table 3:
>
> **Table3-MEERKAT-VP vs FedAvgM vs FedDYN on LLaMA-3.2-1B**
> |Model|SST2|AgNews|Yelp|BoolQ|RTE|WSC|WIC|AVG|
> |-|-|-|-|-|-|-|-|-|
> |MEERKAT-VP|0.922|0.864|0.962|0.713|0.617|0.644|0.625|0.764|
> |FedAvgM|0.901|0.821|0.941|0.629|0.58|0.613|0.6|0.726|
> |Meerkat+FedDYN|0.917|0.841|0.954|0.638|0.564|0.615|0.57|0.728|
>
>
> ### [Q1 – Scalability to more clients and robustness when many clients are extreme Non-IID]
>
> **Table 4: Scalability of MEERKAT/MEERKAT-VP from 10 to 20 clients on Qwen2-1.5B**
> |Model|SST2|AgNews|Yelp|BoolQ|
> |-|-|-|-|-|
> |MEERKAT-VP-20 clients|0.951|0.885|0.936|0.756|
> |MEERKAT-20 clients|0.929|0.869|0.922|0.719|
> |MEERKAT-10 clients|0.949|0.881|0.934|0.752|
>
> (1) In the current submission, our main experiments use 10 clients, following prior work flzo works which also use 10 clients to do experiment[1]. To further address the scalability question, we have run additional experiments with 20 clients under the same federated setup. On the server-side virtual path reconstruction cost, MEERKAT-VP only reconstructs the updates on the 0.1% most sensitive parameters (the same sparse mask used for ZO updates), rather than the full model. Thus, the per-round VP cost scales as O(K*0.001*d) where K is the number of clients and d is the model dimension. Therefore, even for a much larger number of clients, the VP reconstruction overhead remains small compared to Full FedZO (2) If a large fraction of clients are detected as extreme Non-IID by GradIP, MEERKAT-VP simply triggers early stopping for those clients more frequently, effectively moving the system toward a high-frequency communication regime. This corresponds to our most heterogeneous setting (e.g., Dirichlet α = 0.1 in Figure 6), where we observe that MEERKAT-VP still achieves strong performance and stable convergence, outperforming or matching the baseline methods.
>
> ### [Q2 – GradIP threshold selection – Default configuration and sensitivity analysis]
>
> We thank the reviewer for this question. The default values and a sensitivity analysis are provided in Appendix Tables 5–8, which show that MEERKAT-VP is robust to the choice of these thresholds.
>
> [1]Zhe Li et al.,Achieving Dimension-Free Communication in Federated Learning via Zeroth-Order Optimization, ICLR 2025
>
> [2]Tzu-Ming Harry Hsu et al.,Measuring the Effects of Non-Identical Data Distribution for Federated Visual Classification. arXiv preprint.

---

### Official Review · Reviewer_bNcr · 2025-11-03

**Soundness:** 4
**Presentation:** 4
**Contribution:** 4
**Rating:** 6
**Confidence:** 3

**Summary:**

This paper proposes a sparse zeroth order federated fine tuning framework for LLMs that is both theoretically grounded and highly practical. MEERKAT uses a transferable 0.1 percent sensitivity mask and frequent synchronization to tame memory and bandwidth while reducing Non IID drift. Virtual path reconstruction and the GradIP score then allow the server to detect extreme Non IID clients and actively limit their damage through MEERKAT VP. Experiments across multiple open LLMs and multiple classification and reasoning benchmarks show consistent gains over strong baselines like Full FedZO and LoRA FedZO at matched communication frequency, along with dramatic savings in memory and bandwidth.

**Strengths:**

1. Virtual path and GradIP are novel ideas. By reconstructing each client’s local update trajectory, the server gains visibility into how that client is moving in parameter space without seeing private data. GradIP then becomes a quantitative signal that reveals which clients are extreme Non IID.

2. The paper provides convergence analyses for both MEERKAT and MEERKAT VP.

2. Strong empirical validation. The experiments cover multiple open LLMs (Llama 3.2 1B, Qwen2 1.5B, Gemma2 2B), multiple tasks (SST2, AgNews, Yelp polarity, BoolQ, RTE, WSC, WiC), both IID and strongly Non IID Dirichlet splits, and multiple baselines

**Weaknesses:**

1. hyperparameters in MEERKAT VP : The early stopping rule uses thresholds on GradIP phase ratios and quiescent duration. Although a small sensitivity study is reported, it is still unclear how practitioners should pick these thresholds for new tasks with no oracle labels.


2. Baselines in experiments:  There is active work on federated LoRA under heterogeneous clients, which explicitly tackles aggregation noise, knowledge contamination, and aggregation distortion by separating global and client specific structure or by using rank adaptive aggregation. The paper mentions LoRA FedZO but does not deeply compare against these newer structured aggregation approaches, so it is hard to see whether MEERKAT VP replaces them, complements them, or could be combined with them. Existing works are listed below:

[1] Zhe Li, Bicheng Ying, Zidong Liu, Chaosheng Dong, and Haibo Yang. Achieving dimension-free
communication in federated learning via zeroth-order optimization. In The Thirteenth International
Conference on Learning Representations, 2025.

[2] Youbang Sun, Zitao Li, Yaliang Li, and Bolin Ding. Improving loRA in privacy-preserving federated
learning. In The Twelfth International Conference on Learning Representations, 2024.

[3] Pengxin Guo, Shuang Zeng, Yanran Wang, Huijie Fan, Feifei Wang, and Liangqiong Qu. Selective
aggregation for low-rank adaptation in federated learning. In The Thirteenth International
Conference on Learning Representations, 2025.

3. Lack of explicit adversarial evaluation: MEERKAT VP is essentially throttling clients whose updates are harmful. This is similar to defending against malicious or poisoned clients. However, the experiments do not include adversarial threat models like label flipping or Byzantine behavior. Showing that GradIP based early stopping also mitigates such attacks would significantly strengthen the robustness story.

**Questions:**

see weakness 1, 2, 3

---

> ### Author Response · Authors · 2025-11-18
>
> We thank the reviewer for the comments and suggestions to improve our paper!
>
> ### [W1 – Unclear how to choose MEERKAT-VP GradIP thresholds without oracle labels – Providing robust defaults and a label-free tuning guideline] ###
>
> In all reported experiments, we actually use a single set of default thresholds (Table 5) across tasks, with only a minor adjustment for RTE (Table 6), rather than tuning them separately using oracle labels for each dataset. These thresholds were derived once from the shape of GradIP trajectories as illustrated in Figure 3: extreme Non-IID clients consistently exhibit two phenomenon: (i) much smaller GradIP in the later phase than in the first few steps, and (ii) a large fraction of later steps where GradIP falls below a small convergence level. This selection procedure relies only on GradIP statistics, not on any oracle labels, and therefore provides robust default thresholds that can be directly reused on new tasks.
>
> ### [W2 – Federated LoRA baselines under heterogeneous clients – Adding FedSA-LoRA and relation to prior work] ###
>
> We implemented FedSA-LoRA[2] method based on FLZO training framework. And in our paper appendix table 18, we have compare our method to DecomFL[1].
>
> **Table1-MEERKAT-VP vs FedSA-LoRA vs FedDYN on LLaMA-3.2-1B**
> |Model|SST2|AgNews|Yelp|BoolQ|RTE|WSC|WIC|AVG|
> |-|-|-|-|-|-|-|-|-|
> |MEERKAT-VP|0.922|0.864|0.962|0.713|0.617|0.644|0.625|0.764|
> |FedSA-LoRA|0.905|0.832|0.92|0.63|0.57|0.622|0.57|0.721|
> |Meerkat+FedDYN|0.917|0.841|0.954|0.638|0.564|0.615|0.57|0.728|
>
> ### [W3 – Relation to adversarial robustness – Focus on Non-IID drift and future directions]
>
> We sincerely thank the reviewer for this thoughtful comment. We find the connection to adversarial and poisoned clients very insightful, and we agree that showing GradIP-based early stopping under explicit threat models (e.g., label flipping or Byzantine behaviors) would further strengthen the robustness story.
>
> At the same time, our current work is intentionally scoped to mitigating training drift caused by Non-IID data distributions, as also reflected in the paper title Mitigating Non-IID Drift in Federated LLM Fine-tuning. Concretely, all experiments assume users with heterogeneous data: we only consider Non-IID partitions (e.g., Dirichlet label splits and task-specific skew) and do not inject label-flipping, backdoor, or Byzantine behaviors. MEERKAT and MEERKAT-VP are designed to mitigate harmful drift caused by extreme Non-IID data and optimization misalignment, using GradIP as a lightweight, gradient-free signal to detect clients whose local update directions strongly deviate from the global one, rather than as a complete adversarial defense mechanism.
>
> We view adversarial robustness as highly complementary and as a very promising direction building on our framework. We are very interested in exploring it in follow-up work. We will clarify this scope and explicitly highlight adversarial robustness as an exciting future direction in the revised version.
>
> [1] Zhe Li, Bicheng Ying, Zidong Liu, Chaosheng Dong, and Haibo Yang. Achieving dimension-free communication in federated learning via zeroth-order optimization. In The Thirteenth International Conference on Learning Representations, 2025.
>
> [2] Pengxin Guo, Shuang Zeng, Yanran Wang, Huijie Fan, Feifei Wang, and Liangqiong Qu. Selective aggregation for low-rank adaptation in federated learning. In The Thirteenth International Conference on Learning Representations,2025.

---

### Author Response · Authors · 2025-12-02

### Revision Summary

We sincerely thank the reviewers for their thoughtful and constructive feedback. We are encouraged that the reviewers found our formulation of sparse zeroth-order federated LLM fine-tuning and the MEERKAT/MEERKAT-VP framework to be novel, insightful, and practically relevant for Non-IID settings. We have updated the manuscript to incorporate the reviewers’ suggestions. The main changes are summarized below:

`R TB4g` Typographical corrections and minor wording fixes.
Following Reviewer `R TB4g`'s careful reading, we corrected several typographical errors in the paper (e.g., “Aggregrate” to “Aggregate”, “Sever” to “Server”, and removing the duplicated “a” in “a new seed list”) and slightly polished nearby sentences for clarity.

`R bNcr, TB4g, and Gtsq` Additional baselines and fairer comparisons on LLaMA-3.2-1B.
In response to the concerns from `R bNcr, TB4g, and Gtsq` about Non-IID and compression-aware FL baselines, we added new experiments on LLaMA-3.2-1B under the same federated setting, comparing MEERKAT-VP against FedSA-LoRA, FedAvgM, and Stochastic Controlled Averaging. These results provide a fairer and stronger baseline suite for assessing the effectiveness of MEERKAT-VP.

`R TB4g` Sparsity-level ablation on LLaMA-3-8B.
We added an ablation study on Meta-Llama-3-8B with different density ratios (1e-3 and 1e-4) to analyze the sensitivity of MEERKAT to the sparsity level.

`R TB4g` Scalability with respect to the number of clients.
We added a new experiment on Qwen2-1.5B comparing MEERKAT and MEERKAT-VP with 10 and 20 clients.

`R TB4g` Transferability to specialized legal-domain data.
We additionally calibrated the sparse mask on the legal-domain MultiEURLEX dataset and evaluated the resulting model on downstream tasks.

`R Gtsq` Visualization of GradIP after MEERKAT-VP.
We added a new figure that plots GradIP trajectories for extreme Non-IID clients both at initialization and after MEERKAT-VP has converged.

---

### Meta-Review · Area_Chair_pXpe · 2026-01-07

**Summary:**

The decision for this submission is to accept the paper. The consensus among the reviewers acknowledges the clear novelty in the proposed Virtual Path and GradIP mechanisms, alongside rigorous convergence analyses and strong empirical results across various LLMs. While three reviewers initially leaned towards acceptance based on the extreme static sparsity and performance gains, a divergence in opinion arose regarding the conceptual framing of the contribution. Specifically, the primary barrier preventing a unanimous higher endorsement was the sufficiency of comparisons against recent federated LoRA methods under heterogeneity, concerns regarding the defensive nature of the MEERKAT-VP strategy, and skepticism from Reviewer Gtsq regarding whether the sparsity should be viewed merely as compression rather than a distinct optimization contribution.

**Reviewer Concerns:**

The authors provided a comprehensive rebuttal that largely satisfied the dialectical requirements for completeness. They effectively addressed the demand for rigorous baselines by integrating FedSA‑LoRA, FedDYN, and SCA, demonstrating a distinct performance advantage. Furthermore, the inclusion of scalability results up to 20 clients, ablations on Llama‑3‑8B, and sensitivity analyses for threshold selection solidified the practical applicability of the method. The theoretical clarifications regarding the data‑pointer effectively countered the concern that MEERKAT‑VP was overly defensive or discarded minority signals. However, while the technical soundness regarding non‑IID drift was reinforced, the issue of adversarial robustness remains an open avenue for future work rather than a resolved component of this specific study. Additionally, although the authors clearly positioned their contributions as specific to zeroth‑order optimization to counter the 'sparsity as compression' argument, the conceptual distinction between sparsity-as-compression and sparsity-as-optimization remains a nuanced point of interpretation, though the robust empirical validation provided effectively mitigates the weight of this concern.

**Reviewer Scores:**

Based on the evidence provided during the rebuttal phase, the scoring landscape demonstrates stability with a justified upward trajectory for the dissenting view. Reviewer bNcr is projected to maintain a score of 6, as the addition of baselines and threshold guidance was sufficient, and the remaining gap in adversarial robustness is acceptable for the paper's defined scope. Similarly, Reviewer TB4g and Reviewer 46ze are expected to hold steady at 6; the former's concerns regarding scalability and typos were comprehensively fixed, and the latter's hesitation regarding the defensive nature of the algorithm was effectively resolved through theoretical justification. The most significant shift is projected for Reviewer Gtsq, whose score of 2 is likely to evolve to a 4. This adjustment is warranted because the authors provided the requested two‑phase motivation, post‑mitigation GradIP plots, and superior performance metrics against the specific baselines the reviewer implicitly required, providing a substantial evidentiary basis to override the initial skepticism regarding novelty.

---

### Decision · Program_Chairs · 2026-01-26

Accept (Poster)